# Riemannian Flow Matching for Brain Connectivity Matrices via Pullback Geometry

**Antoine Collas**[*]**, Ce Ju, Nicolas Salvy, Bertrand Thirion**
Inria, CEA, Université Paris-Saclay
Palaiseau, France
[*]`antoine.collas@inria.fr`

## Abstract

Generating realistic brain connectivity matrices is key to analyzing population heterogeneity in brain organization, understanding disease, and augmenting data in challenging classification problems. Functional connectivity matrices lie in constrained spaces—such as the set of symmetric positive definite or correlation matrices—that can be modeled as Riemannian manifolds. However, using Riemannian tools typically requires redefining core operations (geodesics, norms, integration), making generative modeling computationally inefficient. In this work, we propose DIFFEOCFM, an approach that enables conditional flow matching (CFM) on matrix manifolds by exploiting pullback metrics induced by global diffeomorphisms on Euclidean spaces. We show that Riemannian CFM with such metrics is equivalent to applying standard CFM after data transformation. This equivalence allows efficient vector field learning, and fast sampling with standard ODE solvers. We instantiate DIFFEOCFM with two different settings: the matrix logarithm for covariance matrices and the normalized Cholesky decomposition for correlation matrices. We evaluate DIFFEOCFM on three large-scale fMRI datasets with more than 4600 scans from 2800 subjects (ADNI, ABIDE, OASIS-3) and two EEG motor imagery datasets with over 30000 trials from 26 subjects (BNCI2014-002 and BNCI2015-001). It enables fast training and achieves state-of-the-art performance, all while preserving manifold constraints.
Code: `https://github.com/antoinecollas/DiffeoCFM`

## 1 Introduction

**Brain imaging connectivity and Riemannian geometry**  Modern neuroimaging analyses map brain functional signals toward *connectivity matrices*—covariance or correlation estimates between regions of interest or sensor channels [37]. These structured representations are used in many applications, such as motor imagery classification [4, 51, 41, 44], brain age prediction [16, 22, 54, 55], or disease diagnosis [15]. They are central to the analysis of signals from many neuroimaging modalities, such as functional magnetic resonance imaging (fMRI), electroencephalography (EEG), and magnetoencephalography (MEG). Brain connectivity matrices are *symmetric positive definite* (SPD) or lie in the open *elliptope* of full-rank correlation matrices, and thus belong to smooth matrix manifolds—$\mathbb{S}_d^{++}$ and its submanifold $\text{Corr}_d$—defined respectively by

$$\mathbb{S}_d^{++} = \left\{ \boldsymbol{\Sigma} \in \mathbb{R}^{d \times d} \,\middle|\, \boldsymbol{\Sigma}^\top = \boldsymbol{\Sigma}, \boldsymbol{\Sigma} \succ 0 \right\} \text{ and } \text{Corr}_d = \left\{ \boldsymbol{\Sigma} \in \mathbb{S}_d^{++} \,\middle|\, \text{diag}(\boldsymbol{\Sigma}) = \mathbf{1} \right\}. \tag{1}$$

Several Riemannian metrics have been proposed to analyse these data such as the affine-invariant metric [65, 7], the log-Euclidean metric [3], or the Euclidean-Cholesky metric [68]. They enable the definition of Riemannian operations such as geodesics, exponential maps, and parallel transport, which extend standard Euclidean operations to the manifold. Building on these manifolds, a wide

39th Conference on Neural Information Processing Systems (NeurIPS 2025).

range of machine learning algorithms have been designed for classification [4], regression [16, 22, 55], or dimension reduction [27].

**Deep generative models on manifolds**  Deep generative modeling has rapidly advanced with the success of generative adversarial networks (GANs)[28], variational autoencoders (VAEs)[39], autoregressive models such as PixelRNN [70], normalizing flows [21, 40], large-scale autoregressive models [9], and more recently, diffusion models [31, 66] and flow matching [49, 47, 2, 48]. These last two methods learn to synthesize data by estimating continuous-time stochastic (diffusion) or deterministic (flow) dynamics that interpolate between a source and a target distribution. Diffusion models do so by reversing a noise injection process, while flow matching aligns a learned time-dependent vector field to the velocity field of a straight-line path. Both paradigms offer good scalability and state-of-the-art results on diverse domains, from natural images [23], to speech [43], and protein structure generation [32]. More recent efforts have extended generative modeling to Riemannian manifolds. Riemannian score-based generative modeling [17] and Riemannian flow matching [12] provide general formulations for sampling on manifolds. The latter learns time-dependent vector fields on the tangent bundle that match the velocity of geodesic paths between source and target distributions. These approaches offer principled tools for generating data with geometric constraints, by treating the data domain as a manifold. These novel frameworks have already been applied to several applications such as materials discovery [56], robotics [20], or climate science [12].

**Contributions**  In this work, we address a novel and challenging issue in neuroimaging: generating realistic brain connectivity data from actual human neuroimaging data. This challenge stems from the unique structure of brain connectivity data, which is represented by SPDs or correlation matrices that lie on non-Euclidean manifolds. Moreover, neuroimaging datasets typically have limited sample sizes, making realistic data generation particularly valuable. We propose a novel Riemannian flow matching method, DIFFEOCFM, based on pullback geometry, defined by a global diffeomorphism $\phi : \mathcal{M} \to E$, where $E$ is Euclidean space. This method is an efficient framework that *guarantees manifold-constrained outputs by construction while avoiding computationally expensive operations specific to SPD or correlation manifolds*. We instantiate DIFFEOCFM with two diffeomorphisms tailored to different neuroimaging data: the matrix logarithm for SPD matrices and the normalized Cholesky decomposition for correlation matrices. Finally, we evaluate DIFFEOCFM on three large-scale fMRI datasets (ADNI, ABIDE, and OASIS-3; over 4600 scans from 2800 subjects) and two EEG motor imagery datasets (BNCI2014-002 and BNCI2015-001; 30000 trials from 26 subjects), demonstrating that DIFFEOCFM is capable of generating realistic, neurophysiologically meaningful samples, as validated by multiple statistical metrics.

**Notations**  $\mathcal{M}$ is a smooth manifold with tangent space $T_x\mathcal{M}$ and Riemannian norm $\|\cdot\|_x$. We write $\gamma$ for geodesics, and $\dot{\gamma}(t) \triangleq \frac{d}{dt}\gamma(t)$ for curve's speed. Let $\phi : \mathcal{M} \to E$ be a global diffeomorphism to Euclidean space $E$, with differential $\mathrm{D}\,\phi(x) : T_x\mathcal{M} \mapsto E$ and inverse $(\mathrm{D}\,\phi(x))^{-1}$. The pushforward of a distribution $p$ on $\mathcal{M}$ is $\phi_{\#}p$, defined via $\int f\,d(\phi_{\#}p) = \int (f \circ \phi)\,dp$ for any $f$ continuous on $E$. We denote $x \mid y \sim p(\cdot \mid y)$ for conditional sampling with label $y \in \mathcal{Y}$. Let $\mathbb{S}_d$ be the space of symmetric $d \times d$ matrices, $\mathbb{S}_d^{++}$ the SPD cone, and $\mathrm{Corr}_d$ the set of correlation matrices. Let $\mathrm{LT}_d^1$ be the set of lower-triangular matrices with unit diagonal. We define $\mathrm{vec}_{\mathrm{sl}} : \mathbb{S}_d \mapsto \mathbb{R}^{d(d-1)/2}$ as the operator extracting strictly lower-triangular entries, and $\mathrm{vec}_{\mathrm{lt}}$ for the full lower-triangular part (diagonal included), with $\sqrt{2}$ scaling off-diagonal terms.

## 2  Background

**Pullback Manifolds with Euclidean Spaces**  Let $\mathcal{M}$ be a smooth manifold, $E$ a Euclidean space, and $\phi : \mathcal{M} \to E$ a global diffeomorphism (a smooth bijection with a smooth inverse). The diffeomorphism $\phi$ induces a Riemannian metric on $\mathcal{M}$ by pulling back the Euclidean metric $g_E$ on $E$

$$(\phi^* g_E)_x(\xi, \eta) \triangleq g_E\left(\mathrm{D}\,\phi(x)[\xi], \mathrm{D}\,\phi(x)[\eta]\right), \quad \xi, \eta \in T_x\mathcal{M}. \tag{2}$$

This metric induces a Riemannian norm on the tangent space $T_x\mathcal{M}$ at each point $x \in \mathcal{M}$: $\|\xi\|_x = \sqrt{(\phi^* g_E)_x(\xi, \xi)}$. The pair $(\mathcal{M}, \phi^* g_E)$ is then called a *pullback manifold*, and $\phi^* g_E$ is the *pullback metric* of $g_E$ by $\phi$. The pullback manifold is geodesically complete and admits globally unique

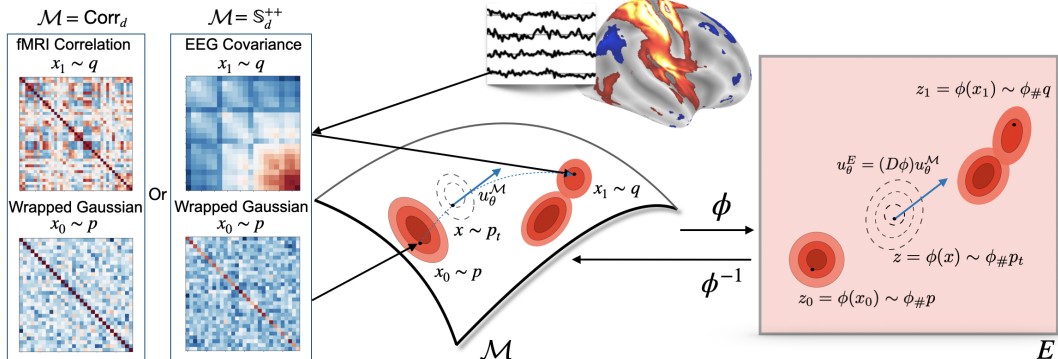

Figure 1: **Overview of DIFFEOCFM.** DIFFEOCFM is a principled framework for *deep generative modeling* on matrix manifolds. It reformulates Riemannian conditional flow matching (CFM) on a pullback manifold $(\mathcal{M}, \phi^* g_E)$ as *conventional CFM in Euclidean space* $E$, via a global diffeomorphism $\phi: \mathcal{M} \to E$. The reformulation preserves geometry in two ways: *(i)* the learned Euclidean vector field $u_\theta^E$ satisfies (7), ensuring that training $u_\theta^E$ is equivalent to training $u_\theta^{\mathcal{M}}$; *(ii)* the flow trajectories obey $\phi(x(t)) = z(t)$, so that integrating in $E$ and pulling back via $\phi^{-1}$ yields the same samples as integrating directly on $\mathcal{M}$. This allows both training and sampling to be carried out efficiently in $E$, while remaining equivalent to operating on $\mathcal{M}$. On the left, fMRI correlation and EEG spatial covariance matrices lie on $\mathcal{M} = \text{Corr}_d$ and $\mathcal{M} = \mathbb{S}_d^{++}$, respectively. These matrices are mapped to $E$ through $\phi$, a time-dependent vector field $u_\theta^E$ is trained in $E$, and integration is performed in $E$ before mapping back via $\phi^{-1}$ to yield connectivity manifold constrained matrices.

geodesics [68, Chap. 7]. Moreover, many Riemannian operations reduce to simple computations in $E$. Given two points $x_0, x_1 \in \mathcal{M}$, the geodesic $\gamma: [0, 1] \to \mathcal{M}$ connecting $x_0$ to $x_1$ and its associated riemannian distance are given by

$$\gamma(t) = \phi^{-1}\left((1 - t)\phi(x_0) + t\phi(x_1)\right) \text{ and } d_{\mathcal{M}}(x_0, x_1) = \|\phi(x_0) - \phi(x_1)\|_E \qquad (3)$$

i.e., the pullbacks of the Euclidean straight line and distance in $E$ joining $\phi(x_0)$ and $\phi(x_1)$. The Fréchet mean [29] of a set of points $\{x^{(n)}\}_{n=1}^N \subset \mathcal{M}$ with respect to the Riemannian distance is

$$\bar{x} \triangleq \arg\min_{x \in \mathcal{M}} \sum_{n=1}^N d_{\mathcal{M}}(x, x^{(n)})^2 = \phi^{-1}\left(\frac{1}{N} \sum_{n=1}^N \phi(x^{(n)})\right). \qquad (4)$$

**Riemannian CFM** CFM [49, 47, 2, 48] was recently extended to Riemannian manifolds [12], providing a principled framework for learning time-dependent vector fields that transport samples between probability distributions defined on such spaces. Given a manifold $\mathcal{M}$, a vector field $u_\theta^{\mathcal{M}}: [0, 1] \times \mathcal{M} \times \mathcal{Y} \to T\mathcal{M}$ is trained to match the velocity of geodesics connecting samples from source and target distributions. The Riemannian CFM loss is defined as

$$\mathcal{L}(\theta) \triangleq \mathbb{E}_{t, y, x_0|y, x_1|y} \left\|u_\theta^{\mathcal{M}}(t, \gamma(t), y) - \dot{\gamma}(t)\right\|_{\gamma(t)}^2, \qquad (5)$$

where $y \in \mathcal{Y}$ is a condition variable (such as a disease status), and $x_0|y \sim p(\cdot|y)$, $x_1|y \sim q(\cdot|y)$ and $t \sim \mathcal{U}([0, 1])$. Hence, this loss is computationally intensive, as it requires evaluating geodesics $\gamma(t)$ between $x_0$ and $x_1$, their derivatives $\dot{\gamma}(t)$, and Riemannian norms $\|\cdot\|_{\gamma(t)}$. Once trained, new samples on $\mathcal{M}$ are generated by solving the Riemannian ODE

$$\dot{x}(t) = u_\theta^{\mathcal{M}}(t, x(t), y), \quad x(0) = x_0 \sim p(\cdot \mid y) \qquad (6)$$

and returning $x(1)$ as a sample from the learned approximation of $q(\cdot \mid y)$.

## 3 DIFFEOCFM: Conditional Flow Matching on Pullback Manifolds

Pullback manifolds provide a natural setting for using Riemannian CFM [12] in practical generative modeling tasks. Indeed, when $\mathcal{M}$ is equipped with a pullback metric $\phi^* g_E$ induced by a global

diffeomorphism $\phi : \mathcal{M} \to E$, both training and sampling can be performed entirely in the Euclidean space $E$. We prove that this equivalence is *exact*: the learned vector field in $E$ corresponds to one on $\mathcal{M}$, and the ODE solutions in $E$ map to those on $\mathcal{M}$ via $\phi$. This result motivates DIFFEOCFM, a conditional flow matching framework that performs all computations in $E$, avoiding costly geometric operations—such as computing geodesics, Riemannian norms, or manifold integration—while guaranteeing manifold-constrained outputs. An overview of the method is shown in Figure 1.

## 3.1 Training and sampling with a diffeomorphism

**Training**    Rather than learning a vector field $u_\theta^{\mathcal{M}}$ directly on the manifold $\mathcal{M}$, DIFFEOCFM trains its Euclidean counterpart $u_\theta^E$ via the pullback:

$$u_\theta^E(t, z, y) \triangleq \mathrm{D}\,\phi(\phi^{-1}(z)) \left( u_\theta^{\mathcal{M}}(t, \phi^{-1}(z), y) \right). \tag{7}$$

Indeed, in this case, the loss function (5) can be expressed in terms of the Euclidean vector field $u_\theta^E$ as shown in the following proposition.

**Proposition 1** (Riemannian CFM loss function on pullback manifolds). *The Riemannian CFM loss* (5) *can be re-expressed in terms of the Euclidean vector field $u_\theta^E$* (7) *as*

$$\mathcal{L}(\theta) = \mathbb{E}_{t,\, y,\, z_0|y,\, z_1|y} \left\| u_\theta^E\left(t, (1-t)z_0 + tz_1, y\right) - (z_1 - z_0) \right\|_E^2,$$

*where $z_0|y \sim \phi_\# p(\cdot|y)$ and $z_1|y \sim \phi_\# q(\cdot|y)$.*

It should be noted that this new loss function is much simpler to compute than the original Riemannian CFM one, as it does not require computing geodesics, their derivatives, or Riemannian norms.

**Sampling**    DIFFEOCFM generates new samples by solving the ODE

$$\dot{z}(t) = u_\theta^E\left(t, z(t), y\right), \quad z(0) = z_0 \sim \phi_\# p(\cdot \mid y). \tag{8}$$

Despite this simple form, the procedure is fully Riemannian: the generated trajectory corresponds exactly to a manifold-valued solution under $\phi^{-1}$. Indeed, the following proposition states the equivalence of the solutions of the ODEs (6) and (8).

**Proposition 2** (Equivalence of ODE solutions). *The solution $x(t)$ to* (6) *satisfies*

$$x(t) = \phi^{-1}(z(t)) \quad \text{for all } t \in [0, 1],$$

*where $z(t)$ is the solution of the ODE* (8) *with initial condition $z_0 = \phi(x_0)$.*

The previous result establishes that the Riemannian and ODEs define equivalent flows through the diffeomorphism $\phi$. In practice, these ODEs are solved numerically using explicit Runge–Kutta integrators. The next proposition shows that the equivalence also holds at the discrete level: applying the same Runge–Kutta scheme in $E$ or on $\mathcal{M}$ yields iterates related by $\phi$.

**Proposition 3** (Equivalence of Runge–Kutta iterates). *Let $\{x_\ell\}$ be the iterates produced on $\mathcal{M}$ by an explicit Riemannian Runge–Kutta scheme applied to the ODE* (6). *Then, the iterates are*

$$x_\ell = \phi^{-1}(z_\ell) \quad \text{for all } \ell \in \mathbb{N},$$

*where $\{z_\ell\}$ are the iterates obtained by applying the same scheme (same coefficients and step size) to the ODE* (8) *with initial condition $z_0 = \phi(x_0)$.*

Overall, the training and sampling algorithms for DIFFEOCFM are summarized in Algorithm 1 and 2, respectively.

| **Algorithm 1:** DIFFEOCFM: Training | **Algorithm 2:** DIFFEOCFM: Sampling |
|---|---|
| **Input:** step size $h$; samplers $\pi_\mathcal{Y}$, $\phi_\# p$, $\phi_\# q$ | **Input:** label $y$; steps $L$; step size $h$; trained $\theta^\star$ |
| **Output:** Trained parameters $\theta^\star$ | **Output:** Generated sample $x$ |
| Initialize $\theta$; | Sample $z_0 \sim \phi_\# p(\cdot \mid y)$; |
| **while** *not converged* **do** | **for** $\ell = 0$ **to** $L-1$ **do** |
| $\quad$ Sample $y \sim \pi_\mathcal{Y}$, $t \sim \mathcal{U}([0,1])$; | $\quad$ $z_{\ell+1} \leftarrow$ Runge-Kutta-step$(u_{\theta^\star}^E, z_\ell, y, h)$; |
| $\quad$ Sample $z_0 \sim \phi_\# p(\cdot \mid y)$, $z_1 \sim \phi_\# q(\cdot \mid y)$; | **end** |
| $\quad$ $\mathcal{L} \leftarrow \|u_\theta^E(t, (1-t)z_0 + tz_1, y) - (z_1 - z_0)\|_E^2$; | $x \leftarrow \phi^{-1}(z_L)$; |
| $\quad$ $\theta \leftarrow$ optimizer-step$(\mathcal{L})$; | |
| **end** | |

## 3.2 Diffeomorphic Embeddings for Generative Modeling of Brain Connectivity Matrices

To generate brain connectivity matrices, we map $\mathbb{S}_d^{++}$ and $\mathrm{Corr}_d$ to Euclidean spaces via global diffeomorphisms: the matrix logarithm for SPD matrices and the normalized Cholesky map for correlation matrices. These maps define pullback metrics on the tangent spaces $T_{\boldsymbol{\Sigma}}\mathbb{S}_d^{++} = \mathbb{S}_d$ and $T_{\boldsymbol{\Sigma}}\mathrm{Corr}_d = \{\boldsymbol{\xi} \in \mathbb{S}_d \mid \mathrm{diag}(\boldsymbol{\xi}) = 0\}$, of respective dimensions $d(d+1)/2$ and $d(d-1)/2$. This allows DIFFEOCFM to perform efficient, geometry-aware generation for both matrix types. Note that we selected these two diffeomorphisms for their simplicity and ease of implementation. However, other choices are possible. For correlation matrices, alternative parameterizations are discussed in [68, Chapter 7]. For SPD matrices, one can use the Cholesky factor with a logarithm applied to the diagonal, leading to the log-Cholesky metric [46], which also defines a global diffeomorphism.

**Covariance matrices: Log–Euclidean metric**  On $\mathbb{S}_d^{++}$, we define the global diffeomorphism $\phi_{\mathbb{S}_d^{++}} : \mathbb{S}_d^{++} \mapsto \mathbb{R}^{d(d+1)/2}$ by composing the matrix logarithm with the vectorization map $\mathrm{vec}_{\mathrm{lt}}$:

$$\phi_{\mathbb{S}_d^{++}}(\boldsymbol{\Sigma}) = \mathrm{vec}_{\mathrm{lt}}(\log(\boldsymbol{\Sigma})) \quad \text{and} \quad \phi_{\mathbb{S}_d^{++}}^{-1}(\boldsymbol{\eta}) = \exp(\mathrm{vec}_{\mathrm{lt}}^{-1}(\boldsymbol{\eta})), \tag{9}$$

where $\log$ and $\exp$ denote the matrix logarithm and exponential, respectively. This mapping induces the *Log–Euclidean metric* on $\mathbb{S}_d^{++}$ by pulling back the standard Euclidean inner product from $\mathbb{R}^{d(d+1)/2}$: $g_{\boldsymbol{\Sigma},\mathbb{S}_d^{++}}(\boldsymbol{\xi},\boldsymbol{\eta}) = \mathrm{tr}(\mathrm{D}\log(\boldsymbol{\Sigma})[\boldsymbol{\xi}]\,\mathrm{D}\log(\boldsymbol{\Sigma})[\boldsymbol{\eta}])$, as introduced in Arsigny et al. [3].

**Correlation matrices: Euclidean–Cholesky metric**  On $\mathrm{Corr}_d$, the matrix logarithm is no longer a diffeomorphism. More generally, defining a log-based diffeomorphism is nontrivial—for example, the Riemannian logarithm associated with the affine-invariant metric does not admit a closed-form expression in this setting (see Appendix F). Instead, we use the normalized Cholesky map, a global diffeomorphism onto $\mathrm{LT}_d^1$, the space of lower-triangular matrices with unit diagonal:

$$\mathrm{nchol}(\boldsymbol{\Sigma}) = \mathrm{diag}(\mathrm{chol}(\boldsymbol{\Sigma}))^{-1}\,\mathrm{chol}(\boldsymbol{\Sigma}), \tag{10}$$

where $\mathrm{chol}(\boldsymbol{\Sigma})$ is the unique Cholesky factor with positive diagonal. Its inverse is

$$\mathrm{nchol}^{-1}(\mathbf{L}) = \mathbf{D}^{-1/2}\mathbf{L}\mathbf{L}^\top\mathbf{D}^{-1/2}, \quad \text{with } \mathbf{D} = \mathrm{diag}(\mathbf{L}\mathbf{L}^\top). \tag{11}$$

Then, we define the diffeomorphism $\phi_{\mathrm{Corr}_d} : \mathrm{Corr}_d \mapsto \mathbb{R}^{d(d-1)/2}$ and its inverse as

$$\phi_{\mathrm{Corr}_d}(\boldsymbol{\Sigma}) = \mathrm{vec}_{\mathrm{sl}}(\mathrm{nchol}(\boldsymbol{\Sigma})) \quad \text{and} \quad \phi_{\mathrm{Corr}_d}^{-1}(\boldsymbol{\eta}) = \mathrm{nchol}^{-1}(\mathrm{vec}_{\mathrm{sl}}^{-1}(\boldsymbol{\eta})). \tag{12}$$

This map induces the *Euclidean–Cholesky metric*, a Riemannian metric obtained by pulling back the Euclidean metric from the vector space $\mathbb{R}^{d(d-1)/2}$ [68]: $g_{\boldsymbol{\Sigma},\mathrm{Corr}_d}(\boldsymbol{\xi},\boldsymbol{\eta}) = \mathrm{D}\,\phi(\boldsymbol{\Sigma})[\boldsymbol{\xi}]^\top\,\mathrm{D}\,\phi(\boldsymbol{\Sigma})[\boldsymbol{\eta}]$.

**Label, source, and target distributions**  To train DIFFEOCFM on labeled brain connectivity data, we define class-conditional source and target distributions in the Euclidean space $E$ induced by the diffeomorphism $\phi$. Given a dataset $\{(x^{(n)}, y^{(n)})\}_{n=1}^N$ of manifold-valued matrices, we map each sample to $E$ via $z^{(n)} = \phi(x^{(n)})$. For each class $y$, we fit a Gaussian distribution to the embedded samples $\{z^{(n)} : y^{(n)} = y\}$ to define the class-conditional source distribution $\phi_\# p(\cdot \mid y)$. The target distribution $\phi_\# q(\cdot \mid y)$ is defined as the empirical distribution over the same class-$y$ samples.

## 3.3 Related work

Denoising Diffusion Probabilistic Models (DDPMs) [31] and CFM [47, 2] have emerged as robust, state-of-the-art generative models in Euclidean spaces. Several extensions have been proposed to handle data that lie on Riemannian manifolds. These include Riemannian Score-Based Generative Modeling [17], SPD-DDPM [45], and Riemannian CFM [12]. These methods tailor the loss function and ODE/SDE solvers to the geometry of a specific manifold. However, this geometric fidelity comes at a cost: manifold-specific operations such as Riemannian gradients, exponential/logarithm maps, or parallel transport must be implemented to compute the loss function and the integration on the manifold. For instance, SPD-DDPM requires a specialized neural architecture, an SPDNet, which is computationally expensive and significantly slower to train than Euclidean counterparts; see Figure 4 in Appendix for a comparison. References [35] and [38] explore more general settings by learning bridge matches on arbitrary manifolds or data-driven Riemannian metrics, whereas our

approach focuses on Riemannian geometries defined via known pullback diffeomorphisms. In [26], the authors leverage reparameterisation with normalising flows to learn probability densities on Lie groups (non-Euclidean spaces), and thus can be seen as an early precursor to our approach.

In contrast, CorrGAN [52] offers a pragmatic approach to generating correlation matrices by training and sampling entirely in Euclidean space using a GAN. Geometric constraints, such as positive definiteness and unit diagonal, are enforced via post-processing. While the method is simple and fast, the post-processing step can change the generated data in unwanted ways and reduce their quality.

The proposed method combines the simplicity of Euclidean training with the rigor of Riemannian geometry by using a diffeomorphism $\phi$ to embed structured matrices, apply standard CFM, and map samples back. To the best of our knowledge, it is the only method that enables generation of both SPD and correlation matrices within a unified framework. A detailed comparison of baseline methods—highlighting their assumptions, strengths, and limitations—is provided in Appendix Section H.

## 4 Empirical benchmarks

These benchmarks were designed to evaluate whether the generated data *(i.)* match the test distribution and *(ii.)* enable classifiers trained on them to generalize to real data. This evaluation uses two human brain imaging modalities: *three fMRI datasets* and *two EEG datasets*.

### 4.1 Metrics

The metrics used are method-agnostic; that is, they are computed solely from the generated samples. They fall into two categories: (i) *quality metrics*, which assess how well the generated data approximate the real data distribution; and (ii) *classification accuracy score (CAS) metrics*, which evaluate the usefulness of generated data for training classifiers.

**Quality Metrics**  We assess how closely the generated samples align with the real data distribution using the $\alpha$-precision and $\beta$-recall metrics introduced by [1]. We compute these metrics using the `EvaGeM` library[1]. In contrast to [1], which uses a Deep neural network, `EvaGeM` employs a One-Class SVM, providing a more stable and hyperparameter-robust estimator across datasets. These metrics quantify the fidelity (how realistic the generated samples are) and the diversity (how well they span the true data distribution). We also report the harmonic mean of the two, denoted $\alpha, \beta$-F1.

**Classification accuracy score metrics**  We follow the CAS protocol [64], training a classifier on generated samples and evaluating it on real test data. Specifically, we assess classification utility using a logistic regression with `liblinear` solver, balances class weights and a 5-fold cross-validation to select the inverse regularization strength $C$ from the grid $\{10^{-4}, 10^{-3}, \ldots, 10^4\}$. High scores indicate that the generated data preserves task-relevant information. On fMRI datasets, the task is disease classification (control vs. patient), while for EEG, it is a two-class motor imagery problem in a brain–computer interface setting. We report ROC-AUC and F1 scores.

### 4.2 Datasets

The experiments include both fMRI and EEG datasets; additional details are provided in Appendix G.

**fMRI datasets**  We use three publicly available resting-state fMRI datasets. The ABIDE dataset [53] consists of 900 subjects (one scan each), including both neurotypical and autistic individuals with a mean age of 17 years, collected across 19 international sites. The ADNI dataset [73] comprises 1,900 scans from 900 older adults (mean age 74), covering normal ageing, mild cognitive impairment, and Alzheimer's disease. The OASIS-3 dataset [42] includes 1,000 subjects and 1,800 longitudinal sessions collected over 10 years, targeting healthy ageing and neurodegenerative conditions, with a mean participant age of 71 years. We $z$-score the time series and then compute *correlation matrices* (Corr$_d$) using the OAS estimator [15]. We report mean and standard deviations computed across 10 random train-test splits with subject-level grouping, ensuring that scans from the same subject do not appear in both training and test sets.

---

[1] `https://github.com/nicolassalvy/EvaGeM`

**EEG datasets** We use two publicly available EEG motor imagery datasets from the BCI competition. The BNCI2014-002 dataset [67] includes 13 subjects performing right-hand and feet imagery, recorded with 15 channels over 1 session, with 80 trials per class. The BNCI2015-001 dataset [25] comprises 12 subjects, 13 channels over 2 or 3 sessions, and 100 trials per class for the same motor imagery tasks. To process the raw data, we compute *covariance matrices* ($\mathbb{S}_d^{++}$) using the OAS estimator [13], following standard practices [51]. We report performance on a leave-one-session-out protocol on BNCI2014-002 and on cross-session experiments on BNCI2015-001. The reported standard deviations are computed over 5 inner splits and averaged over sessions.

## 4.3  Baselines

We present the baselines used for comparison with DIFFEOCFM, including an oracle baseline (REAL DATA) that treats test data as if it were generated samples.

**REAL DATA** This oracle baseline treats the test set as if it were generated data when computing metrics. It provides an upper bound on the achievable performance, both for quality and CAS metrics, showing the best any generative model could hope to match.

**DIFFEOGAUSS** Given a diffeomorphic embedding $\phi : \mathcal{M} \mapsto E$, we model each class-conditional distribution $q(\cdot \mid y)$ as the push-forward of a Gaussian $\mathcal{N}(\mu_y, \Sigma_y)$ fitted to the embedded training data $z^{(n)} = \phi(x^{(n)})$. Samples are drawn in $E$ and mapped back to $\mathcal{M}$ via $\phi^{-1}$, yielding a wrapped Gaussian distribution on the manifold [62, 18].

**TRIANGDDPM and TRIANGCFM** These baselines apply generative models to the lower-triangular part of SPD or correlation matrices, a common heuristic in manifold modeling [52]. For fMRI, they use the strictly lower-triangular entries ($\phi : \text{Corr}_d \mapsto \mathbb{R}^{d(d-1)/2}$); for EEG, all lower-triangular entries ($\phi : \mathbb{S}_d^{++} \mapsto \mathbb{R}^{d(d+1)/2}$). TRIANGDDPM uses a DDPM [2], while TRIANGCFM trains a vector field using the standard CFM loss. Since $\phi$ is not a diffeomorphism, generated matrices that do not lie on the manifold are projected back onto it. In particular, generated matrices are not necessarily positive definite, so we apply a projection to ensure all eigenvalues are at least $\epsilon > 0$ with $\Sigma_{\text{proj}} = (1 - \alpha)\Sigma + \alpha\mathbf{I}$; see Appendix M for more details. These methods trade geometric fidelity for simplicity, relying on post hoc projections to enforce constraints.

**RIEMCFM** This method applies Riemannian CFM [12] directly on the SPD manifold using the affine-invariant metric [65]. The target conditional vector field is computed analytically from the Riemannian logarithm and exponential maps. Unlike TRIANGDDPM/TRIANGCFM, RIEMCFM preserves the intrinsic geometry of $\mathbb{S}_d^{++}$ throughout training and sampling, yielding valid SPD matrices at every time step without post-hoc corrections. However, it requires computing geodesics, their derivatives and Riemannian norms under the affine-invariant metric, making it substantially more computationally expensive than all other presented methods. The reference implementation[2] focuses exclusively on the SPD manifold under the affine-invariant metric, and does not provide a corresponding construction for correlation matrices.

**DIFFEOCFM (proposed)** We apply the log–Euclidean map $\phi_{\mathbb{S}_d^{++}}$ (9) for EEG and the normalized Cholesky map $\phi_{\text{Corr}_d}$ (12) for fMRI in Algorithm 1 and 2. These diffeomorphisms allow Euclidean training with CFM while ensuring manifold-valid samples without post-processing.

**Deep learning and training/sampling setups** TRIANGDDPM, TRIANGCFM and DIFFEOCFM employ a two-layer MLP with $512$ hidden units, trained using AdamW [50] with a learning rate of $10^{-3}$ and batch size $64$. Training runs for $200$ epochs on fMRI and $2000$ epochs on EEG. RIEMCFM has a 6-layer MLP with $512$ hidden units trained with AdamW (learning rate of $10^{-4}$), as recommended in [12]. These four methods use the dopri5 method from the `torchdiffeq` [11] library for time integration.
All experiments were run within 10 hours on a single Nvidia A40 GPU with a 32-cores cpu.

---

[2] https://github.com/facebookresearch/riemannian-fm

# 5 Results

We report both quantitative results and a neurophysiological plausibility study. Quantitative comparisons are summarized in Table 1, with additional analysis of projection effects in Table 2. Neurophysiological relevance is assessed in Figure 2, which shows class-conditional fMRI connectomes via Fréchet means and topographic maps of EEG Common Spatial Patterns (CSP) filters. To complement Table 1, we provide a visual summary of our findings in Figure 3 ( Appendix I). Together, these results show that DIFFEOCFM produces realistic, class-conditional samples that preserve key features of brain connectivity. For a more detailed discussion of the baselines, please refer to Appendix H.

Table 1: **Performance of generative models on 3 fMRI and 2 EEG datasets, evaluated with quality and Classification Accuracy Score (CAS) metrics.** Quality metrics ($\alpha$-precision, $\beta$-recall, and $\alpha,\beta$-F1) assess alignment with the real distribution. CAS metrics (ROC-AUC and F1) evaluate downstream performance: a classifier is trained on generated data to predict *disease status* (fMRI) or *motor imagery class* (EEG), and tested on held-out real samples. *Real Data* rows use real samples to compare training and test distributions, serving as empirical upper bounds. The proposed method is denoted DIFFEOCFM . mean $\pm$ std are reported. **Bold** values denote the best method and any methods that are not significantly worse than it (one-sided paired Wilcoxon signed-rank test, $\alpha = 0.05$).

| Dataset | Method | Quality Metrics | | | CAS Metrics | | Time (s.) | |
| | | $\alpha$-Precision $\uparrow$ | $\beta$-Recall $\uparrow$ | $\alpha,\beta$-F1 $\uparrow$ | ROC-AUC $\uparrow$ | F1 $\uparrow$ | Training $\downarrow$ | Sampling $\downarrow$ |
|---|---|---|---|---|---|---|---|---|
| ABIDE | *Real Data* | $0.80_{\pm 0.08}$ | $0.79_{\pm 0.08}$ | $0.79_{\pm 0.03}$ | $0.67_{\pm 0.06}$ | $0.59_{\pm 0.07}$ | N/A | N/A |
| | DIFFEOGAUSS | $0.56_{\pm 0.06}$ | $0.29_{\pm 0.06}$ | $0.38_{\pm 0.06}$ | $\mathbf{0.66_{\pm 0.04}}$ | $\mathbf{0.53_{\pm 0.06}}$ | $\mathbf{0.07_{\pm 0.03}}$ | $\mathbf{0.06_{\pm 0.00}}$ |
| | TRIANGDDPM | $0.04_{\pm 0.02}$ | $0.00_{\pm 0.00}$ | $0.00_{\pm 0.00}$ | $0.53_{\pm 0.06}$ | $0.47_{\pm 0.12}$ | $33.80_{\pm 1.19}$ | $0.37_{\pm 0.05}$ |
| | TRIANGCFM | $0.04_{\pm 0.02}$ | $0.00_{\pm 0.00}$ | $0.00_{\pm 0.00}$ | $0.52_{\pm 0.05}$ | $0.40_{\pm 0.18}$ | $48.78_{\pm 1.27}$ | $0.79_{\pm 0.78}$ |
| | DIFFEOCFM | $\mathbf{0.77_{\pm 0.09}}$ | $\mathbf{0.48_{\pm 0.07}}$ | $\mathbf{0.59_{\pm 0.08}}$ | $0.64_{\pm 0.06}$ | $\mathbf{0.58_{\pm 0.07}}$ | $32.78_{\pm 0.96}$ | $0.40_{\pm 0.04}$ |
| ADNI | *Real Data* | $0.91_{\pm 0.03}$ | $0.85_{\pm 0.06}$ | $0.88_{\pm 0.03}$ | $0.62_{\pm 0.05}$ | $0.62_{\pm 0.05}$ | N/A | N/A |
| | DIFFEOGAUSS | $0.02_{\pm 0.01}$ | $0.51_{\pm 0.08}$ | $0.04_{\pm 0.02}$ | $\mathbf{0.60_{\pm 0.05}}$ | $0.29_{\pm 0.13}$ | $\mathbf{0.14_{\pm 0.01}}$ | $\mathbf{0.18_{\pm 0.01}}$ |
| | TRIANGDDPM | $0.02_{\pm 0.00}$ | $0.00_{\pm 0.00}$ | $0.00_{\pm 0.00}$ | $0.53_{\pm 0.05}$ | $0.18_{\pm 0.11}$ | $90.03_{\pm 2.01}$ | $0.62_{\pm 0.08}$ |
| | TRIANGCFM | $0.02_{\pm 0.00}$ | $0.00_{\pm 0.01}$ | $0.01_{\pm 0.01}$ | $0.56_{\pm 0.04}$ | $0.34_{\pm 0.13}$ | $87.37_{\pm 2.13}$ | $0.63_{\pm 0.09}$ |
| | DIFFEOCFM | $\mathbf{0.62_{\pm 0.11}}$ | $\mathbf{0.77_{\pm 0.02}}$ | $\mathbf{0.68_{\pm 0.06}}$ | $\mathbf{0.63_{\pm 0.04}}$ | $\mathbf{0.47_{\pm 0.10}}$ | $88.01_{\pm 2.90}$ | $0.69_{\pm 0.11}$ |
| OASIS-3 | *Real Data* | $0.88_{\pm 0.04}$ | $0.87_{\pm 0.03}$ | $0.88_{\pm 0.02}$ | $0.73_{\pm 0.05}$ | $0.63_{\pm 0.06}$ | N/A | N/A |
| | DIFFEOGAUSS | $0.51_{\pm 0.04}$ | $0.30_{\pm 0.04}$ | $0.38_{\pm 0.04}$ | $\mathbf{0.70_{\pm 0.05}}$ | $0.41_{\pm 0.07}$ | $\mathbf{0.10_{\pm 0.01}}$ | $\mathbf{0.13_{\pm 0.00}}$ |
| | TRIANGDDPM | $0.03_{\pm 0.01}$ | $0.00_{\pm 0.00}$ | $0.00_{\pm 0.00}$ | $0.54_{\pm 0.06}$ | $0.41_{\pm 0.14}$ | $70.39_{\pm 1.99}$ | $0.50_{\pm 0.06}$ |
| | TRIANGCFM | $0.06_{\pm 0.01}$ | $0.00_{\pm 0.00}$ | $0.00_{\pm 0.00}$ | $0.52_{\pm 0.06}$ | $0.41_{\pm 0.14}$ | $67.92_{\pm 2.31}$ | $0.52_{\pm 0.07}$ |
| | DIFFEOCFM | $\mathbf{0.60_{\pm 0.05}}$ | $\mathbf{0.35_{\pm 0.04}}$ | $\mathbf{0.44_{\pm 0.04}}$ | $\mathbf{0.67_{\pm 0.06}}$ | $\mathbf{0.53_{\pm 0.07}}$ | $67.83_{\pm 1.83}$ | $0.57_{\pm 0.05}$ |
| BNCI 2014-002 | *Real Data* | $0.70_{\pm 0.05}$ | $0.60_{\pm 0.05}$ | $0.64_{\pm 0.03}$ | $0.83_{\pm 0.01}$ | $0.75_{\pm 0.02}$ | N/A | N/A |
| | DIFFEOGAUSS | $0.46_{\pm 0.04}$ | $\mathbf{0.77_{\pm 0.02}}$ | $0.57_{\pm 0.03}$ | $0.80_{\pm 0.02}$ | $\mathbf{0.73_{\pm 0.02}}$ | $\mathbf{0.06_{\pm 0.01}}$ | $\mathbf{0.08_{\pm 0.01}}$ |
| | TRIANGDDPM | $0.43_{\pm 0.04}$ | $0.10_{\pm 0.02}$ | $0.16_{\pm 0.03}$ | $0.52_{\pm 0.03}$ | $0.20_{\pm 0.15}$ | $257.88_{\pm 0.14}$ | $0.30_{\pm 0.05}$ |
| | TRIANGCFM | $0.48_{\pm 0.05}$ | $0.22_{\pm 0.03}$ | $0.30_{\pm 0.03}$ | $0.55_{\pm 0.03}$ | $0.24_{\pm 0.11}$ | $251.78_{\pm 0.85}$ | $0.35_{\pm 0.09}$ |
| | RIEMCFM | $\mathbf{0.67_{\pm 0.07}}$ | $0.62_{\pm 0.06}$ | $\mathbf{0.63_{\pm 0.03}}$ | $\mathbf{0.81_{\pm 0.02}}$ | $0.72_{\pm 0.02}$ | $1983.58_{\pm 0.97}$ | $5.28_{\pm 0.58}$ |
| | DIFFEOCFM | $0.62_{\pm 0.04}$ | $0.63_{\pm 0.04}$ | $0.62_{\pm 0.02}$ | $\mathbf{0.81_{\pm 0.02}}$ | $\mathbf{0.74_{\pm 0.02}}$ | $253.04_{\pm 0.33}$ | $0.59_{\pm 0.08}$ |
| BNCI 2015-001 | *Real Data* | $0.89_{\pm 0.01}$ | $0.89_{\pm 0.01}$ | $0.89_{\pm 0.00}$ | $0.73_{\pm 0.01}$ | $0.67_{\pm 0.01}$ | N/A | N/A |
| | DIFFEOGAUSS | $0.84_{\pm 0.01}$ | $\mathbf{0.90_{\pm 0.01}}$ | $0.86_{\pm 0.01}$ | $\mathbf{0.73_{\pm 0.01}}$ | $\mathbf{0.68_{\pm 0.01}}$ | $\mathbf{0.07_{\pm 0.01}}$ | $0.16_{\pm 0.01}$ |
| | TRIANGDDPM | $0.73_{\pm 0.03}$ | $0.55_{\pm 0.03}$ | $0.63_{\pm 0.02}$ | $0.60_{\pm 0.02}$ | $0.59_{\pm 0.13}$ | $319.94_{\pm 5.88}$ | $0.37_{\pm 0.07}$ |
| | TRIANGCFM | $0.79_{\pm 0.03}$ | $0.73_{\pm 0.03}$ | $0.76_{\pm 0.02}$ | $0.61_{\pm 0.02}$ | $0.59_{\pm 0.07}$ | $313.22_{\pm 2.16}$ | $0.38_{\pm 0.08}$ |
| | RIEMCFM | $\mathbf{0.93_{\pm 0.04}}$ | $0.84_{\pm 0.05}$ | $0.88_{\pm 0.01}$ | $\mathbf{0.73_{\pm 0.01}}$ | $0.66_{\pm 0.02}$ | $2753.93_{\pm 0.31}$ | $11.02_{\pm 0.59}$ |
| | DIFFEOCFM | $\mathbf{0.92_{\pm 0.01}}$ | $0.86_{\pm 0.02}$ | $\mathbf{0.89_{\pm 0.01}}$ | $\mathbf{0.73_{\pm 0.01}}$ | $0.65_{\pm 0.01}$ | $319.83_{\pm 0.33}$ | $1.02_{\pm 0.08}$ |

## 5.1 Quantitative Study

**Quality Metrics**   As shown in Table 1, DIFFEOCFM consistently matches or outperforms all baseline generative models across datasets in terms of $\alpha,\beta$-F1, establishing itself as the most robust method for aligning with the true data distribution. It also achieves the highest $\alpha$-precision and $\beta$-recall across the three fMRI datasets. On EEG datasets, DIFFEOGAUSS achieves higher

$\beta$-recall but at the cost of much lower $\alpha$-precision, leading to weaker overall $\alpha, \beta$-F1 scores. In contrast, TRIANGDDPM and TRIANGCFM perform poorly on all quality metrics after projection, due to a substantial degradation in sample quality. As detailed in Table 2, projecting onto the manifold—$\text{Corr}_d$ for fMRI and $\mathbb{S}_d^{++}$ for EEG—introduces a significant performance drop. This is because TRIANGDDPM and TRIANGCFM generate matrices that visually resemble realistic connectivity patterns but contain negative eigenvalues, making them invalid. The projection step corrects these matrices but distorts their structure, leading to sharp decreases in $\alpha, \beta$-F1—up to $-0.74$ on ADNI and $-0.76$ on OASIS-3—rendering TRIANGCFM impractical for use. This degradation is also visible in Figure 5, where post-projection alter the structure of fMRI connectomes (see Appendix M). Compared to RIEMCFM, DIFFEOCFM delivers similar performance on the two EEG datasets, while training $8\times$ faster and sampling $10\times$ faster. Finally, on EEG datasets, DIFFEOCFM nearly matches the performance of REAL DATA in terms of $\alpha, \beta$-F1, suggesting high sample realism.

**CAS Metrics** For the CAS metrics, which evaluate downstream predictive performance using ROC-AUC and F1 scores, DIFFEOCFM consistently achieves strong results, often approaching the performance of REAL DATA. It obtains the highest ROC-AUC and F1 scores across all fMRI and EEG datasets. While DIFFEOGAUSS remains competitive in terms of ROC-AUC, DIFFEOCFM substantially outperforms it on F1 scores, with absolute gains of +0.05, +0.18, and +0.12 on ABIDE, ADNI, and OASIS-3, respectively. TRIANGDDPM and TRIANGCFM perform poorly across both CAS metrics on all datasets. This underperformance reflects the effect of the projection step, which alters generated matrices in ways detrimental to downstream classification. It is important to note that the classification pipeline requires inputs to be valid SPD matrices (i.e., elements of $\mathbb{S}_d^{++}$). As a result, CAS metrics cannot be computed for TRIANGDDPM and TRIANGCFM without projection. For this reason, their performance without projection is not reported in Table 2.

Table 2: **Impact of projection onto the manifolds $\mathbb{S}_d^{++}$ and $\text{Corr}_d$: performance difference $\delta = \text{TRIANGCFM} - \text{TRIANGCFM}$ without projections.** Negative $\Delta$ indicate degraded sample quality after projection onto the manifold. It enforces geometric constraints but severely reduces sample fidelity.

| Dataset | $\alpha$-precision | $\beta$-recall | $\alpha,\beta$-F1 |
|---|---|---|---|
| ABIDE | -0.34 | -0.69 | -0.50 |
| ADNI | -0.63 | -0.74 | -0.69 |
| OASIS-3 | -0.52 | -0.76 | -0.64 |
| BNCI2014-002 | +0.13 | -0.56 | -0.19 |
| BNCI2015-001 | +0.00 | -0.19 | -0.09 |

**Plotting of Generated Samples in Real Data Neighborhoods** To qualitatively assess fidelity, we show Figure 10–15 (fMRI) and Figure 16–19 (EEG) in Appendix P the generated samples closest to real ones in Frobenius distance. fMRI results are grouped by control (CN) and patient (non-CN); EEG by motor imagery class. DIFFEOCFM reliably populates the neighborhood of real data, capturing class-conditional structure. We also show TRIANGCFM samples *before projection*, which appear realistic but are not SPD.

## 5.2 Neurophysiological Plausibility Study

**fMRI Connectome Plotting** Figure 2a shows group-level functional connectomes from the ADNI dataset, computed as the Fréchet mean (4) of correlation matrices conditioned on disease status. For comparison, Figure 9 in Appendix O presents the corresponding group-level connectomes derived from the ABIDE and OASIS3 datasets. The Fréchet mean is defined with respect to $\phi_{\text{Corr}_d}$ (12), the diffeomorphism used for the generation. For each class (CN and non-CN), we compare real connectomes (from held-out test subjects) to those generated by DIFFEOCFM. In both cases, non-CN subjects exhibit reduced connectivity across hemispheres and between frontal and posterior regions—patterns commonly associated with mild cognitive impairment and Alzheimer's disease [19].

**EEG Topographic Map** Figure 2b presents the group-level topographies of the first CSP filter across all subjects, derived from EEG generated by DIFFEOCFM, alongside those from real EEG recordings in the BNCI2015-001 dataset. Subject-level topographies of the first CSP filter from the same dataset are further detailed in Figure 7 (Subjects 1–6) and Figure 8 (Subjects 7–12) in Appendix N, providing a detailed comparison across individuals. We visualize CSP spatial filters in the $\alpha$ (8–12 Hz) and $\beta$ (13–30 Hz) bands that distinguish imagined right-hand from feet movements. The filters trained on real and on generated data concentrate on the same contralateral sensorimotor regions, mirroring the close CAS scores between REAL DATA and DIFFEOCFM in Table 1. This confirms that the generative model preserves the physiologically relevant information.

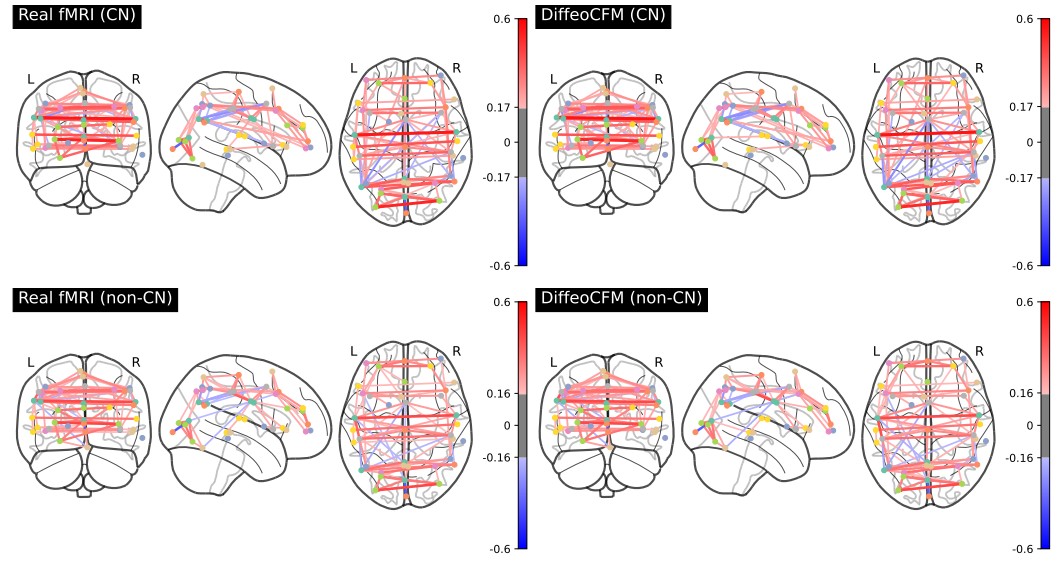

(a) **Class-conditional fMRI functional connectome plotting using the Fréchet mean (ADNI).** Each panel displays class-conditional fMRI functional connectomes using the Fréchet mean (4) of correlation matrices computed with respect to the generation diffeomorphism $\phi_{\mathrm{Corr}_d}$ (12). Left: real data from held-out test subjects; right: samples generated by DIFFEOCFM. The comparison illustrates both the fidelity of generated connectomes and the disease-specific connectivity structure preserved by the model.

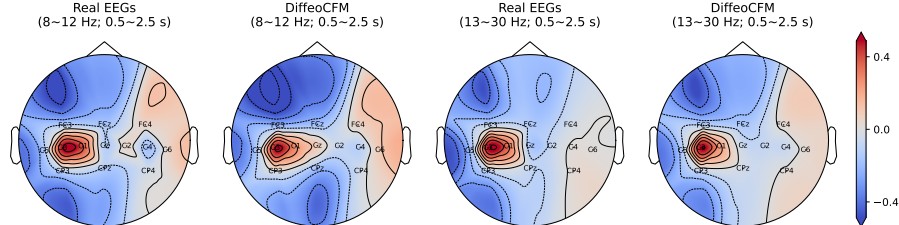

(b) **Group-level topographic map using the first CSP's spatial filter derived from real EEGs (BNCI2015-001) and generated data by DIFFEOCFM** Each map shows the first CSP's spatial filter across 12 subjects in the $\alpha$ ($8 - 12$ Hz) and $\beta$ ($13 - 30$ Hz) frequency bands during the first 2 seconds following stimulus onset. Filters from DIFFEOCFM at the group-level closely resemble real EEGs, preserving discriminative patterns of motor imagery classification.

Figure 2: Neurophysiological Plausibility Study of DIFFEOCFM.

## 6 Conclusions, Limitations, and Future Works

We introduced DIFFEOCFM, an efficient framework for generating brain connectivity matrices. By reformulating Riemannian flow matching through global diffeomorphisms, DIFFEOCFM enabled fast training and sampling while ensuring manifold-constrained outputs by construction. Applied to fMRI and EEG data, it outperformed existing baselines with neurophysiologically plausible samples.

Nonetheless, several limitations remain. First, DIFFEOCFM relies on a global diffeomorphism to Euclidean space, which exists for SPD and correlation matrices but not for all manifolds (e.g., Stiefel). Second, higher parcellation granularity makes the problem intrinsically hard: manifold dimension grows quadratically with region count, and the sample complexity grows exponentially [57]. Third, how alternative connectivity definitions (e.g., partial correlation or graphical-Lasso precision) affect generation quality remains an open question, since the choice of estimator directly defines the ground truth. Fourth, common evaluation metrics like $\alpha$-precision and $\beta$-recall are geometry-agnostic and may miss neurophysiological structure.

Despite these challenges, generative modeling remains a promising direction for neuroimaging and BCI research. For instance, sharing trained generative models, rather than raw data, can facilitate multi-site collaboration with privacy guarantees [10].

## Acknowledgment

This work was supported by grant ANR-22-PESN-0012 under the France 2030 program, managed by the Agence Nationale de la Recherche (ANR). CJ was supported by DATAIA Convergence Institute as part of the "Programme d'Investissement d'Avenir", (ANR-17-CONV-0003) operated by Inria. This work was performed using HPC resources from GENCI–IDRIS (Grant 2025-AD011016067).

Numerical computation was enabled by the scientific Python ecosystem: `Matplotlib` [33], `Scikit-learn` [61], `Numpy` [30], `Scipy` [72], `PyTorch` [59], `fMRIprep` [24], `Nilearn` [14], `joblib` [36], `PyRiemann` [5], `torchcfm` [69], `torchdiffeq` [11], `pandas` [58], and `moabb` [34].

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

# A  Pullback Manifolds with Euclidean Spaces

Section 2 briefly introduced pullback manifolds with Euclidean spaces. We here go more into details to then prove the different propositions of the paper. Let $\mathcal{M}$ be a smooth manifold, $E$ a Euclidean space, and $\phi : \mathcal{M} \to E$ a global diffeomorphism (a smooth bijection with a smooth inverse). The diffeomorphism $\phi$ induces a Riemannian metric on $\mathcal{M}$ by pulling back the Euclidean metric $g_E$ on $E$

$$(\phi^* g_E)_x(\xi, \eta) \triangleq g_E \left( \mathrm{D}\,\phi(x)[\xi], \mathrm{D}\,\phi(x)[\eta] \right), \quad \xi, \eta \in T_x\mathcal{M}. \tag{13}$$

This metric induces a Riemannian norm on the tangent space $T_x\mathcal{M}$ at each point $x \in \mathcal{M}$: $\|\xi\|_x = \sqrt{(\phi^* g_E)_x(\xi, \xi)}$. The pair $(\mathcal{M}, \phi^* g_E)$ is then called a *pullback manifold*, and $\phi^* g_E$ is the *pullback metric* of $g_E$ by $\phi$. The pullback manifold is geodesically complete and admits globally unique geodesics [68, Chap. 7]. Moreover, many Riemannian operations—distances, geodesics, exponential and logarithmic maps, parallel transport, Fréchet means—reduce to simple computations in $E$. Given two points $x, y \in \mathcal{M}$, the geodesic $\gamma : [0, 1] \to \mathcal{M}$ connecting $x$ to $y$ and its associated Riemannian distance are given by

$$\gamma(t) = \phi^{-1} \left( (1-t)\phi(x) + t\phi(y) \right) \quad \text{and} \quad d_\mathcal{M}(x, y) = \|\phi(x) - \phi(y)\|_E \tag{14}$$

i.e., the pullbacks of the Euclidean straight line and distance in $E$ joining $\phi(x)$ and $\phi(y)$. This structure also defines expressions for the exponential and logarithmic maps at any point $x \in \mathcal{M}$. The exponential map $\exp_x : T_x\mathcal{M} \to \mathcal{M}$ and the logarithmic map $\log_x : \mathcal{M} \to T_x\mathcal{M}$ are

$$\exp_x(\xi) = \phi^{-1} \left( \phi(x) + \mathrm{D}\,\phi(x)[\xi] \right) \quad \text{and} \quad \log_x(y) = \left( \mathrm{D}\,\phi(x) \right)^{-1} \left( \phi(y) - \phi(x) \right). \tag{15}$$

When $\mathrm{D}\,\phi(x)$ is not available in closed form, it can be computed via automatic differentiation using libraries such as JAX [8] or PyTorch [60]. Its inverse, $\left( \mathrm{D}\,\phi(x) \right)^{-1}$, can be obtained from the differential of $\phi^{-1}$ using the identity $\left( \mathrm{D}\,\phi(x) \right)^{-1} \circ \mathrm{D}\,\phi^{-1}(\phi(x)) = \mathrm{Id}$. The parallel transport of a tangent vector $\xi \in T_x\mathcal{M}$ along the geodesic $\gamma$ is the pullback of the Euclidean parallel transport from $\phi(x)$ to $\phi(y)$, which is given by

$$\mathrm{PT}_{x \to y}(\xi) = \left( \mathrm{D}\,\phi(y) \right)^{-1} \left( \mathrm{D}\,\phi(x)(\xi) \right). \tag{16}$$

The Fréchet mean [29] of a set of points $\{x^{(n)}\}_{n=1}^N \subset \mathcal{M}$ with respect to the Riemannian distance is

$$\bar{x} \triangleq \underset{x \in \mathcal{M}}{\arg\min} \sum_{n=1}^N d_\mathcal{M}(x, x^{(n)})^2 = \phi^{-1} \left( \frac{1}{N} \sum_{n=1}^N \phi(x^{(n)}) \right). \tag{17}$$

# B  Riemannian Conditional Flow Matching

This section provides a concise overview of Flow Matching, Conditional Flow Matching, and their extension to Riemannian manifolds, complementing Section 2. For clarity, we omit conditioning on variables $y$ such as disease status, age, or sex; the derivations extend naturally to the conditional case.

## B.1  Flow Matching (Intractable Objective)

Flow Matching (FM) [49, 47, 2, 48] aims to learn a time-dependent vector field $u_\theta(t, x)$ that transports samples from a simple source (prior) distribution $p(x)$ (e.g., a standard Gaussian $\mathcal{N}(0, I)$) at $t = 0$ to a target data distribution $q(x)$ at $t = 1$. This transformation is governed by an ordinary differential equation (ODE):

$$\dot{x}(t) = u(t, x(t)), \quad x(0) \sim p_0(x) \tag{18}$$

**Training**  Flow Matching trains a neural network $u_\theta(t, x)$ to approximate a true time-dependent vector field $u(t, x)$ that transports a source distribution $p(x)$ to a target $q(x)$ along an evolving density path $(p_t)_{t \in [0,1]}$. The objective is:

$$\mathcal{L}_{\mathrm{FM}}(\theta) \triangleq \mathbb{E}_{t \sim \mathcal{U}([0,1]),\, x \sim p_t(x)} \|u_\theta(t, x) - u(t, x)\|_2^2. \tag{19}$$

This formulation is generally intractable, as both $p_t(x)$ and $u(t, x)$ are unknown. Indeed, the time evolution of $p_t(x)$ is governed by the continuity equation:

$$\frac{\partial p_t(x)}{\partial t} + \mathrm{div}\left( p_t(x)\, u(t, x) \right) = 0, \tag{20}$$

which expresses conservation of mass along the flow.

**Sampling** To generate a sample $x_1$ from a learned model $u_\theta(t, x)$, one draws an initial sample $x_0 \sim p_0(x)$ and then solves the learned ODE from $t = 0$ to $t = 1$ using a numerical solver such as a Euler or a Runge-Kutta scheme:

$$\dot{x}(t) = u_\theta(t, x(t)) \quad \Rightarrow \quad x_1 = x_0 + \int_0^1 u_\theta(t, x(t)) dt. \tag{21}$$

### B.2 Conditional Flow Matching (CFM)

**Training** Conditional Flow Matching (CFM) makes the training of flow models tractable by defining explicit conditional paths and vector fields. We still consider a source distribution $p$ and a target data distribution $q$. A common choice for the path connecting $x_0$ to $x_1$ is a linear interpolation:

$$x_t(x_0, x_1) = (1 - t)x_0 + tx_1$$

The corresponding target conditional vector field is $u(t, x_t | x_0, x_1) = x_1 - x_0$. The CFM training loss for a neural network $u_\theta(t, x)$ is:

$$\mathcal{L}_{\text{CFM}}(\theta) \triangleq \mathbb{E}_{t \sim \mathcal{U}([0,1]), x_0 \sim p(x_0), x_1 \sim q(x_1)} \| u_\theta(t, (1-t)x_0 + tx_1) - (x_1 - x_0) \|_2^2. \tag{22}$$

Here, $p(x_0)$ is typically a simple noise distribution (e.g., $\mathcal{N}(0, I)$) and $q(x_1)$ is the empirical data distribution. Minimizing this CFM loss has been shown to be equivalent to minimizing the original intractable FM loss under certain conditions [48]. Other conditional path definitions can also be employed.

**Sampling** Sampling is performed by drawing $x_0 \sim p(x_0)$ and integrating the learned vector field $u_\theta(t, x(t))$ from $t = 0$ to $t = 1$:

$$\dot{x}(t) = u_\theta(t, x(t))$$

The solution $x(1)$ is then a sample from the learned approximation of $q(x_1)$.

### B.3 Riemannian Conditional Flow Matching (RCFM)

CFM was recently extended to Riemannian manifolds [12], providing a principled framework for learning time-dependent vector fields that transport samples between probability distributions defined on such spaces.

**Training** Given a manifold $\mathcal{M}$, a vector field $u_\theta^{\mathcal{M}} : [0, 1] \times \mathcal{M} \to T\mathcal{M}$ (where $T\mathcal{M}$ denotes the tangent bundle of $\mathcal{M}$) is trained to match the velocity of geodesics $\gamma(t)$ connecting samples from a source distribution $p(x_0)$ on $\mathcal{M}$ and a target distribution $q(x_1)$ on $\mathcal{M}$. The Riemannian CFM loss is defined as

$$\mathcal{L}(\theta) \triangleq \mathbb{E}_{t \sim \mathcal{U}([0,1]), x_0 \sim p(x_0), x_1 \sim q(x_1)} \left\| u_\theta^{\mathcal{M}}(t, \gamma(t)) - \dot{\gamma}(t) \right\|_{\gamma(t)}^2, \tag{23}$$

where $\gamma(t)$ is the geodesic connecting $x_0$ to $x_1$ such that $\gamma(0) = x_0$ and $\gamma(1) = x_1$, and $\dot{\gamma}(t)$ is its time derivative (velocity vector) which lies in $T_{\gamma(t)}\mathcal{M}$. It should be noted that this loss extends $\mathcal{L}_{\text{CFM}}$. Indeed, for $\mathcal{M} = \mathbb{R}^d$, then we get $\mathcal{L} = \mathcal{L}_{\text{CFM}}$.

**Sampling** Once trained, new samples on $\mathcal{M}$ are generated by solving the Riemannian ODE

$$\dot{x}(t) = u_\theta^{\mathcal{M}}(t, x(t)), \quad x(0) = x_0 \sim p(x_0) \tag{24}$$

and returning $x(1)$ as a sample from the learned approximation of $q(x_1)$. Numerical solution of this ODE typically involves manifold operations like the exponential map.

## C  Proof of Proposition 1: Riemannian CFM loss function on pullback manifolds

**Proposition 1** (Riemannian CFM loss function on pullback manifolds). *The Riemannian CFM loss* (5) *can be re-expressed in terms of the Euclidean vector field $u_\theta^E$* (7) *as*

$$\mathcal{L}(\theta) = \mathbb{E}_{t, y, z_0 | y, z_1 | y} \left\| u_\theta^E(t, (1-t)z_0 + tz_1, y) - (z_1 - z_0) \right\|_E^2,$$

*where $z_0 | y \sim \phi_\# p(\cdot | y)$ and $z_1 | y \sim \phi_\# q(\cdot | y)$.*

*Proof.* Fix a label $y$, draw $x_0 \sim p(\cdot \mid y)$ and $x_1 \sim q(\cdot \mid y)$, and set $z_0 = \phi(x_0)$, $z_1 = \phi(x_1)$. For any $t \in [0, 1]$ let $z_t \triangleq (1 - t)z_0 + tz_1$ and $\gamma(t) = \phi^{-1}(z_t)$.

**Geodesic velocity** Since $z_t$ is a straight line in $E$, the chain rule gives

$$\dot{\gamma}(t) = \mathrm{D}\,\phi^{-1}(z_t)(z_1 - z_0)\,.$$

**Pull-back of the vector field** By definition of $u_\theta^E$ in (7),

$$u_\theta^E(t, z_t, y) = \mathrm{D}\,\phi(\gamma(t))\left(u_\theta^{\mathcal{M}}(t, \gamma(t), y)\right).$$

**Norm preservation** Because the metric on $\mathcal{M}$ is the pull-back of $g_E$, we have $\|\xi\|_{\gamma(t)} = \|\,\mathrm{D}\,\phi(\gamma(t))\,(\xi)\|_E$ for every $\xi \in T_{\gamma(t)}\mathcal{M}$. Applying $\mathrm{D}\,\phi(\gamma(t))$ to the difference $u_\theta^{\mathcal{M}}(t, \gamma(t), y) - \dot{\gamma}(t)$ yields

$$\mathrm{D}\,\phi(\gamma(t))\left(u_\theta^{\mathcal{M}}(t, \gamma(t), y) - \dot{\gamma}(t)\right) = \mathrm{D}\,\phi(\gamma(t))\left(u_\theta^{\mathcal{M}}(t, \gamma(t), y)\right) - \mathrm{D}\,\phi(\gamma(t))(\dot{\gamma}(t))$$
$$= u_\theta^E(t, z_t, y) - \mathrm{D}\,\phi(\phi^{-1}(z_t))\left(\mathrm{D}\,\phi^{-1}(z_t)(z_1 - z_0)\right)\,.$$

Furthermore, since $(\phi \circ \phi^{-1})(z) = z$, we have $\mathrm{D}\,\phi(\phi^{-1}(z)) \circ \mathrm{D}\,\phi^{-1}(z) = \mathrm{Id}_E$. qHence, we get

$$\mathrm{D}\,\phi(\gamma(t))\left(u_\theta^{\mathcal{M}}(t, \gamma(t), y) - \dot{\gamma}(t)\right) = u_\theta^E(t, z_t, y) - (z_1 - z_0)\,.$$

Taking squared Euclidean norms on both sides gives

$$\left\|u_\theta^{\mathcal{M}}(t, \gamma(t), y) - \dot{\gamma}(t)\right\|_{\gamma(t)}^2 = \left\|u_\theta^E(t, z_t, y) - (z_1 - z_0)\right\|_E^2.$$

**Expectation** Finally, averaging over $t \sim \mathcal{U}[0, 1]$, $y \sim \pi_{\mathcal{Y}}$, $x_0 \sim p(\cdot \mid y)$, and $x_1 \sim q(\cdot \mid y)$ gives the desired equality of loss functions, proving Proposition 1. $\qquad\square$

## D Proof of Proposition 2: Equivalence of ODE Solutions

**Proposition 2** (Equivalence of ODE solutions). *The solution $x(t)$ to (6) satisfies*

$$x(t) = \phi^{-1}(z(t)) \quad \text{for all } t \in [0, 1]\,,$$

*where $z(t)$ is the solution of the ODE with $u_\theta^E$ (7) and initial condition $z_0 = \phi(x_0)$.*

*Proof.* Let $y \in \mathcal{Y}$, $x_0 \in \mathcal{M}$ and $z_0 \triangleq \phi(x_0)$. Let the Euclidean trajectory $z : [0, 1] \to E$ be a solution of

$$\dot{z}(t) = u_\theta^E(t, z(t), y)\,, \qquad z(0) = z_0.$$

Then, we define the candidate solution

$$x(t) \triangleq \phi^{-1}(z(t)) \qquad (0 \leq t \leq 1).$$

By the chain rule,

$$\dot{x}(t) = \mathrm{D}\,\phi^{-1}(z(t))(\dot{z}(t)) = \mathrm{D}\,\phi^{-1}(z(t))\left(u_\theta^E(t, z(t), y)\right).$$

For every $z = \phi(x)$, we have (7)

$$u_\theta^E(t, z, y) = \mathrm{D}\,\phi(x)\left(u_\theta^{\mathcal{M}}(t, x, y)\right).$$

Applying this with $x = x(t)$ and $z = z(t)$,

$$\dot{x}(t) = \mathrm{D}\,\phi^{-1}(\phi(x(t)))\left(\mathrm{D}\,\phi(x(t))\left(u_\theta^{\mathcal{M}}(t, x(t), y)\right)\right).$$

Using the identity, $\mathrm{D}\,\phi^{-1}(\phi(x)) \circ \mathrm{D}\,\phi(x) = \mathrm{Id}_{T_x\mathcal{M}}$, we get

$$\dot{x}(t) = u_\theta^{\mathcal{M}}(t, x(t), y)\,.$$

Finally, we check the initial condition,

$$x(0) = \phi^{-1}(z_0) = \phi^{-1}(\phi(x_0)) = x_0.$$

Consequently $x(t) = \phi^{-1}(z(t))$ is a solution on $\mathcal{M}$ of (6). $\qquad\square$

# E   Proof of Proposition 3: Equivalence of Runge–Kutta iterates

**Proposition 3** (Equivalence of Runge–Kutta iterates)**.** *Let $\{x_\ell\}$ be the iterates produced on $\mathcal{M}$ by an explicit Riemannian Runge–Kutta scheme applied to the ODE* (6)*. Then, the iterates are*
$$x_\ell = \phi^{-1}(z_\ell) \quad \text{for all } \ell \in \mathbb{N},$$
*where $\{z_\ell\}$ are the iterates obtained by applying the same scheme (same coefficients and step size) to the ODE with vector field $u_\theta^E$* (7) *and initial condition $z_0 = \phi(x_0)$.*

*Proof.* We prove this for the fourth-order Runge-Kutta method (RK4). Similar proofs can be done for the Euler (RK1) and midpoint (RK2) schemes. The ODEs are:
$$\dot{z}(t) = u_\theta^E(t, z(t), y) \quad \text{in Euclidean space } E$$
$$\dot{x}(t) = u_\theta^{\mathcal{M}}(t, x(t), y) \quad \text{on manifold } \mathcal{M}$$
The key relationships are:

- Vector field relationship: $u_\theta^E(t, z, y) = \mathrm{D}\,\phi(\phi^{-1}(z))(u_\theta^{\mathcal{M}}(t, \phi^{-1}(z), y))$.

- Exponential map on $\mathcal{M}$: $\exp_x(\xi) = \phi^{-1}(\phi(x) + \mathrm{D}\,\phi(x)(\xi))$ for $x \in \mathcal{M}, \xi \in T_x\mathcal{M}$.

- Parallel transport on $\mathcal{M}$ from $y \in \mathcal{M}$ to $x \in \mathcal{M}$: $\mathrm{PT}_{y \to x}(\eta) = (\mathrm{D}\,\phi(x))^{-1}(\mathrm{D}\,\phi(y)(\eta))$ for $\eta \in T_y\mathcal{M}$.

We use induction. Assume $x_\ell = \phi^{-1}(z_\ell)$ for some $\ell$. We show $x_{\ell+1} = \phi^{-1}(z_{\ell+1})$.

**RK4 scheme in $E$:** Given $z_\ell$ at time $t_\ell$, and step size $h$:
$$k_1^E = u_\theta^E(t_\ell, z_\ell, y)$$
$$k_2^E = u_\theta^E(t_\ell + \frac{h}{2}, z_\ell + \frac{h}{2}k_1^E, y)$$
$$k_3^E = u_\theta^E(t_\ell + \frac{h}{2}, z_\ell + \frac{h}{2}k_2^E, y)$$
$$k_4^E = u_\theta^E(t_\ell + h, z_\ell + hk_3^E, y)$$
$$z_{\ell+1} = z_\ell + \frac{h}{6}(k_1^E + 2k_2^E + 2k_3^E + k_4^E)$$

**RK4 scheme on $\mathcal{M}$:** A Riemannian RK4 method involves evaluating $u_\theta^{\mathcal{M}}$ at intermediate points, transporting the resulting tangent vectors to $T_{x_\ell}\mathcal{M}$, combining them, and then using the exponential map. By hypothesis, $\phi(x_\ell) = z_\ell$.

1. **First stage ($k_1^{\mathcal{M}}$):** We have
$$k_1^{\mathcal{M}} = u_\theta^{\mathcal{M}}(t_\ell, x_\ell, y) \in T_{x_\ell}\mathcal{M}.$$
   So,
$$k_1^{\mathcal{M}} = (\mathrm{D}\,\phi(x_\ell))^{-1}(u_\theta^E(t_\ell, \phi(x_\ell), y)) = (\mathrm{D}\,\phi(x_\ell))^{-1}(k_1^E).$$
   The intermediate point is
$$x_A = \exp_{x_\ell}\left(\frac{h}{2}k_1^{\mathcal{M}}\right).$$
   Substituting the expressions, we get
$$x_A = \phi^{-1}\left(\phi(x_\ell) + \mathrm{D}\,\phi(x_\ell)\left(\frac{h}{2}k_1^{\mathcal{M}}\right)\right)$$
$$= \phi^{-1}\left(z_\ell + \mathrm{D}\,\phi(x_\ell)\left(\frac{h}{2}(\mathrm{D}\,\phi(x_\ell))^{-1}(k_1^E)\right)\right)$$
$$= \phi^{-1}\left(z_\ell + \frac{h}{2}k_1^E\right).$$
   Thus,
$$\phi(x_A) = z_\ell + \frac{h}{2}k_1^E.$$

2. **Second stage ($k_2^{\mathcal{M}}$):**
$$k_2^{\mathcal{M}} = u_\theta^{\mathcal{M}}(t_\ell + \frac{h}{2}, x_A, y) \in T_{x_A}\mathcal{M}$$

$$k_2^{\mathcal{M}} = (\mathrm{D}\,\phi(x_A))^{-1}(u_\theta^E(t_\ell + \frac{h}{2}, \phi(x_A), y)) = (\mathrm{D}\,\phi(x_A))^{-1}(k_2^E)$$

Transport $k_2^{\mathcal{M}}$ to $T_{x_\ell}\mathcal{M}$:
$$k_{2,\mathrm{tp}}^{\mathcal{M}} = \mathrm{PT}_{x_A \to x_\ell}(k_2^{\mathcal{M}}) = (\mathrm{D}\,\phi(x_\ell))^{-1}(\mathrm{D}\,\phi(x_A)(k_2^{\mathcal{M}})) = (\mathrm{D}\,\phi(x_\ell))^{-1}(k_2^E)$$

The intermediate point
$$x_B = \exp_{x_\ell}(\frac{h}{2}k_{2,\mathrm{tp}}^{\mathcal{M}})$$

$$x_B = \phi^{-1}\left(\phi(x_\ell) + \mathrm{D}\,\phi(x_\ell)(\frac{h}{2}k_{2,\mathrm{tp}}^{\mathcal{M}})\right)$$

$$= \phi^{-1}\left(z_\ell + \mathrm{D}\,\phi(x_\ell)(\frac{h}{2}(\mathrm{D}\,\phi(x_\ell))^{-1}(k_2^E))\right)$$

$$= \phi^{-1}\left(z_\ell + \frac{h}{2}k_2^E\right)$$

Thus,
$$\phi(x_B) = z_\ell + \frac{h}{2}k_2^E$$

3. **Third stage ($k_3^{\mathcal{M}}$):**
$$k_3^{\mathcal{M}} = u_\theta^{\mathcal{M}}(t_\ell + \frac{h}{2}, x_B, y) \in T_{x_B}\mathcal{M}$$

$$k_3^{\mathcal{M}} = (\mathrm{D}\,\phi(x_B))^{-1}(u_\theta^E(t_\ell + \frac{h}{2}, \phi(x_B), y)) = (\mathrm{D}\,\phi(x_B))^{-1}(k_3^E)$$

Transport $k_3^{\mathcal{M}}$ to $T_{x_\ell}\mathcal{M}$:
$$k_{3,\mathrm{tp}}^{\mathcal{M}} = \mathrm{PT}_{x_B \to x_\ell}(k_3^{\mathcal{M}}) = (\mathrm{D}\,\phi(x_\ell))^{-1}(\mathrm{D}\,\phi(x_B)(k_3^{\mathcal{M}})) = (\mathrm{D}\,\phi(x_\ell))^{-1}(k_3^E)$$

The intermediate point
$$x_C = \exp_{x_\ell}(hk_{3,\mathrm{tp}}^{\mathcal{M}})$$

$$x_C = \phi^{-1}\left(\phi(x_\ell) + \mathrm{D}\,\phi(x_\ell)(hk_{3,\mathrm{tp}}^{\mathcal{M}})\right)$$

$$= \phi^{-1}\left(z_\ell + \mathrm{D}\,\phi(x_\ell)(h(\mathrm{D}\,\phi(x_\ell))^{-1}(k_3^E))\right)$$

$$= \phi^{-1}\left(z_\ell + hk_3^E\right)$$

Thus,
$$\phi(x_C) = z_\ell + hk_3^E$$

4. **Fourth stage ($k_4^{\mathcal{M}}$):**
$$k_4^{\mathcal{M}} = u_\theta^{\mathcal{M}}(t_\ell + h, x_C, y) \in T_{x_C}\mathcal{M}$$
$$k_4^{\mathcal{M}} = (\mathrm{D}\,\phi(x_C))^{-1}(u_\theta^E(t_\ell + h, \phi(x_C), y)) = (\mathrm{D}\,\phi(x_C))^{-1}(k_4^E)$$
Transport $k_4^{\mathcal{M}}$ to $T_{x_\ell}\mathcal{M}$:
$$k_{4,\mathrm{tp}}^{\mathcal{M}} = \mathrm{PT}_{x_C \to x_\ell}(k_4^{\mathcal{M}}) = (\mathrm{D}\,\phi(x_\ell))^{-1}(\mathrm{D}\,\phi(x_C)(k_4^{\mathcal{M}})) = (\mathrm{D}\,\phi(x_\ell))^{-1}(k_4^E)$$

5. **Final update on $\mathcal{M}$:** The combined tangent vector in $T_{x_\ell}\mathcal{M}$ is:

$$\Delta x_{\mathrm{tangent}} = \frac{h}{6}(k_1^{\mathcal{M}} + 2k_{2,\mathrm{tp}}^{\mathcal{M}} + 2k_{3,\mathrm{tp}}^{\mathcal{M}} + k_{4,\mathrm{tp}}^{\mathcal{M}})$$

$$= \frac{h}{6}\left((\mathrm{D}\,\phi(x_\ell))^{-1}(k_1^E) + 2(\mathrm{D}\,\phi(x_\ell))^{-1}(k_2^E) + 2(\mathrm{D}\,\phi(x_\ell))^{-1}(k_3^E) + (\mathrm{D}\,\phi(x_\ell))^{-1}(k_4^E)\right)$$

$$= (\mathrm{D}\,\phi(x_\ell))^{-1}\left(\frac{h}{6}(k_1^E + 2k_2^E + 2k_3^E + k_4^E)\right)$$

Then, $x_{\ell+1} = \exp_{x_\ell}(\Delta x_{\text{tangent}})$.

$$
\begin{aligned}
x_{\ell+1} &= \phi^{-1}\left(\phi(x_\ell) + \mathrm{D}\,\phi(x_\ell)(\Delta x_{\text{tangent}})\right) \\
&= \phi^{-1}\left(\phi(x_\ell) + \mathrm{D}\,\phi(x_\ell)\left((\mathrm{D}\,\phi(x_\ell))^{-1}\left(\frac{h}{6}(k_1^E + 2k_2^E + 2k_3^E + k_4^E)\right)\right)\right) \\
&= \phi^{-1}\left(z_\ell + \frac{h}{6}(k_1^E + 2k_2^E + 2k_3^E + k_4^E)\right) \\
&= \phi^{-1}(z_{\ell+1})
\end{aligned}
$$

The base case $x_0 = \phi^{-1}(z_0)$ is given by the problem setup ($z_0 = \phi(x_0)$). Therefore, by induction, $x_\ell = \phi^{-1}(z_\ell)$ for all $\ell \in \mathbb{N}$ when the RK4 scheme is applied as described. $\qquad\square$

## F The affine-invariant metric for correlation matrices

The set of correlation matrices $\mathrm{Corr}_d$ can be viewed as a quotient manifold of symmetric positive definite matrices $\mathbb{S}_d^{++}$ by the action of positive diagonal matrices $\mathbb{D}_d^{++}$ [16, 68]:

$$
\mathrm{Corr}_d = \mathbb{S}_d^{++}/\mathbb{D}_d^{++}, \tag{25}
$$

where two matrices $\mathbf{\Sigma}, \mathbf{\Sigma}' \in \mathbb{S}_d^{++}$ are equivalent if there exists $\mathbf{D} \in \mathbb{D}_d^{++}$ such that $\mathbf{\Sigma}' = \mathbf{D}\mathbf{\Sigma}\mathbf{D}$.

This quotient structure naturally induces a Riemannian metric on $\mathrm{Corr}_d$ from the affine-invariant metric on $\mathbb{S}_d^{++}$, defined by

$$
\langle \boldsymbol{\xi}, \boldsymbol{\eta}\rangle_{\mathbf{\Sigma}}^{\mathbb{S}_d^{++}} = \mathrm{tr}(\mathbf{\Sigma}^{-1}\boldsymbol{\xi}\mathbf{\Sigma}^{-1}\boldsymbol{\eta}). \tag{26}
$$

The canonical projection (submersion) $\pi : \mathbb{S}_d^{++} \to \mathrm{Corr}_d$ is given by diagonal normalization:

$$
\pi(\mathbf{\Sigma}) = \mathrm{diag}(\mathbf{\Sigma})^{-1/2}\mathbf{\Sigma}\,\mathrm{diag}(\mathbf{\Sigma})^{-1/2}. \tag{27}
$$

The resulting geometry on $\mathrm{Corr}_d$ requires solving an optimization problem to compute the Riemannian logarithm:

$$
\log_{\mathbf{A}}^{\mathrm{Corr}_d}(\mathbf{B}) = \mathrm{D}\,\pi(\mathbf{A})\left(\log_{\mathbf{A}}^{\mathbb{S}_d^{++}}(\mathbf{D}\mathbf{B}\mathbf{D})\right), \tag{28}
$$

where

$$
\mathbf{D} = \arg\min_{\mathbf{D}\in\mathbb{D}_d^{++}} d_{\mathbb{S}_d^{++}}(\mathbf{A}, \mathbf{D}\mathbf{B}\mathbf{D}). \tag{29}
$$

Since no closed-form solution for $\mathbf{D}$ is known, the logarithmic map on $(\mathrm{Corr}_d, \langle\cdot, \cdot\rangle^{\mathrm{Corr}_d})$ cannot be expressed analytically (contrary to $\log_{\mathbf{A}}^{\mathbb{S}_d^{++}}$).

# G   Datasets and Preprocessing

**Resting-state fMRI datasets and preprocessing**   We used three publicly available resting-state fMRI datasets—ABIDE [53], ADNI [73], and OASIS-3 [42]—spanning a wide age range and neurological conditions. ABIDE includes 900 subjects (one scan each; mean age 17), both neurotypical and autistic, from 19 international sites. ADNI comprises 1900 scans from 900 older adults (mean age 74), covering normal ageing, mild cognitive impairment, and Alzheimer's disease. OASIS-3 includes 1000 participants and 1800 longitudinal sessions collected over 10 years (mean age 71), targeting healthy and pathological ageing.

Preprocessing was performed with `fMRIPrep` [24], which applies bias-field correction, skull stripping, tissue segmentation, nonlinear normalization to MNI space, motion correction, and confound estimation. Regional time series were extracted with `Nilearn` [14] using the MSDL atlas [71], followed by nuisance regression (including motion, CSF/WM signals, and global signal). Time series were $z$-scored and screened for minimum length, numerical anomalies (e.g., zeros or extreme values), and the conditioning of their covariance matrices.

**EEG datasets**   Sessions with missing or non-standard protocol (e.g., 2C in BNCI2015-001) were discarded. We further filtered time series whose covariance matrices have high values (max entry $> 10^4$) or statistical outliers based on Mahalanobis distance from the group mean. This retained the top 90% most consistent trials for downstream modeling.

**BNCI2014-002**: The BNCI2014-002[3] dataset, provided through the BNCI Horizon 2020 initiative, includes recordings from 13 subjects engaged in motor imagery tasks. Participants were instructed to imagine movements of either their right hand or both feet for a sustained duration of 5 seconds, guided by visual cues. EEG signals were captured using an amplifier and active Ag/AgCl electrodes at a sampling rate of 512 Hz, with a total of 15 electrode channel applied to each subject. The experimental protocol comprised eight sessions per participant, each including 80 trials for hand and foot imagery, summing to 160 trials. EEG epochs were extracted from 3.0 to 8.0 seconds relative to cue onset, aligning with the motor imagery window.

**BNCI2015-001**: The BNCI2015-001[4] dataset, released as part of the BNCI Horizon 2020 project, contains EEG recordings from 12 individuals who performed motor imagery tasks involving either the right hand or both feet. EEG data were acquired at a 512 Hz sampling rate, preprocessed with a 0.5–100 Hz bandpass filter and a 50 Hz notch filter. Each trial spanned 5 seconds, beginning at 3.0 seconds after cue onset and ending at 8.0 seconds, corresponding to the motor imagery phase. The dataset provides EEG data from 13 electrode channels arranged in the following order: FC3, FCz, FC4, C5, C3, C1, Cz, C2, C4, C6, CP3, CPz, and CP4. For most participants (Subjects 1–8), recordings were conducted over two sessions on consecutive days. In contrast, Subjects 9–12 completed three sessions. Each session consisted of 100 trials per class, resulting in 200 trials per session per subject.

---

[3]https://neurotechx.github.io/moabb/generated/moabb.datasets.BNCI2014_002.html
[4]https://neurotechx.github.io/moabb/generated/moabb.datasets.BNCI2015_001.html

## H  Comparison of Baseline Methods: Assumptions, Strengths, and Limitations

In this section, we summarize the key assumptions, strengths, and limitations of the baseline methods discussed throughout the paper, based on our experimental findings.

- DIFFEOGAUSS: This is a simple yet practical generative approach that operates directly on SPD or correlation manifolds. While it yields competitive results in terms of the CAS metrics, it generally underperforms compared to deep generative models in terms of sample quality, see Table 1.

- TRIANGDDPM or TRIANGCFM: They are deep generative models that achieve strong performance across both CAS and quality-based metrics. However, their outputs are limited to triangular entities, requiring a post-processing projection step to obtain valid positive definite matrices. Unfortunately, this step alters the eigenvalue spectrum significantly, which degrades the quality and structure of the generated matrices (see Table 2 and Appendix M).

- SPD-DDPM: This work extends DDPM to the space of SPD manifolds under the affine-invariant Riemannian metric. However, in our experiments, we found that its implementation repeatedly relies on sampling from Gaussian distributions on SPD manifolds, as well as backpropagating through matrix operations in SPDNet. Both components are computationally intensive in theory and practice, making the method prohibitively slow. In fact, for a number of training samples above 64, we were unable to obtain results within a reasonable time frame (see Figure 4). However, in the fMRI experiments, the data have at least 1000 samples.

- RIEMANNIAN CFM: This is a foundational contribution in the family of flow matching methods, providing a general theoretical framework for generative modeling on Riemannian manifolds. However, its construction is geometry-specific and requires a dedicated implementation for each manifold. Furthermore, on SPD matrices, we observe that it is roughly $8\times$ slower to train and $10\times$ slower to sample from than DIFFEOCFM (see Table 1), due to the repeated computation of affine-invariant geodesics, their derivatives and Riemannian norms.

- DIFFEOCFM: The proposed approach introduces a novel Riemannian CFM framework using the pullback geometry, enabling direct data generation on SPD and correlation manifolds, particularly helpful in neuroscience and neuroengineering tasks. It trains and sampels efficiently (see Figure 4 and Table 1), achieves strong performance on both quality and CAS metrics (see Table 1), and consistently demonstrates neurophysiological plausibility across two distinct evaluation protocols in both fMRI and EEG (see Section 5.2).

# I Summary Figure

While Table 1 provides a comprehensive numerical evaluation of all methods, a visual representation can offer clearer insights into the practical trade-offs between model performance and computational efficiency. To this end, Figure 3 plots a unified performance metric,the Average F1 Score, against both training and sampling times. This allows for a direct comparison of the performance-cost profile of each method for both fMRI and EEG data.

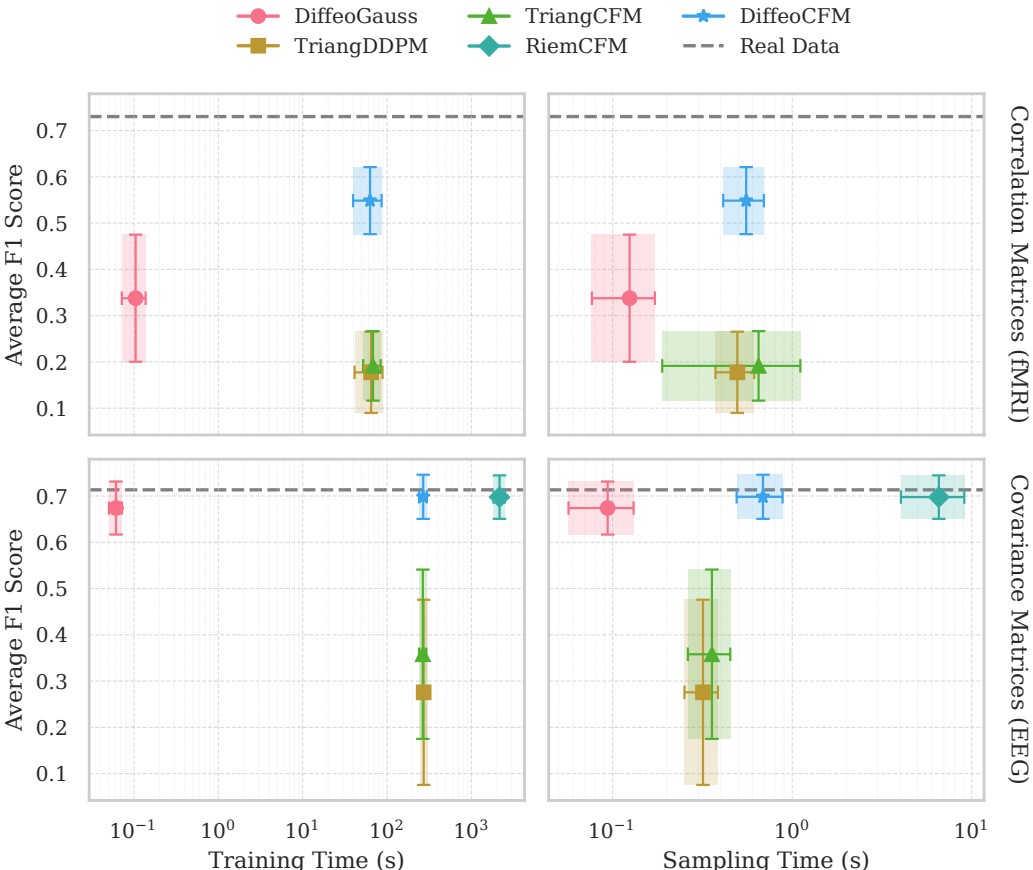

Figure 3: **Trade-off between generative performance and computational cost for fMRI (top) and EEG (bottom) data.** The figure plots the Average F1 Score against Training Time (left) and Sampling Time (right). The Average F1 Score is the mean of the quality metric ($\alpha, \beta$-F1) and the CAS F1-score from Table 1. Each point marks the mean performance across all splits and datasets for a given modality, with error bars and shaded regions indicating the standard deviation. The dashed gray line represents the *Real Data* baseline, which serves as an empirical upper bound. Time is shown on a logarithmic scale.

## J  Scalability and Computational cost of SPD–DDPM [45] vs DIFFEOCFM

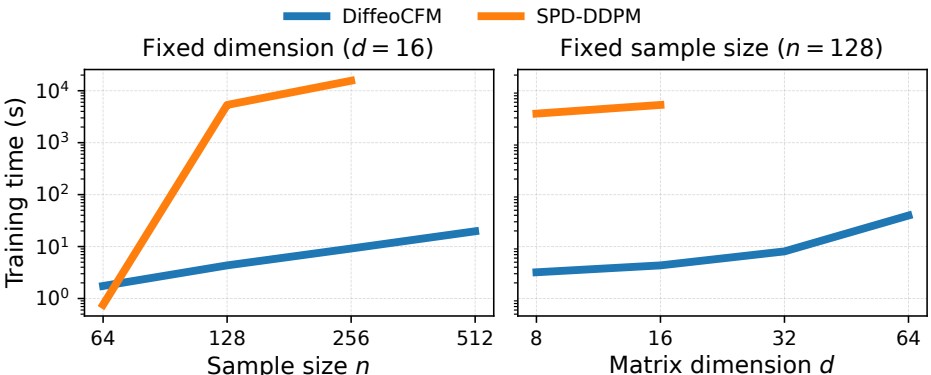

Figure 4: **Training efficiency on $\mathbb{S}_d^{++}$ (simulated data).** Log-scale training time over 1000 epochs for generating SPD matrices. We vary either the number of samples $n$ or the matrix dimension $d$, keeping the other fixed. DIFFEOCFM leverages the diffeomorphism $\phi_{\mathbb{S}_d^{++}}$ (9) and is at least $1000\times$ faster than SPD-DDPM [45]. Importantly, DIFFEOCFM *can also generate natively correlation matrices (Corr$_d$), and it performs with same training time as on* $\mathbb{S}_d^{++}$.

## K  Ablation Study

We conduct an ablation study to analyze the sensitivity of our proposed DIFFEOCFM model to its neural network architecture. The experiments are performed on the ABIDE dataset using the Cholesky diffeomorphism. We investigate two key architectural hyperparameters: the width (size of hidden layers) and the depth (number of hidden layers).

First, we fix the network depth to a single hidden layer and vary its width from 8 to 1024, as detailed in Table 3. Second, we fix the layer width to 512 neurons and vary the depth from one to four hidden layers, with results presented in Table 4. For both experiments, performance is evaluated using the $\alpha, \beta$-F1 score, which is the harmonic mean of $\alpha$-precision and $\beta$-recall. We also report the average training and sampling times over 10 splits.

The results from Table 3 indicate that performance generally improves with network width, peaking at 256 and 512 hidden units. A wider network of 1024 units shows a drop in performance. Regarding network depth, Table 4 clearly shows that a single hidden layer achieves the best results. Deeper models exhibit a consistent decline in performance. Based on these findings, a single hidden layer with a width of 512 was chosen for our main experiments, as it offers the best performance without unnecessary complexity.

Table 3: Ablation study on the **width** of the hidden layer for the DIFFEOCFM model on the ABIDE dataset. The network depth is fixed to one hidden layer. We report the mean and standard deviation of the $\alpha, \beta$-F1 score, training time, and sampling time across 10 splits. **Best** values are in bold.

| Hidden dim. | $\alpha, \beta$-F1 | Training time (s.) | Sampling time (s.) |
|---|---|---|---|
| 8 | $0.47 \pm 0.06$ | $29.07 \pm 1.07$ | $0.35 \pm 0.01$ |
| 16 | $0.47 \pm 0.09$ | $\mathbf{27.46 \pm 1.24}$ | $0.29 \pm 0.03$ |
| 32 | $0.54 \pm 0.05$ | $28.36 \pm 1.26$ | $\mathbf{0.26 \pm 0.02}$ |
| 64 | $0.57 \pm 0.07$ | $29.69 \pm 1.31$ | $0.30 \pm 0.03$ |
| 128 | $0.60 \pm 0.05$ | $31.01 \pm 1.32$ | $0.35 \pm 0.04$ |
| 256 | $\mathbf{0.63 \pm 0.07}$ | $31.39 \pm 0.96$ | $0.32 \pm 0.04$ |
| 512 | $\mathbf{0.63 \pm 0.06}$ | $31.36 \pm 1.14$ | $0.31 \pm 0.02$ |
| 1024 | $0.49 \pm 0.12$ | $32.34 \pm 1.17$ | $0.36 \pm 0.05$ |

Table 4: Ablation study on the **depth** of the neural network for the DIFFEOCFM model on the ABIDE dataset. The width of all hidden layers is fixed to 512. We report the mean and standard deviation of the $\alpha, \beta$-F1 score, training time, and sampling time across 10 splits. **Best** values are in bold.

| Depth | $\alpha, \beta$-F1 | Training time (s.) | Sampling time (s.) |
|---|---|---|---|
| 1 | **0.62 ± 0.05** | 33.06 ± 1.05 | 0.38 ± 0.03 |
| 2 | 0.52 ± 0.09 | **32.93 ± 1.00** | **0.36 ± 0.02** |
| 3 | 0.46 ± 0.05 | 34.36 ± 1.00 | 0.46 ± 0.04 |
| 4 | 0.50 ± 0.08 | 35.68 ± 1.01 | 0.48 ± 0.08 |

# L   Constraint Satisfaction ($\mathbb{S}_d^{++}$ and $\mathbf{Corr}_d$)

Table 5 reports the fraction of generated samples that satisfy structural matrix constraints across datasets. Both DIFFEOGAUSS and DIFFEOCFM systematically generate matrices that are symmetric, positive definite, and (for correlation matrices) have unit diagonal, achieving a perfect $1.00$ score on all datasets. In contrast, TRIANGCFM without projection produces matrices that are symmetric and have unit diagonal by construction but fail to ensure positive definiteness. This issue is particularly evident in fMRI datasets, where none of the generated samples satisfy the SPD constraint, and persists to a lesser extent on EEG datasets (e.g., $29\%$ validity on BNCI2015-001). These results highlight the importance of geometry-aware methods like DIFFEOCFM, which inherently respect the manifold structure of connectivity matrices without relying on post hoc projections.

Table 5: **Fraction of generated samples satisfying matrix constraints.** We report the fraction of samples satisfying symmetry, positive definiteness, and unit diagonal constraints across datasets.

| Dataset | Method | Sym. | Pos. def. | Unit diag. |
|---|---|---|---|---|
| ABIDE | DIFFEOGAUSS | **1.00 ± 0.00** | **1.00 ± 0.00** | **1.00 ± 0.00** |
| | TRIANGCFM (no proj.) | **1.00 ± 0.00** | 0.00 ± 0.00 | **1.00 ± 0.00** |
| | DiffeoCFM | **1.00 ± 0.00** | **1.00 ± 0.00** | **1.00 ± 0.00** |
| ADNI | DIFFEOGAUSS | **1.00 ± 0.00** | **1.00 ± 0.00** | **1.00 ± 0.00** |
| | TRIANGCFM (no proj.) | **1.00 ± 0.00** | 0.00 ± 0.00 | **1.00 ± 0.00** |
| | DiffeoCFM | **1.00 ± 0.00** | **1.00 ± 0.00** | **1.00 ± 0.00** |
| OASIS-3 | DIFFEOGAUSS | **1.00 ± 0.00** | **1.00 ± 0.00** | **1.00 ± 0.00** |
| | TRIANGCFM (no proj.) | **1.00 ± 0.00** | 0.00 ± 0.00 | **1.00 ± 0.00** |
| | DiffeoCFM | **1.00 ± 0.00** | **1.00 ± 0.00** | **1.00 ± 0.00** |
| BNCI 2014-002 | DIFFEOGAUSS | **1.00 ± 0.00** | **1.00 ± 0.00** | |
| | TRIANGCFM (no proj.) | **1.00 ± 0.00** | 0.00 ± 0.00 | |
| | DiffeoCFM | **1.00 ± 0.00** | **1.00 ± 0.00** | |
| BNCI 2015-001 | DIFFEOGAUSS | **1.00 ± 0.00** | **1.00 ± 0.00** | |
| | TRIANGCFM (no proj.) | **1.00 ± 0.00** | 0.29 ± 0.02 | |
| | DiffeoCFM | **1.00 ± 0.00** | **1.00 ± 0.00** | |

# M   Projection

Since TRIANGDDPM and TRIANGCFM operate in Euclidean spaces, the generated matrices are not guaranteed to be positive definite. To address this, we apply a projection to ensure all eigenvalues are at least $\epsilon > 0$. Let $\lambda_{\min}(\mathbf{\Sigma})$ denote the smallest eigenvalue of $\mathbf{\Sigma}$. If $\lambda_{\min}(\mathbf{\Sigma}) < \epsilon$, the matrix is projected using an affine transformation:

$$\mathbf{\Sigma}_{\text{proj}} = (1 - \alpha)\mathbf{\Sigma} + \alpha\mathbf{I} \tag{30}$$

where $\mathbf{I}$ is the identity matrix. The parameter $\alpha$ is chosen such that the smallest eigenvalue of $\mathbf{\Sigma}_{\text{proj}}$ becomes exactly $\epsilon$. Given that the eigenvalues of $\mathbf{\Sigma}_{\text{proj}}$ are $\lambda_i(\mathbf{\Sigma}_{\text{proj}}) = (1 - \alpha)\lambda_i(\mathbf{\Sigma}) + \alpha$, setting

the new minimum eigenvalue to $\epsilon$ yields $\epsilon = (1 - \alpha)\lambda_{\min}(\mathbf{\Sigma}) + \alpha$. Solving for $\alpha$, we get:

$$\alpha = \frac{\epsilon - \lambda_{\min}(\mathbf{\Sigma})}{1 - \lambda_{\min}(\mathbf{\Sigma})} . \tag{31}$$

This formulation guarantees that if $\lambda_{\min}(\mathbf{\Sigma}) < \epsilon$, the new minimum eigenvalue is $\epsilon$. Typically, $\epsilon$ is a small positive value (e.g., $10^{-8}$), ensuring $\epsilon \leq 1$. In this regime, and given $\lambda_{\min}(\mathbf{\Sigma}) < \epsilon$, it follows that $0 < \alpha \leq 1$, making the projection an interpolation. An important property of this projection is that if the original matrix $\mathbf{\Sigma}$ has ones on its diagonal ($\mathrm{diag}(\mathbf{\Sigma}) = \mathbf{1}$), this property is preserved in $\mathbf{\Sigma}_{\mathrm{proj}}$, as $\mathrm{diag}(\mathbf{\Sigma}_{\mathrm{proj}}) = (1 - \alpha)\mathbf{1} + \alpha\mathbf{1} = \mathbf{1}$.

We randomly select 5 sample images from each generated dataset produced by the TRIANGCFM method across the 5 datasets. These images are not positive definite matrices, as they contain negative eigenvalues. We apply a projection algorithm to convert these matrices into valid correlation matrices, and both the original and projected versions are shown in Figure 5 and Figure 6. In each image, we annotate the maximum and minimum eigenvalues of the corresponding matrix. As can be seen, after the projection algorithm, all negative eigenvalues are eliminated, and the matrices become valid positive definite matrices. *Unfortunately, although this step is necessary for* TRIANGDDPM *and* TRIANGCFM*, it alters the spectrum of the matrices, which is usually crucial in the downstream classification tasks. The advantage of the proposed* DIFFEOCFM *lies in the fact that it does not require this step, and thus naturally preserves the matrix spectrum.*

# N    DIFFEOCFM on EEGs

## N.1    Filter's Topographic Map

**Common Spatial Pattern**

The Common Spatial Pattern (CSP) filter is a supervised spatial filtering technique widely used in EEG-based signal analysis, particularly for motor imagery classification. It enhances class-discriminative information by projecting multi-channel EEG data onto a low-dimensional space that maximizes variance differences between two classes [6]. Formally, for a signal $x(t) \in \mathbb{R}^{N_C}$, the CSP algorithm seeks a projection matrix $W \in \mathbb{R}^{N_C \times N_C}$ such that the transformed signal $x_{CSP}(t) = W^T x(t)$ yields maximally discriminative variance patterns across classes, where $N_C$ is the number of electrode channels. Each column vector $w_j \in \mathbb{R}^{N_C}$ $(j = 1, \cdots, N_C)$ of matrix $W$ is called the filter of CSP.

Generally, CSP is a spatial filtering method that effectively enhances the discrimination of mental states characterized by event-related de-synchronization/synchronization (ERD/ERS), which commonly occurs in the alpha (8–12 Hz) and beta (13–30 Hz) bands during motor imagery tasks [63].

**Experimental Settings**

In this experiment, we aim to demonstrate whether the proposed model, DIFFEOCFM, can learn similar spatial filter patterns $\{w_j\}_{j=1}^{N_C}$ in the alpha and beta frequency bands during motor imagery, as reflected in their corresponding topographies.

Specifically, we conduct this analysis only on the BNCI2015-001 dataset, as it provides the predefined order of 13 EEG channels, allowing us to correctly map each channel to its corresponding position in the topographic montage. In contrast, the BNCI2014-002 dataset does not offer such channel ordering information.

All filters were computed using data from session 1 of each subject individually, in order to remain consistent with the cross-session experimental setting. The filters for real data were computed directly using the CSP algorithm on session 1 data from each subject, focusing on the 8-12 Hz (alpha) and 13-30 Hz (beta) frequency bands during the first 2 seconds following the cue. For the generated data, filters were obtained by applying the same CSP procedure on generated datasets produced by DIFFEOCFM, with the same number of samples as in session 1. In both cases, only the spatial filter corresponding to the largest eigenvalue from CSP's generalized eigenvalue decomposition was retained for analysis.

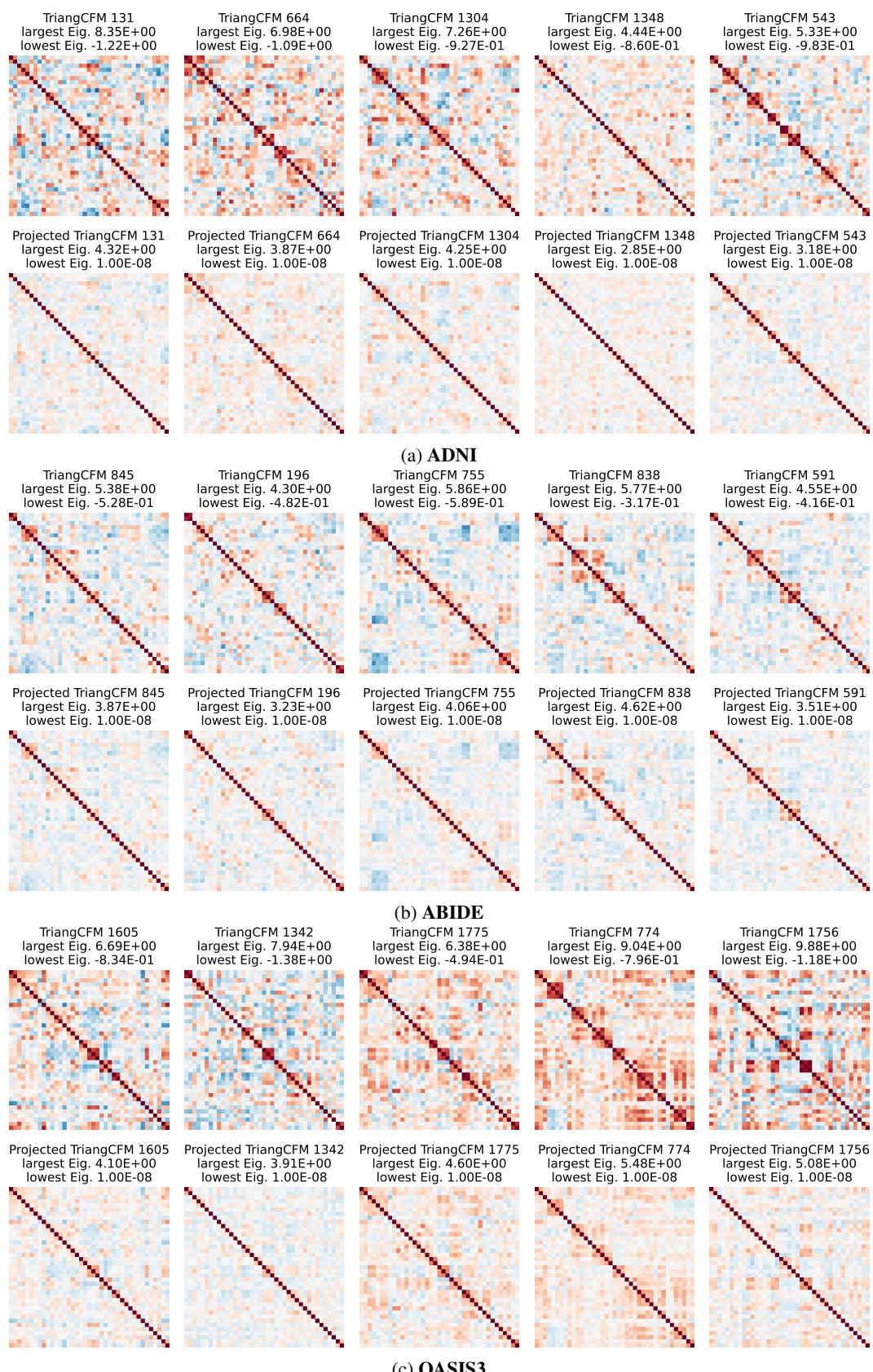

Figure 5: Comparison of fMRI matrices before and after projection

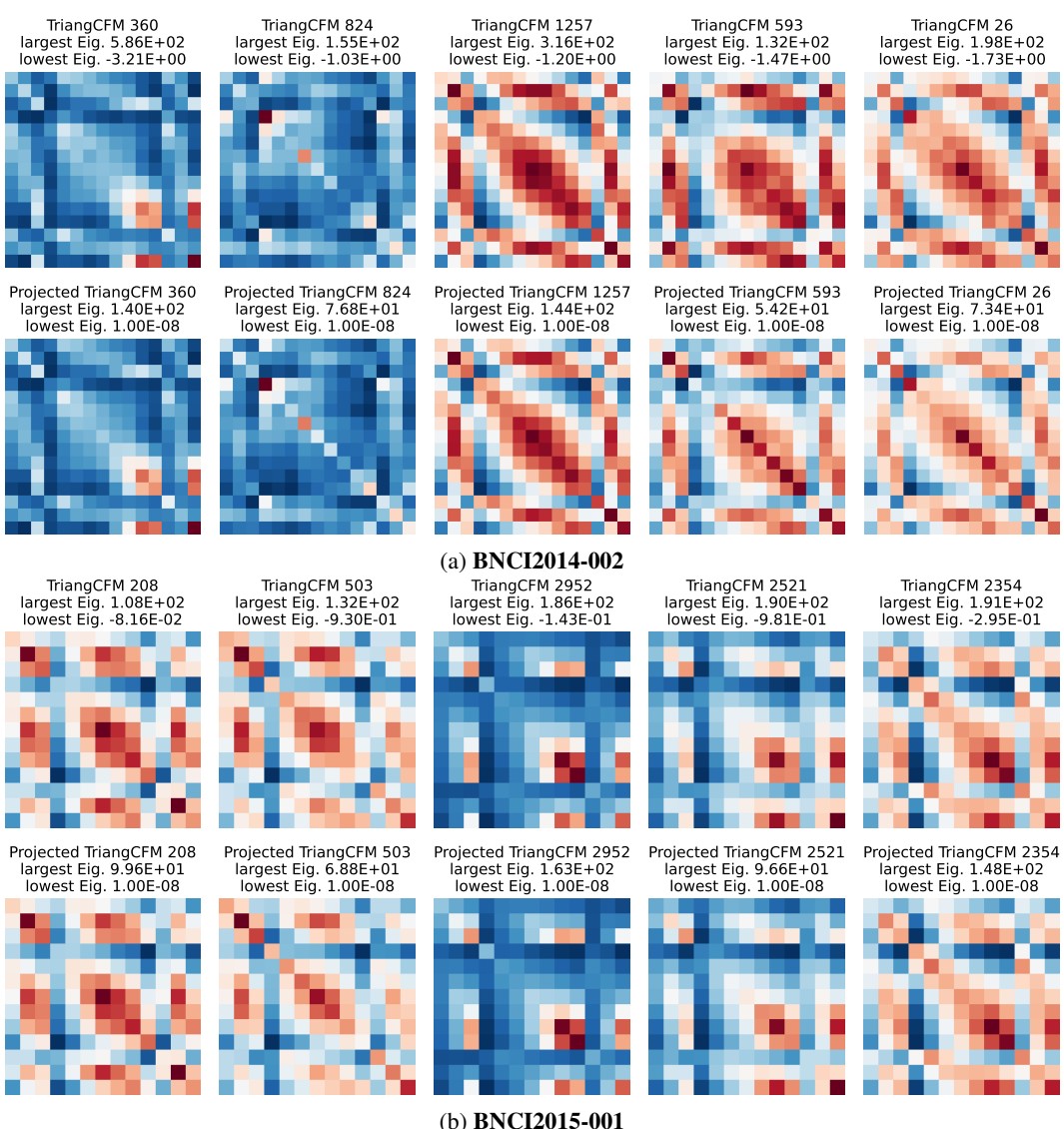

(a) **BNCI2014-002**

(b) **BNCI2015-001**

Figure 6: Comparison of EEG matrices before and after projection

## N.2 Results and Analysis

Figure 7 shows the topographic map using the first CSP filter for each subject in the BNCI2015-001 dataset (Subjects 1-6), while Figure 8 displays the topographic map using the first CSP's filter for each subject in the same dataset (Subjects 7-12). We can observe that, except for Subjects 5, 6, 10, and 12, the topographies for the remaining subjects appear highly similar, with a noticeable increase in signal amplitude in the C3 region. Moreover, the average topographic map of these subjects, shown in Figure 2b, also reveals a very similar filter pattern.

This results show that the model, DIFFEOCFM, used for generating the generated data has successfully captured the key characteristics and patterns of the real EEGs. This similarity in CSP's filters indicates that **the generated data closely resembles the real data in terms of spatial patterns of brain activity**, which is crucial for validating the effectiveness of the model in simulating realistic neural processes, particularly for motor imagery tasks. Additionally, it may imply that **the model has learned to preserve the underlying structure and discriminative features that are typically seen in real EEG data during motor imagery**.

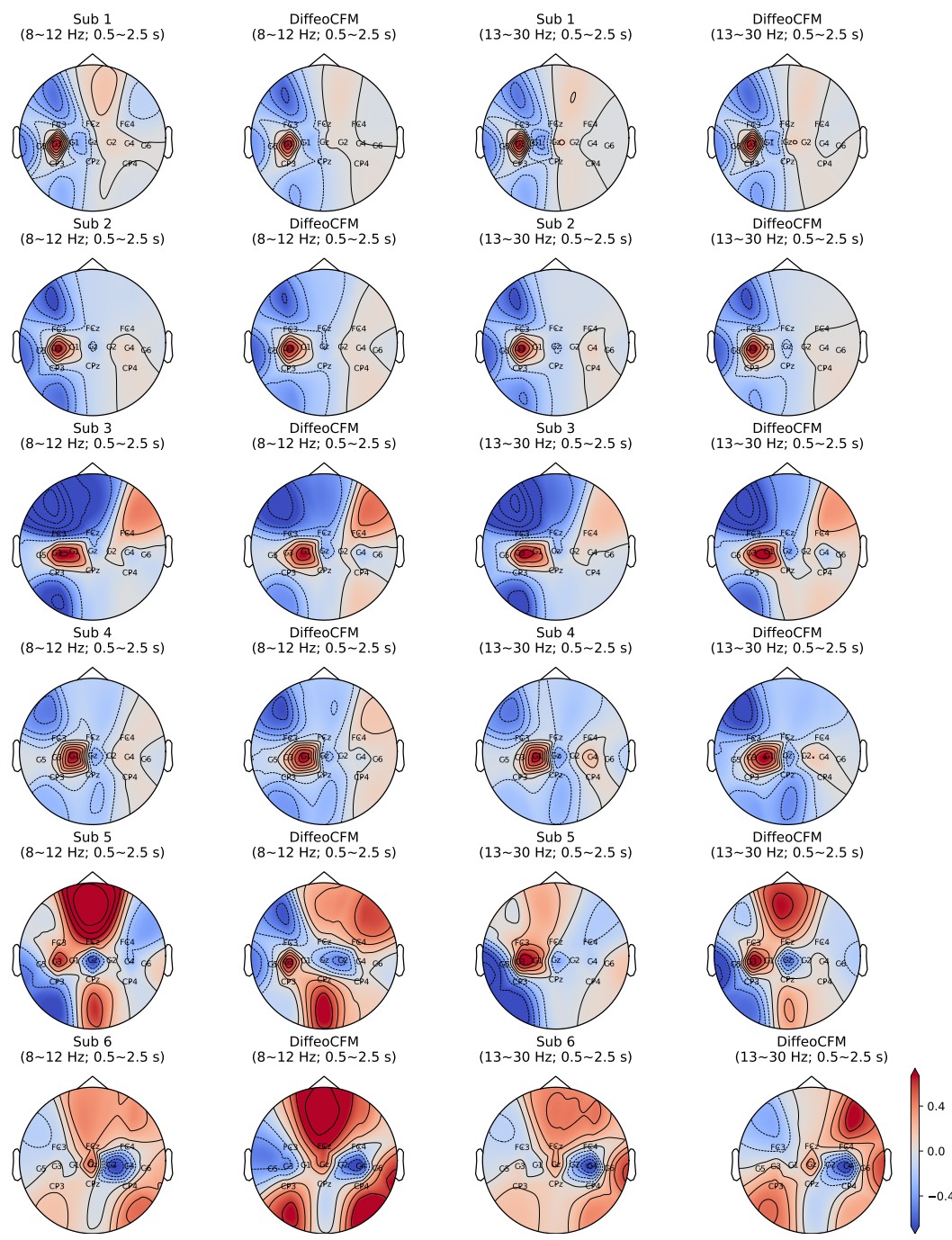

Figure 7: Subject-level topographic map using the first CSP's filter derived from real EEGs (BNCI2015-001, Sub 1-6) and generated data by DIFFEOCFM

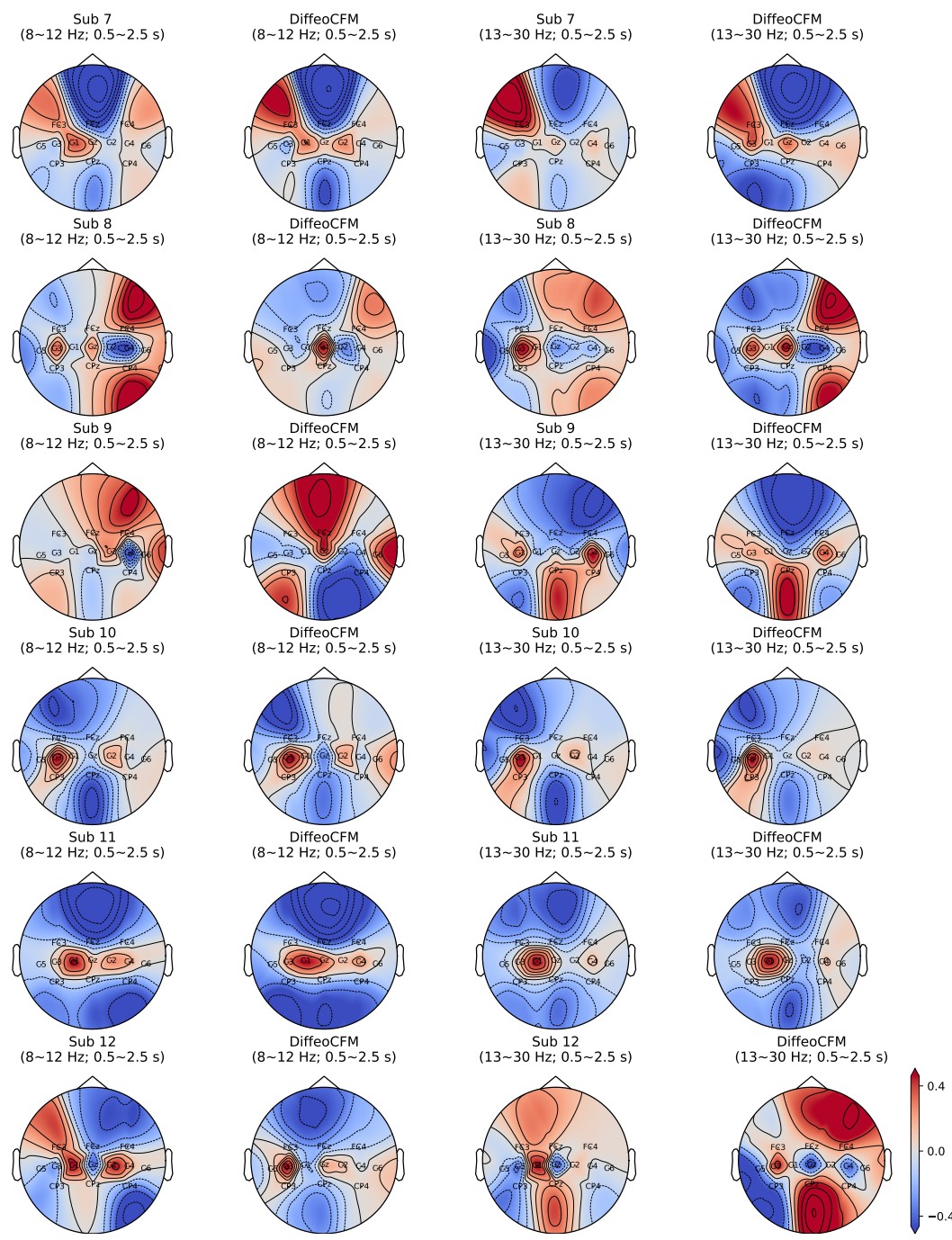

Figure 8: Subject-level topographic map using the first CSP's filter derived from real EEGs (BNCI2015-001, Sub 7-12) and generated data by DIFFEOCFM

# O  DIFFEOCFM on fMRI

## O.1  fMRI Connectome Plotting

**Experimental Settings**

We use Nilearn [14] for neuroimaging analysis and visualization based on the MSDL brain atlas. The atlas is retrieved using the fetch_atlas_msdl function, which provides probabilistic maps of functional brain networks in the form of a 4D NIfTI image. The maps attribute extracts this image, while find_probabilistic_atlas_cut_coords computes representative 3D coordinates for each network, typically corresponding to the spatial center of the activation, to support anatomical localization. In addition, predefined region labels are extracted to annotate the corresponding brain areas. These components together enable the visualization of brain networks, including plotting the regions with spatial coordinates and labels.

The connectome plots are grouped into two categories: "CN" and "non-CN". The adjacency_matrix used in nilearn.plotting.plot_connectome is computed as the Fréchet mean of the corresponding group of connectivity matrices in all the 5-fold experiments, and all edge thresholds in the plots are set to 90%.

**Results and Analysis**

Figure 9a and Figure 9b visualize group-averaged functional connectomes derived from two distinct neuroimaging datasets using the Fréchet mean. By thresholding edges to retain only the strongest 10% of connections (90% edge threshold), these plots emphasize dominant, statistically reliable interactions between brain regions while filtering out noise. The use of the Fréchet mean ensures that the group-level adjacency matrices respect the intrinsic geometric structure of covariance data, avoiding distortions caused by arithmetic averaging. **This comparison highlights both the fidelity of data generation methods by aligning generated and real data connectomes using the Fréchet mean and potential differences in functional network organization between the two cohorts.** The workflow underscores the utility of geometric statistics and stringent thresholds for interpreting brain connectivity in heterogeneous populations.

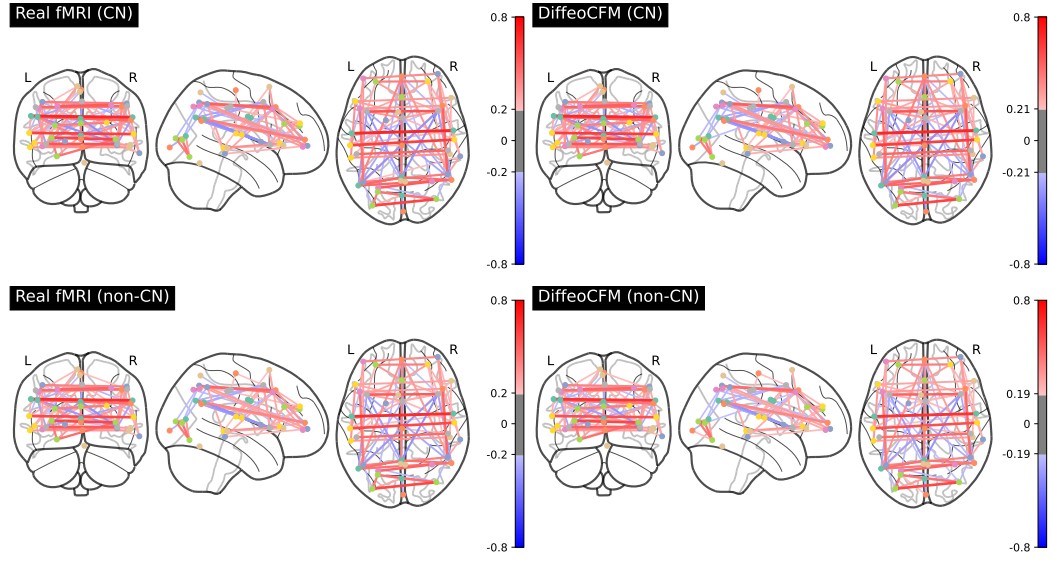

(a) **Class-conditional fMRI functional connectomes using the Fréchet mean (ABIDE).**

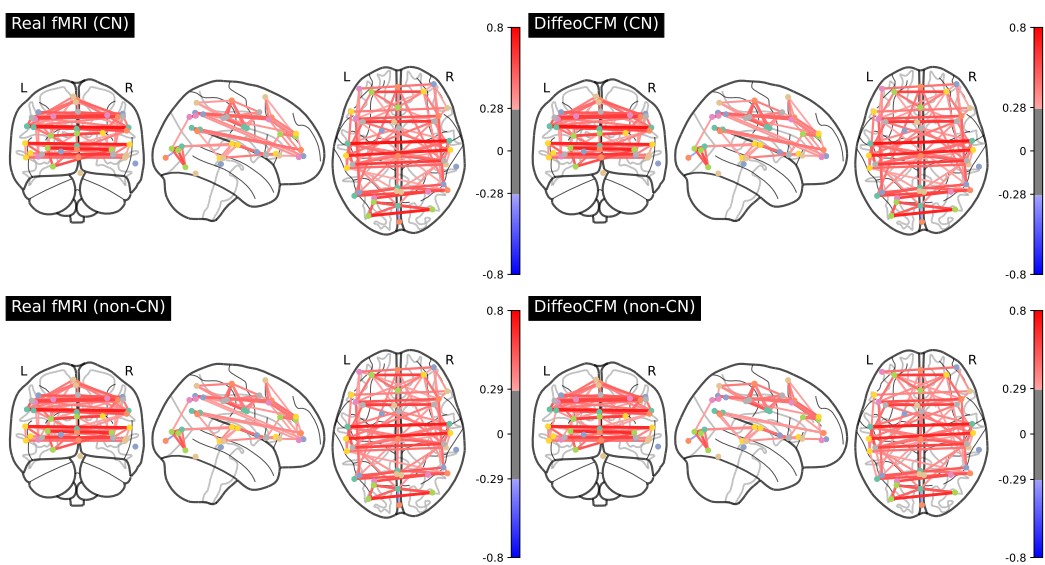

(b) **Class-conditional fMRI functional connectomes using the Fréchet mean (OASIS3).**

Figure 9: Group-level functional connectomes, computed as Fréchet means from two separate neuroimaging datasets, are visualized after applying a 90% edge threshold to retain only the top 10% of strongest connections. This thresholding highlights dominant and statistically robust interactions between brain regions, effectively reducing noise. Leveraging the Fréchet mean preserves the underlying geometric structure of the covariance matrices, avoiding potential distortions introduced by conventional Euclidean averaging.

# P    Plotting of Generated Samples in Real Data Neighborhoods

Figure 10-15 present the selected generated fMRI samples that are nearest to real data points in terms of Frobenius distance, categorized by "CN" and "non-CN" from the ADNI, ABIDE, and OASIS3 datasets.. Figure 16-19 show the selected generated EEG samples that are nearest to real samples in Frobenius distance, separately for hand and feet motor imagery tasks from the BNCI2014-002 and BNCI2015-001 datasets. Each real sample is annotated with its index in the original dataset. The first, second, and third columns correspond to samples generated by DIFFEOGAUSS, TRIANGCFM, and DIFFEOCFM, respectively. The numeric values on the generated samples indicate the Frobenius distance to the corresponding real sample. Each generated sample shown is the nearest one (in Frobenius distance) to the real sample within its class. Due to its consistently highest recall across multiple datasets, DIFFEOCFM is more likely to produce generated samples that are closer to real samples, i.e., with the smallest Frobenius distance, compared to those generated by other methods. The matrices generated by TRIANGCFM shown in the figures are presented without applying the projection step. Before applying the projection, the outputs of TRIANGCFM already exhibit strong recall scores.

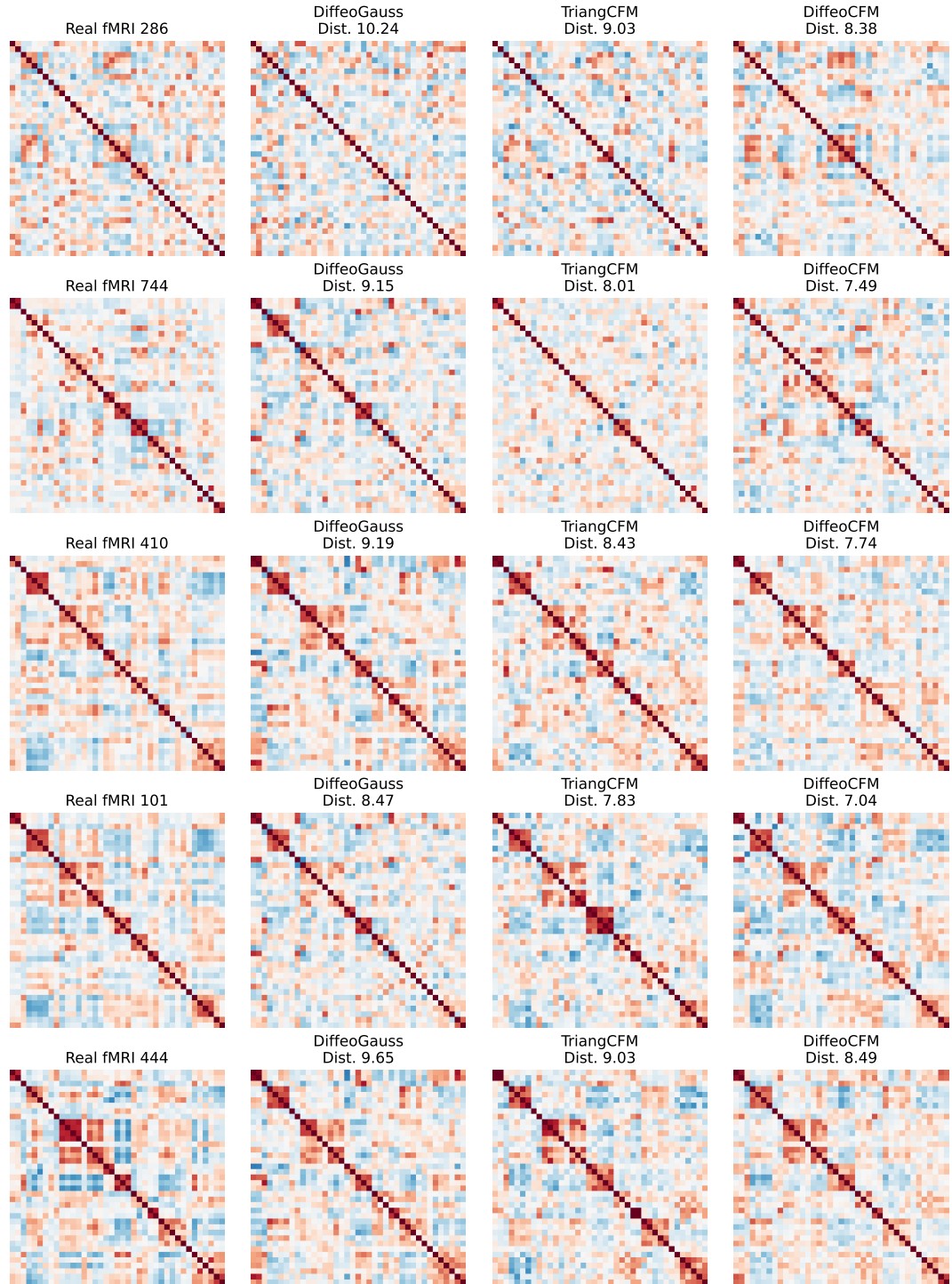

Figure 10: Nearest Generated Samples in Real Data Neighborhoods: ADNI-CN Cohort. The matrices generated by TRIANGCFM shown in the figures are presented without applying the projection step.

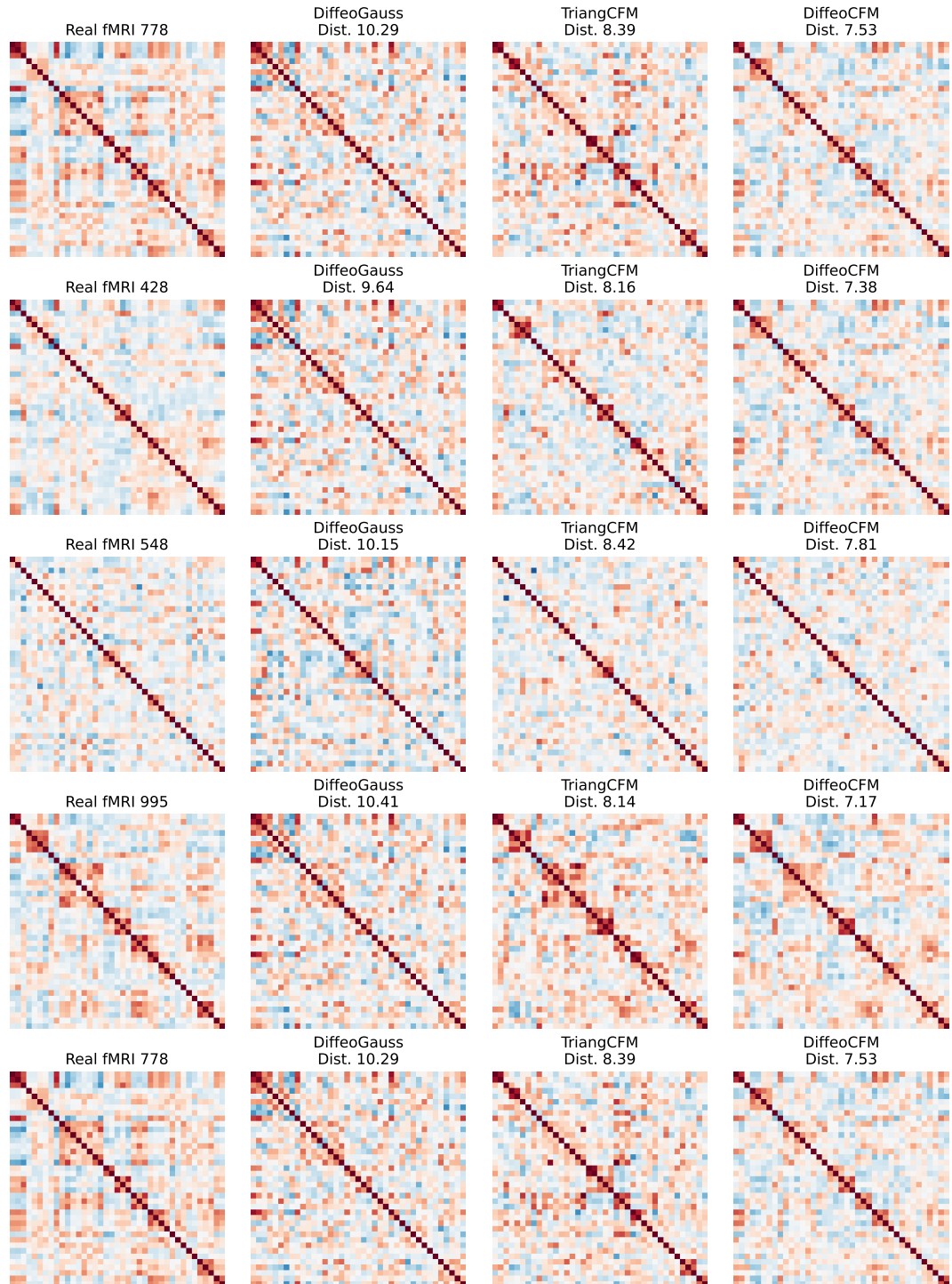

Figure 11: Nearest Generated Samples in Real Data Neighborhoods: ADNI-nonCN Cohort. The matrices generated by TRIANGCFM shown in the figures are presented without applying the projection step.

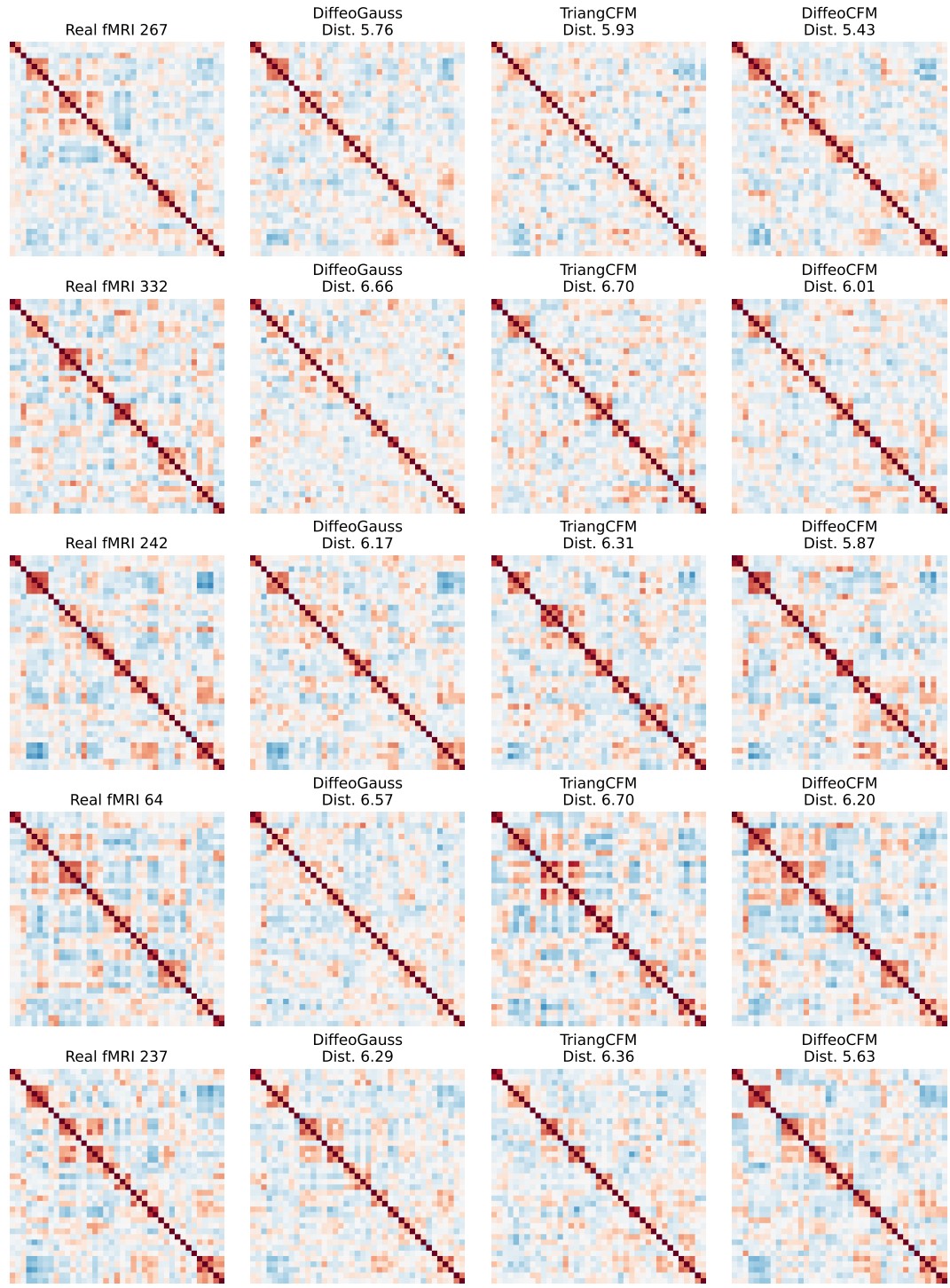

Figure 12: Nearest Generated Samples in Real Data Neighborhoods: ABIDE-CN Cohort. The matrices generated by TRIANGCFM shown in the figures are presented without applying the projection step.

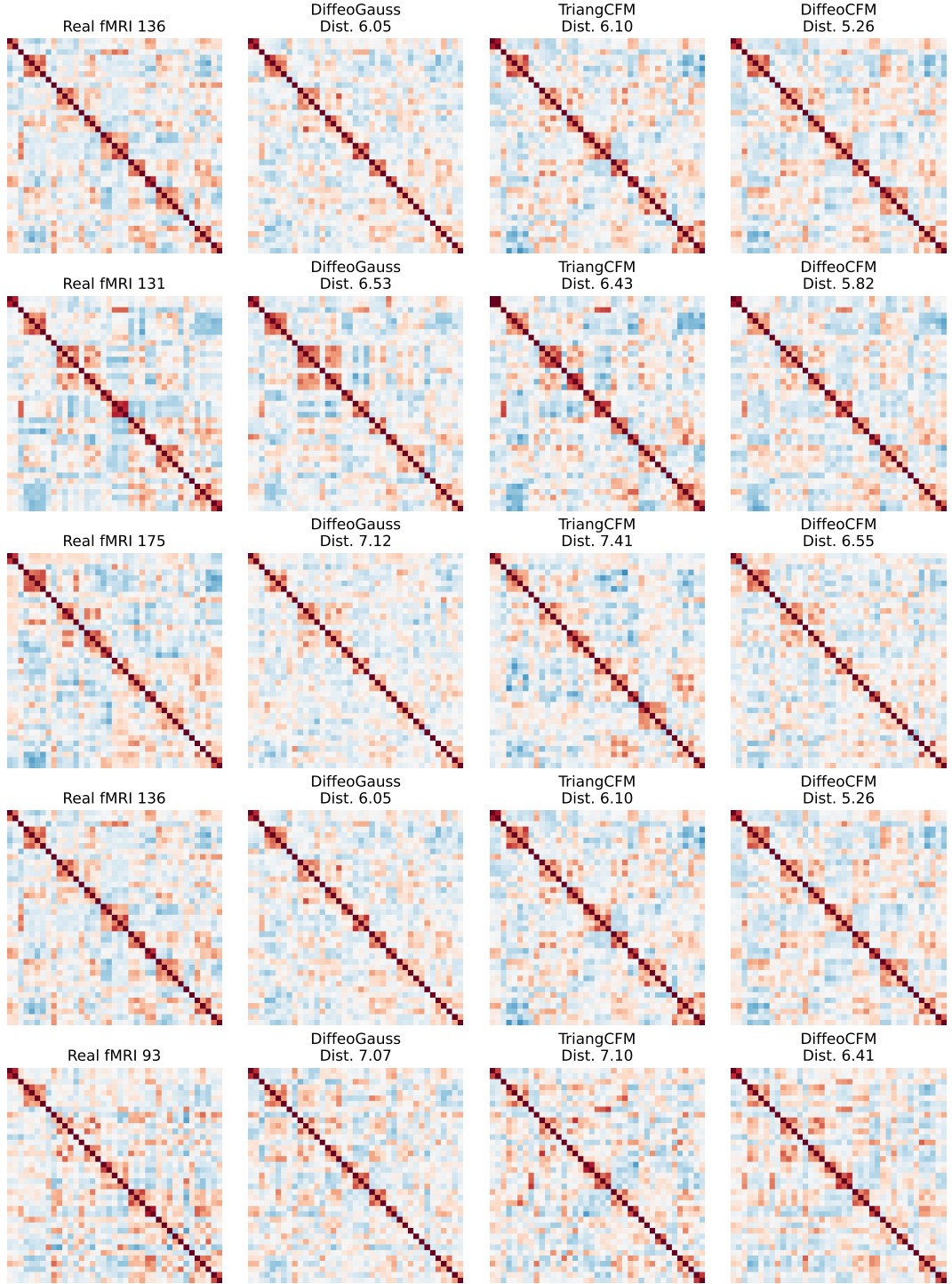

Figure 13: Nearest Generated Samples in Real Data Neighborhoods: ABIDE-nonCN Cohort. The matrices generated by TRIANGCFM shown in the figures are presented without applying the projection step.

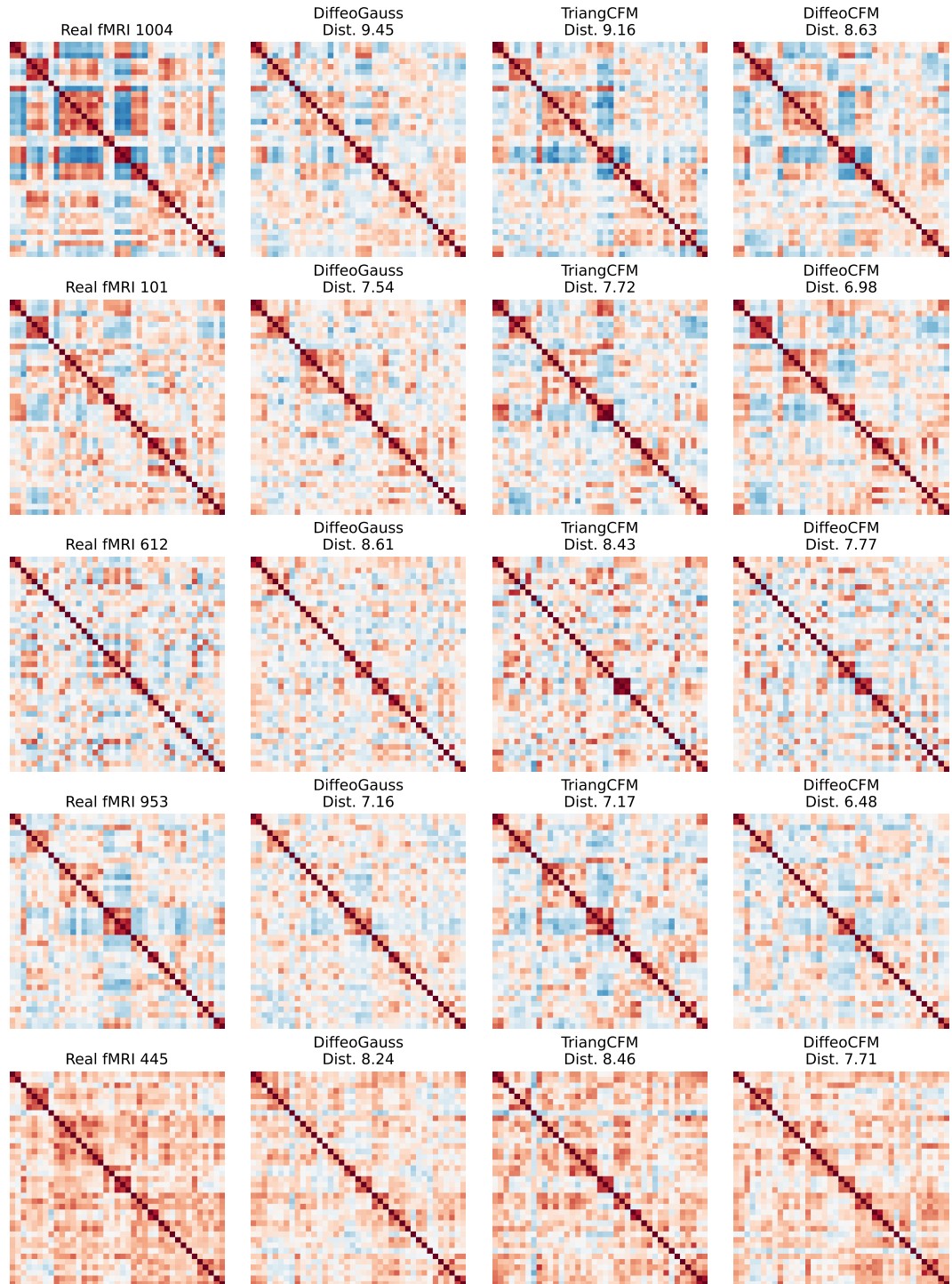

Figure 14: Nearest Generated Samples in Real Data Neighborhoods: OASIS3-CN Cohort. The matrices generated by TRIANGCFM shown in the figures are presented without applying the projection step.

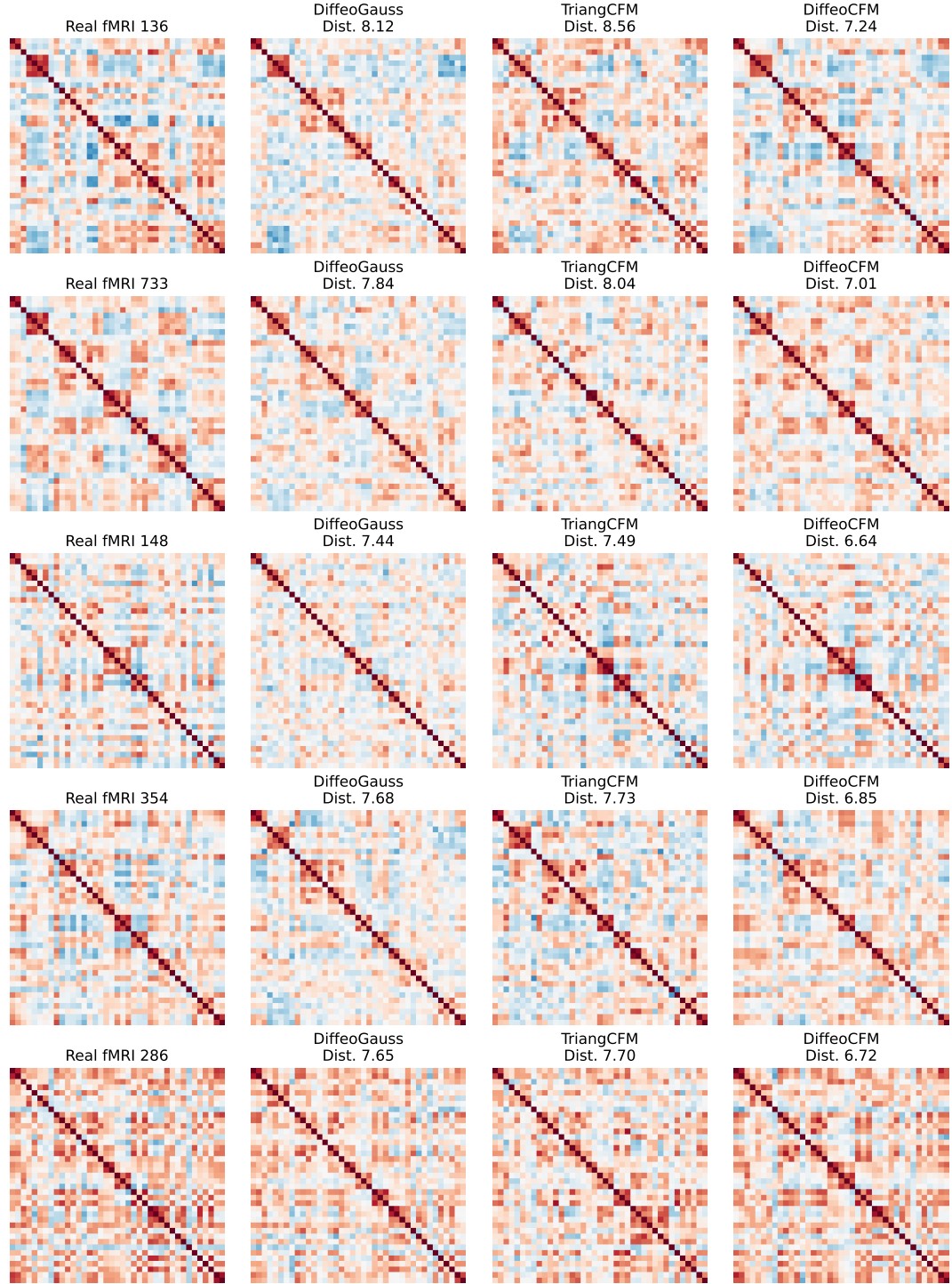

Figure 15: Nearest Generated Samples in Real Data Neighborhoods: OASIS3-nonCN Cohort. The matrices generated by TRIANGCFM shown in the figures are presented without applying the projection step.

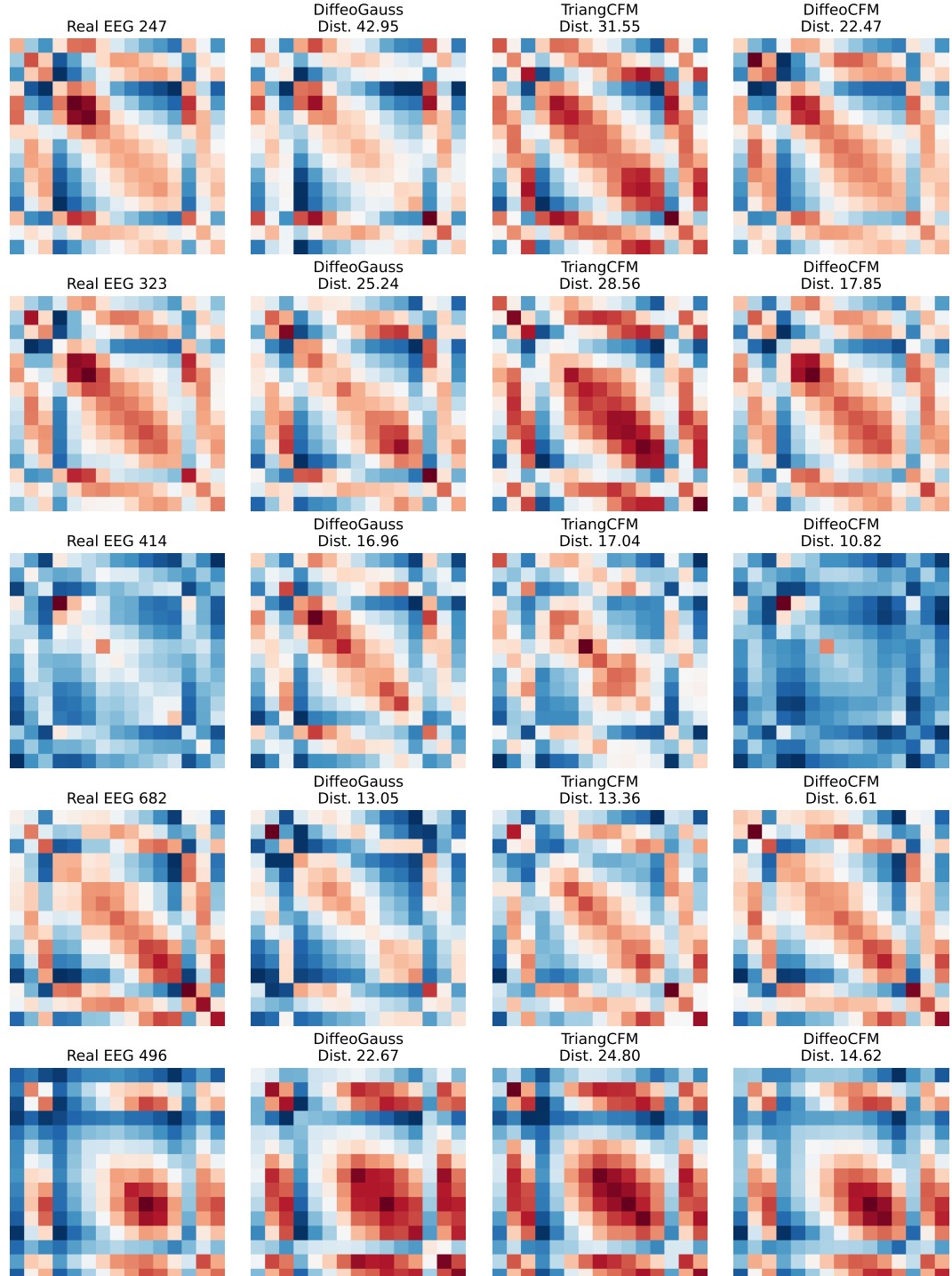

Figure 16: Nearest Generated Samples in Real Data Neighborhoods: BNCI2014-002-Right Hand.

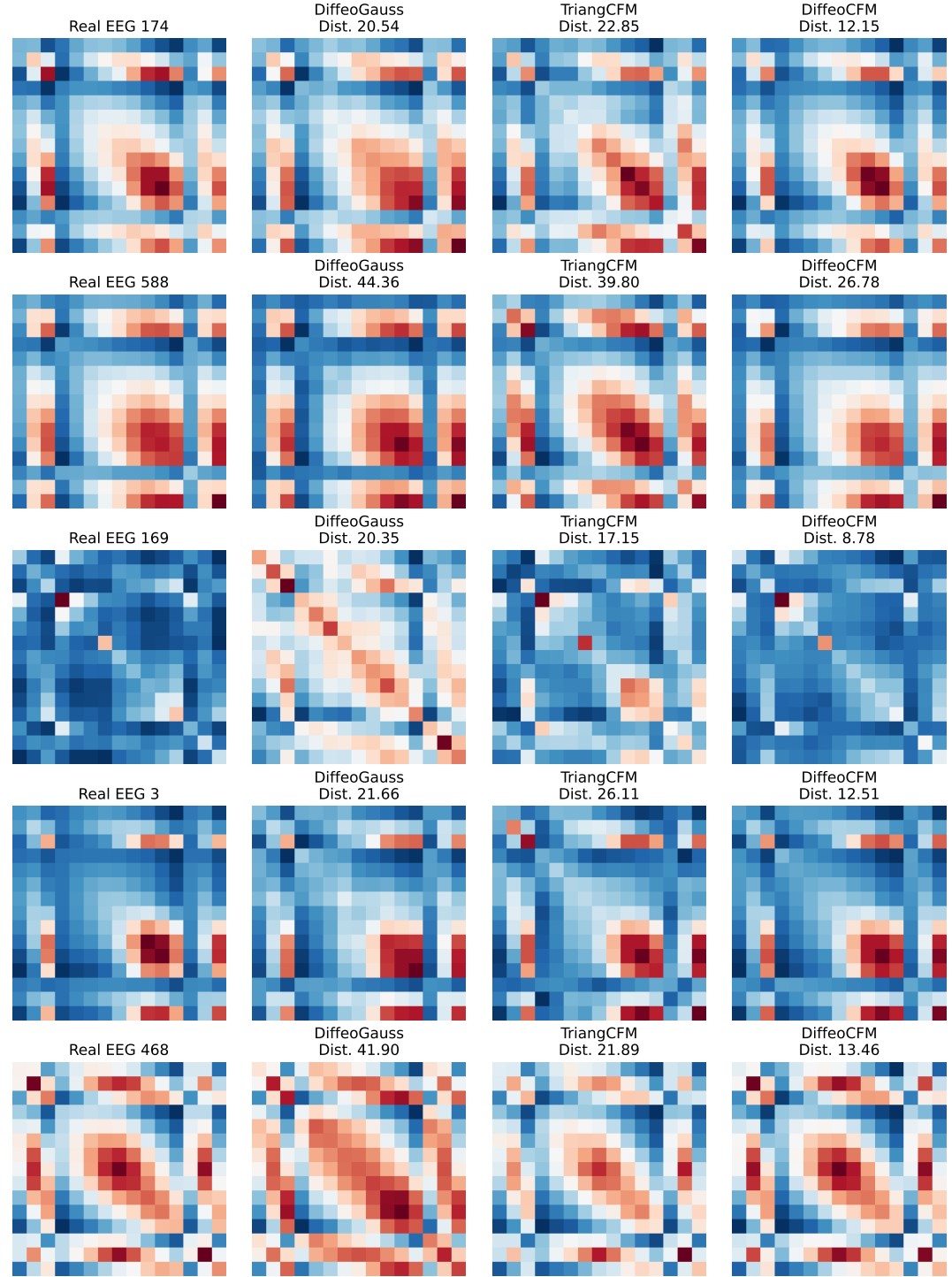

Figure 17: Nearest Generated Samples in Real Data Neighborhoods: BNCI2014-002-Feet.

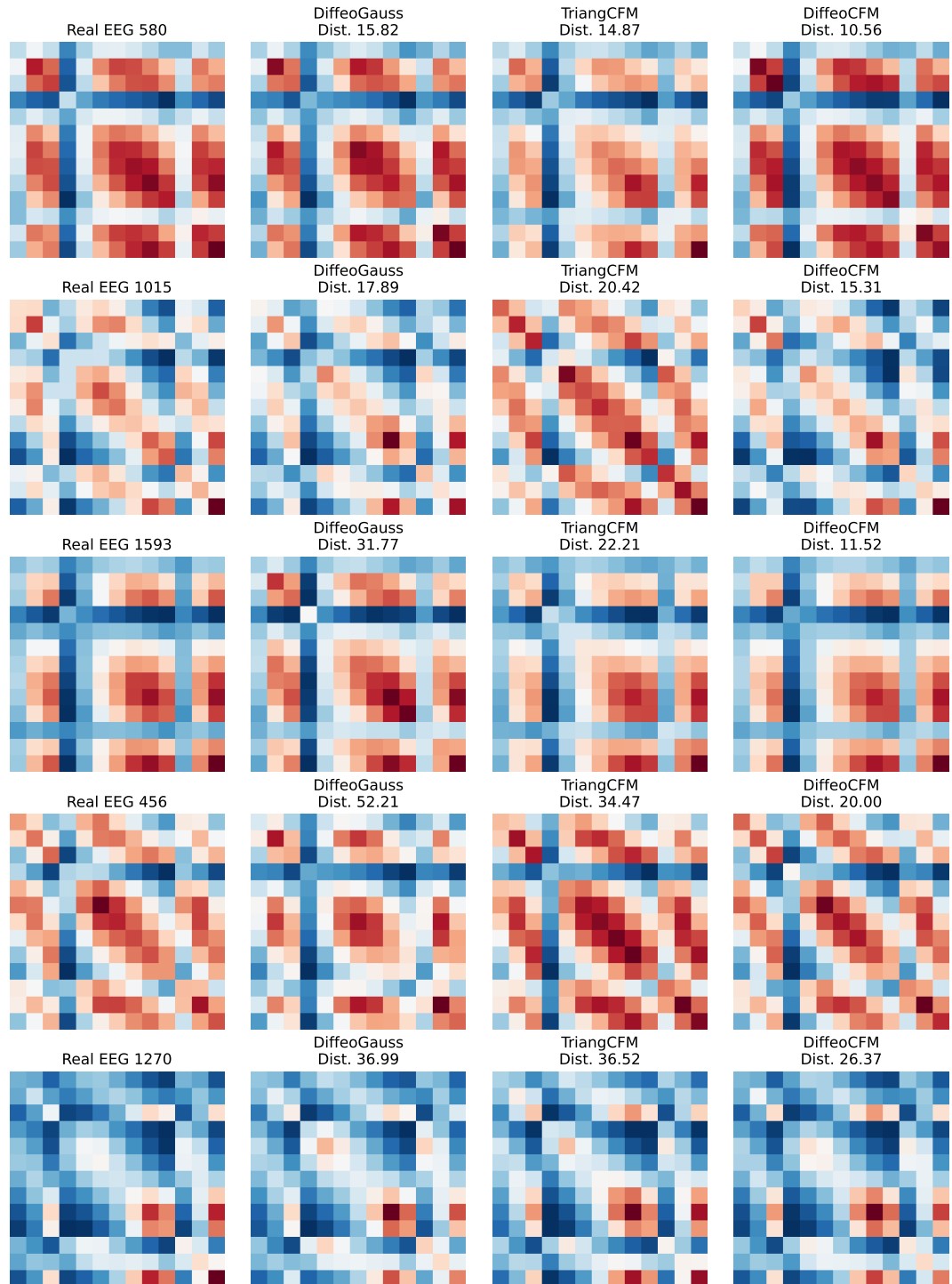

Figure 18: Nearest Generated Samples in Real Data Neighborhoods: BNCI2015-001-Right Hand.

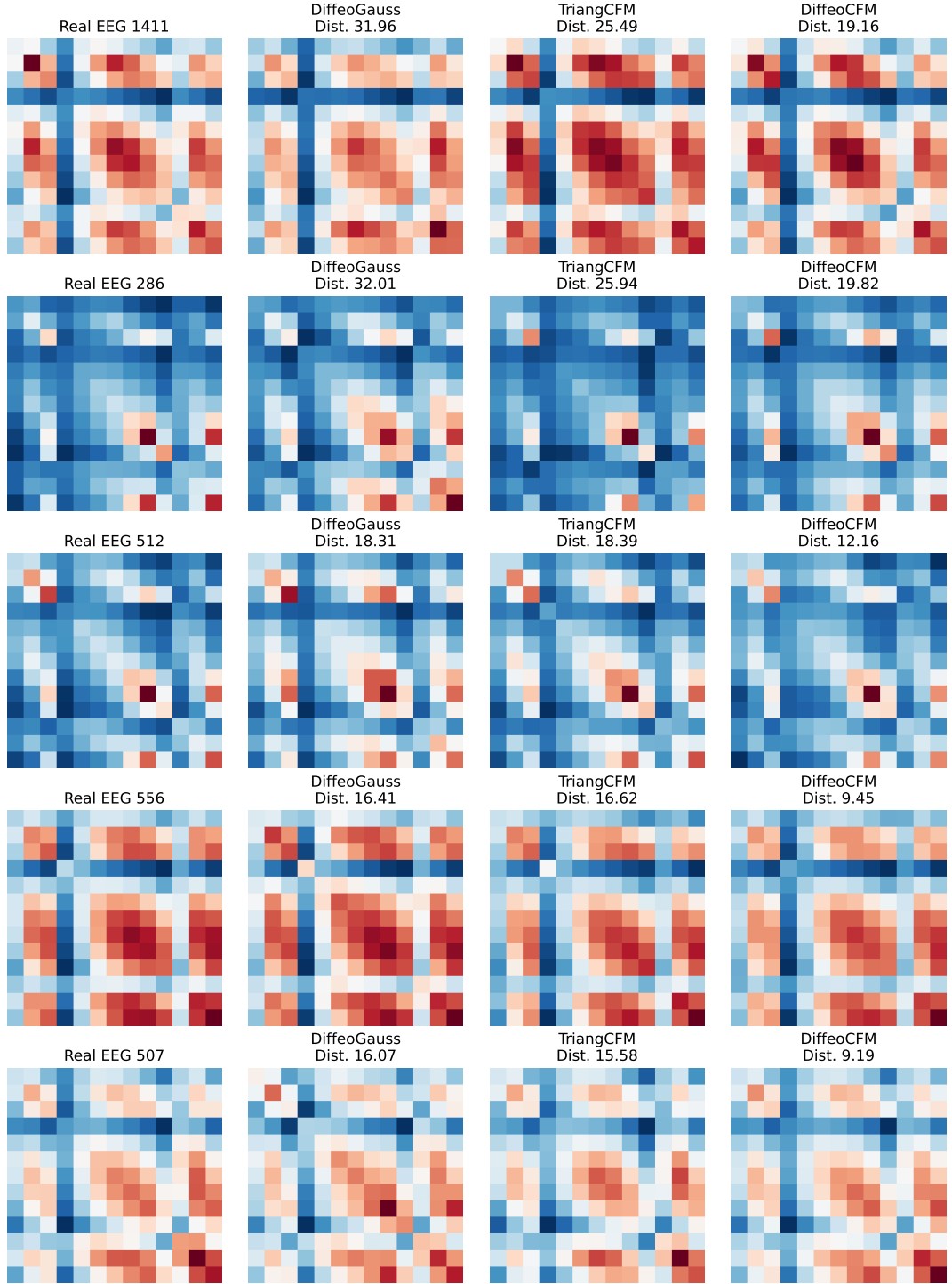

Figure 19: Nearest Generated Samples in Real Data Neighborhoods: BNCI2015-001-Feet.

