# OpenReview forum: "Riemannian Flow Matching for Brain Connectivity Matrices via Pullback Geometry"
_NeurIPS.cc/2025/Conference — NeurIPS 2025 poster_

### Official Review · Reviewer_hktp · 2025-06-30

**Clarity:** 3
**Significance:** 2
**Originality:** 2
**Rating:** 4
**Confidence:** 3

**Summary:**

This paper proposes a new method for generating realistic brain connectivity matrices, i.e. symmetric positive definite (SPD) and correlation matrices. Generative modeling with such data requires one to take into account their Riemannian geometries, which is a computationally expensive task. To reduce the computational complexity of the task, the authors focus on Pullback metrics which turns Riemannian computations into Euclidean computations. Generative modeling can then be easily and efficiently performed on Euclidean space. Experimental results on fMRI and EEG motor imagery datasets show the effectiveness of the proposed method.

**Questions:**

- State-of-the-art works for generative modeling on Riemannian manifolds such as [9,14] use NLL (Negative log-likelihood) scores
and NFE (number of function evaluations) as evaluation metrics. I think those metrics are relevant in the context of the present work
and are worth investigating (e.g. by using synthetic data as in [9,14]). This would further demonstrate the advantage of the proposed
method w.r.t. [9,14]. Currently, those metrics are not considered in the experiments. Could the authors elaborate on this choice of
evaluation metrics ?

- The authors do not use the methods in [9,14] for comparison with the proposed method. I carefully checked those references [9,14] and noticed that both works are applicable (and can be easily adapted) to SPD manifolds, making them strong baselines. The authors of [9] even included SPD manifolds as part of their code. However, the authors  state in Appendix G that "its current implementations are limited to a few specific manifolds such as spheres and tori, which are not directly applicable to our task.". Could the authors elaborate on their choice of baselines ?

- The work in [A] is closely related to the present work but is missing in the references. The proposed generative method
for SPD manifolds can be directly derived from [A], since SPD manifolds with Log-Euclidean metrics have Lie group structure.
I think it should be cited in the related works and eventually be used in comparison with the proposed method.

**References**

Luca Falorsi, Pim de Haan, Tim R. Davidson, Patrick Forré: Reparameterizing Distributions on Lie Groups. AISTATS 2019: 3244-3253.

**Ethical Concerns:**

["NO or VERY MINOR ethics concerns only"]

**Final Justification:**

In my initial review, I raised a concern regarding the evaluation metrics and a lack of comparison of the proposed methods against some state-of-the-art methods. During the rebuttal, the authors clearly explained their choice of evaluation metrics. They also provided experimental results which demonstrate the advantage of their methods w.r.t state-of-the-art methods. For this reason, I have decided to update my initial score from 2 to 4.

**Limitations:**

Yes

**Paper Formatting Concerns:**

I did not notice any major formatting issues in this paper.

**Quality:**

2

**Strengths And Weaknesses:**

Strengths:
- The proposed method relies on Pullback metrics which are well-established and have proven efficient in extending traditional machine learning models to the Riemannian settting.
- Theoretical justifications are provided in the supplemental material.
- The considered applications and datasets are relevant. Experimental results are promising.

Weaknesses:
- Important baselines are missing, making it hard to evaluate real benefits of the proposed method w.r.t. existing works.

---

> ### Author Rebuttal · Authors · 2025-07-29
>
> We thank the reviewer for their detailed feedback and for highlighting the strengths of our work. We are pleased that the **use of pullback metrics**, the **theoretical justification** provided in the supplementary material, and the **relevance of the considered applications and datasets** were appreciated.
>
> We now address the reviewer’s comments and questions, recalling each point in their own words before providing our response.
>
>
> ## Choice of baselines
> > Important baselines are missing, making it hard to evaluate real benefits of the proposed method w.r.t. existing works.
> > The authors do not use the methods in [1,2] for comparison with the proposed method. [...] Could the authors elaborate on their choice of baselines ?
>
> **Adding baseline SPD-CFM [1].** DiffeoCFM (proposed) is, to our knowledge, the first generative model that handles both SPD and correlation matrices while respecting manifold structure and being computationaly efficient. We acknowledge that [1] includes an SPD-manifold implementation, which we initially overlooked, and thank the reviewer for pointing it out. We adapted the official implementation of SPD-CFM to use the same conditional prior as DiffeoCFM and evaluated on BNCI 2014‑002 by holding out each of the first two sessions in turn for testing while training on all remaining sessions. The results are shown in the next table; BNCI 2015‑001 yielded similar results.
>
> Quality metric: $\alpha$,$\beta$-F1
>
> Classification Accuracy Score (CAS) metrics: ROC-AUC, F1
>
> | Method   | $\alpha$,$\beta$-F1   | ROC-AUC  | F1  | Training time (s.)
> |:-----------------|:-----------------------------------------------------------|:-----------------------------------------------|:------------------------------------------|:--------------------------------------------------------|
> | Real Data         | 0.67 ± 0.07                                                | 0.86 ± 0.01                                    | 0.77 ± 0.02                               | N/A                                        |
> | DiffeoGauss                        | 0.56 ± 0.06                                                | 0.83 ± 0.01                                    | 0.75 ± 0.02                               | **0.09 ± 0.01**                                    |
> | TriangDDPM                            | 0.15 ± 0.04                                                | 0.51 ± 0.03                                    | 0.22 ± 0.12                               | 265.24 ± 1.96                                           |
> | TriangCFM                                   | 0.26 ± 0.04                                                | 0.54 ± 0.03                                    | 0.24 ± 0.14                               | 261.15 ± 1.15                                           |
> | SPD-CFM                                     | 0.59 ± 0.10                                                | 0.82 ± 0.02                                    | 0.73 ± 0.03                               | 1656.78 ± 4.86                                          |
> | DiffeoCFM   (proposed)                         | **0.65 ± 0.07**                                       | **0.84 ± 0.02**                           | **0.76 ± 0.03**                      | 265.94 ± 4.17
>
> **Results interpretation.** SPD-CFM [1] performs slightly worse than DiffeoCFM. Moreover, **SPD-CFM is 6x slower**, which is expected since it relies on geodesic computation and its derivative, the Riemannian norm, and a Riemannian ODE solver. In contrast, DiffeoCFM first maps data to a Euclidean space via the diffeomorphism $\phi$, trains a standard Euclidean CFM [3] (which is very fast), and then maps generated data back to the manifold via $\phi^{-1}$. **We will complete Table 1 of the manuscript with this additional baseline.**
>
> **SPD-DDPM [4].** The Riemannian diffusion framework of [2] is instantiated on the SPD manifold in SPD-DDPM [4]. We have already benchmarked this model in Appendices G–H. Figure 3 (Appendix H) shows that **SPD-DDPM is over 1000 ×** slower than DiffeoCFM due to the heavy computational cost of Riemannian manifold operations, particularly manifold sampling.
>
> **Choice of baselines.** The other baselines were chosen for their complementary properties:
> - DiffeoGauss fits a multivariate Gaussian distribution per class, which—despite its simplicity—performs surprisingly well, suggesting that simple yet effective priors can be valuable for neuroscience applications.
> - TriangCFM and TriangDDPM are straightforward parametrizations that learn the lower-triangular parts with a Euclidean CFM; they are as fast as DiffeoCFM but, lacking diffeomorphic parametrizations, show lower precision/recall.
> - The Real Data baseline compares real-vs-real samples with different splits, providing an upper bound on attainable performance. On EEG, DiffeoCFM reaches this bound, whereas on fMRI, there remains a gap to close.
>
> This setup exposes key challenges in neuroscience and motivates the use of DiffeoCFM as a simple yet high-performing method.
>
> [1] Chen et al., *Flow Matching on General Geometries*, ICLR 2024
>
> [2] De Bortoli et al., *Riemannian score-based generative modelling*, NeurIPS 2022
>
> [3] Lipman et al., *Flow Matching for Generative Modeling*, ICLR 2023
>
> [4] Li et al., *Spd-ddpm: Denoising diffusion probabilistic models in the symmetric positive definite space*, AAAI 2024
>
> ## Choice of evaluation metrics
> > State-of-the-art works for generative modeling on Riemannian manifolds such as [1,2] use NLL (Negative log-likelihood) scores and NFE (number of function evaluations) as evaluation metrics. [...] Currently, those metrics are not considered in the experiments. Could the authors elaborate on this choice of evaluation metrics ?
>
> **NFE.** First, all models in our experiments solve the ODE with the same fixed-step midpoint solver ($dt = 0.001$), yielding an **identical number of function evaluations (NFE)**. We therefore did not report NFE.
>
> **NLL.** Then, regarding **negative log-likelihood (NLL)** versus sample-level metrics $\alpha$‑precision, $\beta$‑recall, and $\alpha,\beta$‑F1 [5]:
> - **NLL feasibility:** Its computation **requires a diffeomorphism** between the prior and the learned distribution so that the change-of-variables log-det Jacobian is well defined. Two important baselines—TriangCFM and TriangDDPM—generate samples in ambient space and project them to the SPD (or correlation) manifold. The projection breaks injectivity, so the log-det Jacobian (and hence NLL) is undefined.
> - **Interpretability:** The sample-level metrics compare generated to real data on a 0–1 scale, with an empirical upper bound from real-vs-real comparisons. They are therefore directly interpretable: in Table 1, DiffeoCFM reaches this bound on EEG and approaches it on fMRI. Hence they highlight generative-model challenges in Neuroscience, whereas absolute NLL values are harder to interpret.
> - **Conditional generation:** Unlike [1, 2], our work targets **conditional generation**—essential in neuroscience, where data must be generated given patient-level variables (here, diagnosis). To assess conditional fidelity, we use the **Classification Accuracy Score (CAS)** [6], which trains a state-of-the-art classifier on generated data and evaluates it on real data, directly testing preservation of the conditional structure.
>
> [5] Alaa et al., *How faithful is your synthetic data? sample-level metrics for evaluating and auditing generative models*, ICML 2022
>
> [6] Ravuri et al., *Classification accuracy score for conditional generative models*, NeurIPS 2019.
>
>
> ## Additional reference: generation on Lie groups
> > The work in [7] is closely related to the present work but is missing in the references. The proposed generative method for SPD manifolds can be directly derived from [7], since SPD manifolds with Log-Euclidean metrics have Lie group structure. I think it should be cited in the related works and eventually be used in comparison with the proposed method.
>
> **Contributions of [7].** The authors extend the reparameterisation trick to Lie groups and demonstrate how normalising flows can leverage this approach to learn probability densities on such spaces.
>
> **Manuscript update.** Since both [7] and DiffeoCFM rely on reparameterisation tricks to estimate probability densities, [7] can be viewed as an early precursor to our approach. We have therefore added the following to the manuscript:
> >> **Added in Section 3.3 – Related Work**
> In [7], the authors leverage reparameterisation with normalising flows to learn probability densities on Lie groups (non-Euclidean spaces), and thus can be seen as an early precursor to our approach.
>
> **How we differ.** Our contribution differs in both method and scale. We adopt the newer flow-matching framework and show that Riemannian Flow Matching [7] with pull-back metrics reduces to our proposed DiffeoCFM. The latter performs all operations in Euclidean space, yielding greater numerical stability and improved training and sampling efficiency. Moreover, while [7] presents low-dimensional examples, our experiments reach 741-dimensional fMRI connectivity (39 × 39 matrices), a particularly challenging regime.
>
> [7] Luca Falorsi et al., *Reparameterizing Distributions on Lie Groups*, AISTATS 2019
>
> We hope these results clarify our choices and demonstrate DiffeoCFM’s advantages. Thank you again for the constructive feedback.

---

> > ### Comment · Reviewer_hktp · 2025-08-03
> >
> > Thank you for the detailed answers.
> >
> > Could you give more details on the experimental setting of SPD-CFM, in particular the architecture of the vector field ? Did you use the default setting or the optimized one ?
> >
> > I am a bit confused by your statement that "The projection breaks injectivity, so the log-det Jacobian (and hence NLL) is undefined." Isn't training and sampling done with a diffeomorphism as presented in Section 3.1 ?

---

> > > ### Author Response · Authors · 2025-08-03
> > >
> > > ## Details of the SPD-CFM setting.
> > >
> > > We followed Algorithm 1 with the “Simple geometry” configuration from [A] and as implemented in their public repository [B]. In practice, our setup for SPD-CFM is as follows.
> > >
> > > | Component        | Value |
> > > |------------------|-------|
> > > | Prior            | Wrapped class-conditional Gaussian $\mathcal{N}(\mu_y, \Sigma_y)$ |
> > > | Manifold         | `manifm.manifolds.SPD()` implementation from [B] |
> > > | Vector-field network | Vector field parameterization proposed in [A], i.e., defined in the ambient space and projected onto the tangent space at every point. In practice, an MLP($x$, $y$, $t$) with input dimension $d(d+1)/2 + K + 1$, where $K$ is the number of conditioning classes; one hidden layer of width 512 with SELU activation; and output dimension $d(d+1)/2$ |
> > > | Projection       | `manifm.model.arch.ProjectToTangent` with metric normalisation from [B] |
> > > | Optimiser        | AdamW, lr 1e-3, cosine schedule with a 10 epochs warm-up |
> > > | Epochs           | 2000 |
> > > | Batch size       | 64 |
> > > | ODE solver       | Euler scheme with 1000 steps |
> > > | | |
> > >
> > > For fairness, in our experiments SPD-CFM and DiffeoCFM share the same hyperparameters, prior, and vector‑field architecture.
> > >
> > > On BNCI 2014-002 (see table in the previous reply), DiffeoCFM matches or outperforms SPD-CFM on all metrics while running about 6 times faster. This is because SPD-CFM computes Riemannian geodesics, their derivatives, and Riemannian norms while DiffeoCFM uses a global diffeomorphism to work in Euclidean space with a standard CFM, then maps samples back. As a result, by specializing to manifolds with global diffeomorphisms, DiffeoCFM achieves similar accuracy at a fraction of the cost.
> > >
> > > [A] Chen et al., *Flow Matching on General Geometries*, ICLR 2024
> > >
> > > [B] github.com/facebookresearch/riemannian-fm
> > >
> > > ---
> > >
> > > ## NLL computation.
> > >
> > > **DiffeoCFM** (proposed) generates samples through a diffeomorphism $\phi$, as explained in Section 3.1, so the Jacobian $J_\phi$ exists everywhere and the change-of-variables formula gives a **valid NLL**.
> > >
> > > **TriangCFM/DDPM** instead learn a classical Flow Matching on the lower-triangular entries of SPD/correlation matrices and projects generated samples back to SPD by
> > >
> > > $\Pi(\Sigma) = (1-\alpha)\Sigma + \alpha \mathrm{I}$ with $\alpha = \frac{\epsilon - \lambda_{\mathrm{min}}(\Sigma)}{1 - \lambda_{\mathrm{min}}(\Sigma)}$ when $\lambda_{\min}(\Sigma)<\epsilon$ ($\alpha=0$ otherwise).
> > >
> > > Therefore, we have $\lambda_{\mathrm{min}}(\Pi(\Sigma)) \geq \epsilon$, e.g. set to $\epsilon = 10^{-8}$.
> > > Hence, $\Pi$ is many-to-one, i.e. not injective. Thus, its Jacobian is singular and $\log|\det J|$ is undefined. **For TriangCFM/DDPM, NLL cannot be computed.**
> > >
> > > Contrary to the NLL, the $\alpha$‑precision, $\beta$‑recall, and $\alpha,\beta$‑F1 [C] metrics operate at the sample level and are therefore model‑agnostic. This makes them suitable for evaluating baselines such as TriangCFM and TriangDDPM, which are the closest methods to DiffeoCFM.
> > >
> > > [C] Alaa et al., *How faithful is your synthetic data? sample-level metrics for evaluating and auditing generative models*, ICML 2022

---

> ### Comment · Reviewer_hktp · 2025-08-04
>
> Thank you for the detailed answer.
>
> So the statement "The projection breaks injectivity, so the log-det Jacobian (and hence NLL) is undefined." is not true for all of your methods if I got it right ?
>
> Concerning the vector field architecture, how many layers did you use ? For the solver, it seems that the authors of SPD-CFM used a different solver from the one you used (see Appendix H, page 22 of that paper).

---

> > ### Author Response · Authors · 2025-08-04
> >
> > You are correct regarding the NLL computation. Our proposed method (DiffeoCFM) has a tractable NLL, whereas some baselines (TriangCFM/TriangDDPM) do not. We include these baselines because they  share the same computational footprint as DiffeoCFM and provide simple and practical alternatives for covariance and correlation generation.
> > The $\alpha$‑precision, $\beta$‑recall, and $\alpha,\beta$‑F1 metrics operate at the sample level and are thus model‑agnostic, enabling fair comparison across all methods, including projection‑based baselines such as TriangCFM and TriangDDPM.
> >
> > Regarding SPD-CFM, we thank you for your suggestions. We reran SPD-CFM with a deeper vector field and the dopri5 ODE solver as in [A]:
> >
> > | Component        | Value |
> > |------------------|-------|
> > | vector field | 6-layer MLP, width 512 |
> > | ODE solver | dopri5 (atol=1e-5, rtol=1e-5) |
> > | Optimizer| AdamW with lr=1e-4 |
> > | | |
> >
> >
> > **New results on BNCI 2014-002:**
> >
> > Quality metric: $\alpha$,$\beta$-F1
> >
> > Classification Accuracy Score (CAS) metrics: ROC-AUC, F1
> >
> > | Method   | $\alpha$,$\beta$-F1   | ROC-AUC  | F1  | Training time (s.)
> > |:-----------------|:-----------------------------------------------------------|:-----------------------------------------------|:------------------------------------------|:--------------------------------------------------------|
> > | Real Data         | 0.67 ± 0.07                                                | 0.86 ± 0.01                                    | 0.77 ± 0.02                               | N/A                                        |
> > | DiffeoGauss                        | 0.56 ± 0.06                                                | 0.83 ± 0.01                                    | 0.75 ± 0.02                               | **0.09 ± 0.01**                                    |
> > | TriangDDPM                            | 0.15 ± 0.04                                                | 0.51 ± 0.03                                    | 0.22 ± 0.12                               | 265.24 ± 1.96                                           |
> > | TriangCFM                                   | 0.26 ± 0.04                                                | 0.54 ± 0.03                                    | 0.24 ± 0.14                               | 261.15 ± 1.15                                           |
> > | SPD-CFM  [A]                                           | **0.65 ± 0.11**                                       | 0.83 ± 0.02                                    | 0.74 ± 0.03                               | 2008.84 ± 2.28                                          |
> > | DiffeoCFM   (proposed)                         | **0.65 ± 0.07**                                       | **0.84 ± 0.02**                           | **0.76 ± 0.03**                      | 265.94 ± 4.17
> >
> >
> > **SPD-CFM and DiffeoCFM are now on par in quality and CAS metrics, but DiffeoCFM still trains ≈ 7 × faster.**
> > These new results will be added to Table 1 of the manuscript.
> >
> >
> > [A] Chen et al., *Flow Matching on General Geometries*, ICLR 2024

---

> > > ### Comment · Reviewer_hktp · 2025-08-06
> > >
> > > Thank you very much for answering all my questions and for the new experimental results.
> > >
> > > My concerns have been addressed and I have no further questions.

---

> > > > ### Author Response · Authors · 2025-08-07
> > > >
> > > > Thank you very much for the constructive discussion and for confirming that your concerns have been addressed.
> > > >
> > > > We are glad that our clarifications and the additional experimental results were helpful. We sincerely appreciate the time and care you dedicated to the review process.

---

### Official Review · Reviewer_P6e5 · 2025-07-02

**Clarity:** 3
**Significance:** 3
**Originality:** 3
**Rating:** 5
**Confidence:** 3

**Summary:**

The paper studies the problem of generating brain connectivity with Riemannian flow matching. Authors provide comprehensive descriptions on the methodology. Experiments on fMRI and EEG data demonstrate the effectiveness of the proposed work.

**Questions:**

Please see above

**Ethical Concerns:**

["NO or VERY MINOR ethics concerns only"]

**Final Justification:**

Thanks authors for addressing the question regarding connectivity matrices. After reading comments, I decide to keep original rating.

**Limitations:**

yes

**Quality:**

3

**Strengths And Weaknesses:**

Strengths:
- The problem being resolved is interesting and of great value.
- The idea of optimizing over objective in the Euclidean space from perspective of using pullback metric is interesting. It simplified the problem and better utilized the power of neural networks.
- The paper is well-written with comprehensive details.
- Experiments show the proposed work can generate decent results on real data.

Weakness:
Since the generation is on connectivity where the connectivity matrices are obtained from brain signal data (such as fMRI), the way of defining connectivity could possibly influence the "ground-truth" thus introduce bias. It would be great if authors can discuss more on it.

---

> ### Author Rebuttal · Authors · 2025-07-29
>
> We thank the reviewer for their thoughtful and encouraging feedback. We are pleased that they found the **addressed problem both interesting and valuable**, and that the proposed **use of the pullback metric to enable Euclidean optimization** was recognized as a **key strength of our approach**. We also appreciate their remarks on the **clarity and completeness** of the manuscript, as well as their **positive assessment of the experimental results on real data**.
>
> We now address the reviewer’s comment, recalling the point in their own words before providing our response.
>
>
> ## Estimation of Connectivity Matrices
> > Since the generation is on connectivity where the connectivity matrices are obtained from brain signal data (such as fMRI), the way of defining connectivity could possibly influence the "ground-truth" thus introduce bias. It would be great if authors can discuss more on it.
>
> Oracle Approximating Shrinkage (OAS) [1] and Ledoit–Wolf [2] are the state-of-the-art shrinkage estimators in recent fMRI and EEG studies [3–5], where they consistently deliver the best classification performance. Accordingly, we estimate covariance matrices with OAS and obtain correlation matrices by z-scoring the time series before applying OAS. Indeed, we chose OAS to maximise the Classification Accuracy Score (CAS) of the “Real data” upper bound in Table 1. CAS is the accuracy of a classifier trained on generated data (conditioned on disease) and tested on real data; a high upper bound gives the strongest baseline for evaluating our generative model.
>
> We agree that the estimator itself may influence the ground-truth and, hence, our results. Because our contribution lies in generative modelling rather than connectivity estimation, we fixed a community-standard pipeline to keep this factor constant.
> To clarify the limitation, we added the following to the manuscript:
> >> **Added in Section 6 Conclusions, Limitations, and Future Work:**
> A full study of how alternative connectivity definitions—e.g., partial correlation or graphical-Lasso precision—affect generation quality remains an open problem that needs to be addressed, as the choice of definition directly determines the ground truth used in the experiments.
>
> [1] Chen et al., *Shrinkage algorithms for MMSE covariance estimation*, IEEE Transactions on Signal Processing 2010
>
> [2] Ledoit et al., *A Well-Conditioned Estimator for Large-Dimensional Covariance Matrices*, Journal of Multivariate Analysis 2004
>
> [3] Kamalaker et al., *Benchmarking functional connectome-based predictive models for resting-state fMRI*, NeuroImage 2019
>
> [4] Pervaiz, et al., *Optimising network modelling methods for fMRI*, NeuroImage 2020
>
> [5] Sabbagh et al., *Predictive regression modeling with MEG/EEG: from source power to signals and cognitive states*, NeuroImage, 2022

---

> > ### Author Response · Authors · 2025-08-05
> >
> > Dear Reviewer P6e5,
> >
> > We are grateful for your thoughtful and constructive feedback. In our previous response, we implemented the following revisions, which we believe further improve the clarity and strength of the manuscript:
> > - We clarified our choice of the Oracle Approximating Shrinkage (OAS) estimator, noting its strong downstream accuracy.
> > - We added a new paragraph in §6 (Limitations & Future Work) that discusses how other connectivity definitions—partial correlation, graphical-Lasso precision, etc.—could alter the ground truth.
> >
> > Could you let us know if these points resolve your concern about connectivity bias, or if further detail would help?

---

### Official Review · Reviewer_mBJ3 · 2025-07-02

**Clarity:** 3
**Significance:** 3
**Originality:** 3
**Rating:** 4
**Confidence:** 5

**Summary:**

This paper introduces DIFFEOCFM, a novel conditional flow matching framework designed for efficient generative modelling on Riemannian manifolds, specifically targeting brain connectivity matrices (SPD and correlation matrices). DIFFEOCFM utilises global diffeomorphisms to map manifold data to Euclidean spaces, thus enabling simpler computations and avoiding expensive manifold-specific operations. Experiments were conducted on three large-scale fMRI datasets (ADNI, ABIDE, OASIS-3) and two EEG motor imagery datasets (BNCI2014-002, BNCI2015-001), demonstrating superior performance in both realism and downstream classification tasks.

**Questions:**

1. The vector field transformation in Eq. (7) relies on computing the differential of the diffeomorphism and its inverse. What are the computational costs and numerical stability issues, particularly for high-dimensional Corr matrices? Have the authors observed any instability in Dϕ(Σ) when Σ is near singular?
2. How would the method extend to manifolds without known diffeomorphic embeddings (e.g., the Stiefel manifold)? Could approximate embeddings or learned bijections (e.g., neural ODEs) be considered?
3. During back-propagation through Dϕ, did you encounter exploding/vanishing gradients?
4. The CSP maps in Figure 2(b) are presented with colour bars and topographic projections. It would be helpful to also report quantitative similarity metrics (e.g., structural similarity index) between real and synthetic filters for more rigorous comparison.
5. α-precision and β-recall are geometry-agnostic. Could authors provide a discussion of a manifold-aware version (e.g., geodesic-based recall) in future work?

**Ethical Concerns:**

["NO or VERY MINOR ethics concerns only"]

**Final Justification:**

The rebuttal response addresses previous concerns and clarifies the ambiguous part. I decided to keep my score as Borderline accept.

**Limitations:**

yes

**Quality:**

3

**Strengths And Weaknesses:**

Strengths:
Quality: The manuscript is technically sound, and the mathematical framework is well-constructed. The pullback approach significantly reduces computational overhead without compromising geometric fidelity, which is convincingly supported by theoretical propositions (Propositions 1-3).
Clarity: The manuscript is clearly structured, thoroughly described, and easy to follow, with well-defined notation and coherent transitions between sections.
Significance: DIFFEOCFM significantly advances generative modelling on Riemannian manifolds by simplifying complex geometric computations. This methodological innovation is impactful, particularly in fields like neuroimaging, where preserving manifold constraints is crucial.
Originality: The paper's primary contribution—leveraging pullback geometry via global diffeomorphisms—is novel and well-motivated, clearly delineating itself from prior works such as SPD-DDPM and traditional Riemannian CFM.

Weaknesses:
1. Key implementation and experimental details—e.g., hyperparameters and projection analyses—are buried in the appendices (J, L); a concise summary in the main text would greatly improve self-containment.
2. Algorithms 1 and 2 omit pseudocode for computing crucial terms such as Dϕ(x) and Dϕ⁻¹(z), which are necessary for understanding Eq. (7) and ensuring reproducibility.
3. No ablation is reported on ODE step size h or network depth, so the robustness–efficiency trade-off of DIFFEOCFM remains unclear.

---

> ### Author Rebuttal · Authors · 2025-07-30
>
> We thank the reviewer for their thoughtful and positive feedback. We are glad that the paper’s **technical soundness, clarity, and originality** were appreciated, that the pullback approach was recognized as **mathematically grounded and computationally efficient**, and that its relevance to neuroimaging was noted. We also appreciate the acknowledgment of our **clear exposition** and the significance of DiffeoCFM for generative modeling on manifolds.
>
> We now address the reviewer’s comments and questions, recalling each point in their own words before providing our response.
>
> ## Experimental details in Appendices
> >Key implementation and experimental details—e.g., hyperparameters and projection analyses—are buried in the appendices (J, L); a concise summary in the main text would greatly improve self-containment.
>
> We moved from the appendices key implementation details in the main text, complementing Appendices J and L, to improve readability without sacrificing completeness.
>
> First, we added the following information for the metrics:
> >> **Added in Section 4.1 Metrics:**
> We follow the CAS protocol, training a classifier on generated samples and evaluating it on real test data. Specifically, we assess classification utility using a logistic regression [...].
>
> Then, we detailed TriangDDPM and TriangCFM and the deep learning setup:
> >> **Added in Section 4.3 Baselines:**
> Since $\phi$ is not a diffeomorphism, generated matrices that do not lie on the manifold are projected back onto it. In particular, generated matrices are not necessarily positive definite, so we apply a projection [...].
>
> >> TriangDDPM, TriangCFM and DiffeoCFM employ a two-layer MLP with $512$ hidden units, trained using AdamW with a learning rate of $10^{-3}$ and batch size $64$. Training runs for $200$ epochs on fMRI and $2000$ epochs on EEG. Time integration follows the midpoint method with a step size of $10^{-3}$.
>
>
> ## Differentials $\mathrm{D} \phi(x)$, $(\mathrm{D} \phi)^{-1}(x)$: usage, cost, stability, backpropagation
> >Algorithms 1 and 2 omit pseudocode for computing crucial terms such as Dϕ(x) and Dϕ⁻¹(z), which are necessary for understanding Eq. (7) and ensuring reproducibility.
>
> >The vector field transformation in Eq. (7) relies on computing the differential of the diffeomorphism and its inverse. What are the computational costs and numerical stability issues, particularly for high-dimensional Corr matrices? Have the authors observed any instability in Dϕ(Σ) when Σ is near singular?
>
> >During back-propagation through Dϕ, did you encounter exploding/vanishing gradients?
>
> Implementing Riemannian CFM [1] with a pullback metric requires $\mathrm{D}\phi(x)$, $(\mathrm{D}\phi)^{-1}(x)$, and backpropagation through them—leading to numerical instability, gradient explosion/vanishing, and high computational cost, which demand careful implementation and tuning.
>
> This motivated DiffeoCFM, which, as shown in Propositions 1–3, is mathematically equivalent to Riemannian CFM with the pullback metric induced by a diffeomorphism $\phi$. In practice, DiffeoCFM proceeds by:
> 1. apply $\phi$ to data on $\mathcal{M}$;
> 2. train and sample a classical CFM in the Euclidean space $E$;
> 3. map generated data back to $\mathcal{M}$ via $\phi^{-1}$.
>
> These steps, detailed in Algorithms 1–2, avoid computing $\mathrm{D}\phi(x)$ or any Riemannian operations—only $\phi$ and $\phi^{-1}$ are needed, both easy to implement. It should be noted that Eq (7) is used solely to prove Propositions 1, 2, and 3.
>
> [1] Chen et al., *Flow Matching on General Geometries*, ICLR 2024
>
> [2] Lipman et al., *Flow Matching for Generative Modeling*, ICLR 2023
>
>
> ## Step size and network depth
> >No ablation is reported on ODE step size h or network depth, so the robustness–efficiency trade-off of DIFFEOCFM remains unclear.
>
> We studied the robustness–efficiency trade‑off by varying network depth and ODE step size on the fMRI ADNI dataset, reporting $\alpha,\beta$‑F1 (mean ± std), training, and sampling times over 5‑fold cross‑validation.
>
>
> |   Depth | $\alpha,\beta$-F1      | Training time (s.)   | Sampling time (s.)   |
> |-------------:|:------------|:---------------------|:---------------------|
> |            1 | **0.65 ± 0.06** | **35.07 ± 1.36**         | **3.75 ± 0.81**          |
> |            2 | 0.56 ± 0.04 | 37.23 ± 1.18         | 5.35 ± 0.54          |
> |            3 | 0.52 ± 0.05 | 44.73 ± 2.00         | 5.31 ± 0.93          |
> |            4 | 0.48 ± 0.09 | 49.60 ± 1.64         | 7.74 ± 1.28          |
>
> |   ODE step size | $\alpha,\beta$-F1  | Training time (s.)   | Sampling time (s.)   |
> |------------:|:------------|:---------------------|:---------------------|
> |      0.0001 | **0.64 ± 0.05** | 33.86 ± 2.25         | 52.12 ± 0.24         |
> |      0.001  | 0.63 ± 0.05 | 33.50 ± 1.97        | 2.17 ± 0.68          |
> |      0.01   | 0.61 ± 0.06 | 33.22 ± 1.02         | 0.20 ± 0.03          |
> |      0.1    | 0.55 ± 0.09 | 33.63 ± 0.43       | **0.06 ± 0.01**          |
>
>
> Results show that performance saturates with 1 hidden layer, while deeper networks increase computation without improving quality. For the ODE step size, $10^{-3}$ offers a good trade‑off between sample quality and sampling time. This confirms that **DiffeoCFM maintains high quality across a broad range of settings**, with a favourable robustness–efficiency balance.
>
>
> ## Extension to manifold without global diffeomorphisms
> >How would the method extend to manifolds without known diffeomorphic embeddings (e.g., the Stiefel manifold)? Could approximate embeddings or learned bijections (e.g., neural ODEs) be considered?
>
> This work targets brain‑connectivity data—SPD and correlation matrices—where known global diffeomorphisms $\phi$ enable fast, stable training via Euclidean CFM [2] on embedded data.
>
> As noted in the conclusion, DiffeoCFM requires such a diffeomorphism and does not apply to manifolds like the Stiefel. Even general algorithms [1] fail on manifolds without closed‑form geodesics. Two possible extensions are:
> 1. **Approximate geodesic operators** using retractions/inverses, as in $R$/$RL$-barycentres on Stiefel and Grassmann manifolds [3, 4].
> 2. **Ambient‑space methods** that progressively map back to the manifold without explicit retractions, e.g., landing algorithms for the orthogonal group [5].
>
> Exploring these directions is promising but beyond our present scope.
>
> [3] Kaneko, et al., *Empirical arithmetic averaging over the compact Stiefel manifold* IEEE Transactions on Signal Processing, 2012
>
> [4] Bouchard, et al., *Beyond $R$-barycenters: an effective averaging method on Stiefel and Grassmann manifolds* IEEE Signal Processing Letters, 2025
>
> [5] Ablin, et al., *Fast and accurate optimization on the orthogonal manifold without retraction*, ICML 2022
>
>
> ## Quantitative evaluation of spatial filters
> >The CSP maps in Figure 2(b) are presented with colour bars and topographic projections. It would be helpful to also report quantitative similarity metrics (e.g., structural similarity index) between real and synthetic filters for more rigorous comparison.
>
> Our study includes such a task‑level test: Table 1 reports CAS [6] when training on real or synthetic data and testing on held‑out real sets. For EEG, real vs synthetic scores match within the standard deviation, showing CSP filters from both have the same discriminative power.
>
> Figure 2 visually confirms this result: in both $\alpha$- (8–12 Hz) and $\beta$- (13–30 Hz) bands, filters from synthetic data reproduce the contralateral C3/C4 patterns of real data during right-hand vs. feet imagery [7,8], as quantified in Table 1.
>
> To clarify this point, we have added the following paragraph in the paper:
> >> **Added in Section 5.2 Neurophysiological Plausibility Study:**
> We visualize CSP spatial filters in the $\alpha$ (8–12 Hz) and $\beta$ (13–30 Hz) bands that distinguish imagined right-hand from feet movements. The filters trained on real and on generated data concentrate on the same contralateral sensorimotor regions, mirroring the close CAS scores between Real data and DiffeoCFM in Table 1. This confirms that the generative model preserves the physiologically relevant decision boundaries.
>
> Applying SSIM to EEG topographies can be misleading: (1) topographies are computed at subject/group level, making SSIM across trials impossible to compute; (2) single‑trial maps contain high non‑neural noise, so pixel‑wise metrics risk capturing artifacts rather than $\alpha$/$\beta$ ERD/ERS patterns. This motivated using CAS [6] and CSP filter inspection.
>
> [6] Ravuri et al., *Classification accuracy score for conditional generative models*, NeurIPS 2019.
>
> [7] Pfurtscheller et al., *Event-related EEG/MEG synchronization and desynchronization: basic principles*, Clinical Neurophysiology, 1999.
>
> [8] Pfurtscheller et al., *Mu rhythm (de)synchronization and EEG single-trial classification of different motor imagery tasks*, NeuroImage, 2006.
>
> ## Manifold-aware α-precision and β-recall
> >α-precision and β-recall are geometry-agnostic. Could authors provide a discussion of a manifold-aware version (e.g., geodesic-based recall) in future work?
>
> This remark echoes our conclusion: these metrics [9], standard in generative modeling, assess generated–real pairwise correlation match via the Frobenius norm $||\Sigma_{\text{real}} - \Sigma_{\text{gen}}||_F^2$, a long‑standing choice in covariance analysis.
>
> A Riemannian distance could reveal other aspects—for instance, the affine‑invariant metric compares eigenvalues in log space, making multiplicative variance changes (e.g., fixed dB shifts in EEG power) linear in distance. While promising, this would change the meaning of $\alpha$‑precision and $\beta$‑recall and is beyond our scope.
>
> [9] Alaa et al., *How faithful is your synthetic data? sample-level metrics for evaluating and auditing generative models*, ICML 2022

---

> > ### Author Response · Authors · 2025-08-05
> >
> > Dear Reviewer mBJ3,
> >
> > We thank you again for your thoughtful and constructive feedback. Guided by your suggestions, our previous response introduced the revisions below, which we believe make the manuscript clearer and stronger:
> > - To address your request for self-containment, we moved key implementation details from Appendices J & L into the main text (§4.1, §4.3).
> > - To clarify computational feasibility, we reminded that DiffeoCFM avoids the costly Jacobians $\mathrm{D}\phi$ and $(\mathrm{D}\phi)^{-1}$ while remaining theoretically equivalent to Riemannian Flow Matching with pullback metrics (Propositions 1-3).
> > - To quantify robustness–efficiency, we added new experiments with ablations on network depth and ODE step size; one hidden layer and $h = 10^{-3}$ give the best trade-off.
> > - To strengthen evaluation, we justified Classification Accuracy Scores as a quantitative test of spatial filters and explain why SSIM is ill-suited for EEG topographies.
> > - To frame future work, we detailed limits on manifolds without global diffeomorphisms and outline two extension paths (retractions, ambient-space methods), and we discuss manifold-aware $\alpha$-precision/$\beta$-recall.
> >
> > Please let us know if these updates address your concerns, or if further detail would help.

---

### Official Review · Reviewer_DK8R · 2025-07-03

**Clarity:** 4
**Significance:** 4
**Originality:** 3
**Rating:** 5
**Confidence:** 3

**Summary:**

The paper introduces DIFFEOCFM, a method designed to address the challenge of generating realistic brain connectivity matrices, such as symmetric positive definite or correlation matrices, reside on Riemannian manifolds. Traditional manifold-aware generative models incur heavy computational overhead. DIFFEOCFM uses a global diffeomorphism to map these manifolds to Euclidean space, enabling efficient application of standard generative techniques while ensuring geometric constraints are maintained. Evaluated on 3 fMRI datasetsand 2  EEG datasets, DIFFEOCFM demonstrates superior performance in quality and classification metrics than baseline methods.

**Questions:**

1) Discussion on scalability to high-dimensional manifolds would be helpful.
2) The choice of a two-layer 512-unit MLP is not fully justified. Could the authors try transformer-style vector fields or neural ODEs, and also explore alternative pullbacks?

**Ethical Concerns:**

["NO or VERY MINOR ethics concerns only"]

**Final Justification:**

I have carefully reviewed the author response and the other reviewers' comments. The responses have addressed several of my concerns, and I believe the additional comparison to SPD-CFM and the provided clarifications meaningfully strengthen the paper. I am therefore maintaining my Accept rating.

**Limitations:**

yes

**Quality:**

3

**Strengths And Weaknesses:**

Strengths:
- The paper is well motivated and well written. The integration of pullback geometry with conditional flow matching is a novel approach that simplifies generative modeling on Riemannian manifolds while preserving geometric constraints.
- DIFFEOCFM achieves state-of-the-art results on multiple fMRI and EEG datasets, with significant improvements in quality and classification metrics over baselines.
- The biological plausibility analysis adds credibility, and generated samples are neurophysiologically meaningful, supporting applications in brain disease research and classification tasks.

Weaknesses:
- The impact of diffeomorphism choice (log-Euclidean vs. affine-invariant, other Cholesky normalizations) or vector-field capacity is not discussed.
- Comparisons with other true manifold baselines such as SPD-DDPM in terms of quality and runtime would be helpful.
- Additional literature could be discussed, such as Jo et al. and Kapusniak et al., which also generate data on manifolds through learned or mixture-based interpolants.

Jo et al. Generative Modeling on Manifolds Through Mixture of Riemannian Diffusion Processes. ICML 2024

Kapusniak et al. Metric Flow Matching for Smooth Interpolations on the Data Manifold. NeurIPS 2024

---

> ### Author Rebuttal · Authors · 2025-07-29
>
> We thank the reviewer for their positive and encouraging feedback. We are glad that the **motivation, writing, and integration of pullback geometry** with conditional flow matching in our proposed method, DiffeoCFM, were appreciated. We are also pleased that the reviewer found our **state-of-the-art** results and the **biological plausibility analysis** convincing, as these are central to the impact we aim to achieve in neuroscience applications.
>
> We now address the reviewer’s comments and questions, recalling each point in their own words before providing our response.
>
> ## Impact of the choice of diffeomorphism and vector field
> > The impact of diffeomorphism choice (log-Euclidean vs. affine-invariant, other Cholesky normalizations) or vector-field capacity is not discussed.
>
> > The choice of a two-layer 512-unit MLP is not fully justified. Could the authors try transformer-style vector fields or neural ODEs, and also explore alternative pullbacks?
>
> First of all, the number of available diffeomorphisms for SPD and correlation manifolds is limited. We selected the normalized Cholesky map for correlation matrices and the logarithm map for SPD matrices because of their simplicity and ease of implementation. Then, identifying alternative mappings and systematically comparing their properties remains an open problem and we believe is beyond the scope of this work. We have added the following note about the choice of the two diffeomorphisms in Section 3.2:
>
> >> **Added in Section 3.2**: We selected these two diffeomorphisms for their simplicity and ease of implementation. However, other choices are possible. For correlation matrices, alternative parameterizations are discussed in [1]. For SPD matrices, one can use the Cholesky factor with a logarithm applied to the diagonal, leading to the log-Cholesky metric [2], which also defines a global diffeomorphism.
>
> Regarding the vector field, it is implemented as a two-layer MLP with 512 hidden units. This architectural choice is consistent with prior work [3, 4, 5], and in our experience, the results remained stable as long as the network was sufficiently large. In our setting, the pullback yields an unstructured vector representation, lacking spatial or sequential structure. As such, transformer architectures do not offer additional meaningful inductive bias while increasing model complexity. Similarly, replacing the MLP with a neural ODE backbone would duplicate the existing continuous-time formulation without a clear benefit.
>
> [1] Thanwerdas, *Riemannian and stratified geometries of covariance and correlation matrices*, Thesis 2022.
>
> [2] Lin et al., *Riemannian geometry of symmetric positive definite matrices via Cholesky decomposition*, SIAM Journal on Matrix Analysis and Applications 2019.
>
> [3] Kotelnikov et al., *TabDDPM: Modelling Tabular Data with Diffusion Models*, ICML 2023.
>
> [4] Shi et al., *Diffusion Schrödinger Bridge Matching*, NeurIPS 2023.
>
> [5] Albergo et al., *Building Normalizing Flows with Stochastic Interpolants*, ICLR 2023.
>
>
> ## Comparisons with other true manifold baselines
>
> > Comparisons with other true manifold baselines such as SPD-DDPM in terms of quality and runtime would be helpful.
>
> **SPD-DDPM** A comparison of training time of DiffeoCFM with SPD-DDPM is provided in Appendices G–H. First, we note that SPD-DDPM is designed specifically for SPD matrices (not correlation matrices) and operates entirely on the SPD manifold. As a result, it relies on expensive Riemannian training and sampling (Riemannian optimization of SPDNet, exp/log map, Riemannian sampling solver, ...), making it impractical even for medium-sized datasets (e.g., 100 samples of dimension 30). Indeed, as shown in Figure 3 (Appendix H), it is over 1000× slower than DiffeoCFM, which shares the same computational footprint as CFM, a method explicitly designed for scalability and widely adopted in computer vision. Due to this computational burden, we were unable to assess the sample quality of SPD-DDPM in our experimental setting.
>
> **SPD-CFM** We compare DiffeoCFM against SPD-CFM [6], which will be added to Table 1. We adapted the official implementation of SPD-CFM to use the same conditional prior as DiffeoCFM and evaluated on BNCI 2014‑002 by holding out each of the first two sessions in turn for testing while training on all remaining sessions. The results are shown in the next table; BNCI 2015‑001 yielded similar results.
>
> Quality metrics = $\alpha$-Precision, $\beta$-Recall, $\alpha$,$\beta$-F1
> Classification Accuracy Score (CAS) metrics: ROC-AUC, F1
>
> | Method   | $\alpha$-Precision   | $\beta$-Recall   | $\alpha$,$\beta$-F1   | ROC-AUC  | F1  | Training time (s.)
> |:-----------------|:---------------------------------------------------------|:-----------------------------------------------------|:-----------------------------------------------------------|:-----------------------------------------------|:------------------------------------------|:--------------------------------------------------------|
> | SPD-CFM          | 0.59 ± 0.08                                              | 0.59 ± 0.07                                          | 0.59 ± 0.10                                                | 0.82 ± 0.02                                    | 0.73 ± 0.03                               | 1656.78 ± 4.86                                          |
> | DiffeoCFM  (proposed)      | **0.62 ± 0.05**                                     | **0.69 ± 0.04**                                          | **0.65 ± 0.07**                                       | **0.84 ± 0.02**                           | **0.76 ± 0.03**                      | **265.94 ± 4.17**
>
> **Results interpretation** SPD-CFM [6] performs slightly worse than DiffeoCFM. Moreover, SPD-CFM is approximately 6x slower, which is expected since it relies on geodesic computation and its derivative, the Riemannian norm, and a Riemannian ODE solver. In contrast, DiffeoCFM first maps data to a Euclidean space via the diffeomorphism $\phi$, trains a standard Euclidean CFM (which is very fast), and then maps generated data back to the manifold via $\phi^{-1}$. We will complete Table 1 of the manuscript with this additional baseline.
>
> [6] Chen et al., *Flow Matching on General Geometries*, ICLR 2024
>
> ## Additional literature
>
> > Additional literature could be discussed, such as Jo et al. and Kapusniak et al., which also generate data on manifolds through learned or mixture-based interpolants.
>
> We have added a brief discussion of [7] and [8] in the Section 3.3 Related-work:
>
> >> **Added in Section 3.3**: References [7] and [8] explore more general settings by learning bridge matches on arbitrary manifolds or data-driven Riemannian metrics, whereas our approach focuses on Riemannian geometries defined via known pullback diffeomorphisms.
>
> Key distinctions are summarized below:
> * Reference [7] addresses a more general setting, learning bridge matching on manifolds where Riemannian logarithms may be undefined or costly to compute. They rely on the spectral decomposition of the Laplace-Beltrami operator to define distances and then reduce the simulation budget during training. It should be noted that they still require the Riemannian norm in their loss function during training and the geodesic (Riemannian exponential mapping) during sampling. In contrast, our work focuses on pullback manifolds such as SPD and correlation spaces, where known diffeomorphisms enable fast and scalable training and sampling without complex Riemannian operators.
> * Reference [8] replaces straight interpolants with curved ones derived from a learned Riemannian metric. While complementary, this line of work is orthogonal to our focus on leveraging existing pullback maps (and their associated Riemannian metrics) for efficient training and generation.
>
> [7] Jo et al., *Generative Modeling on Manifolds Through Mixture of Riemannian Diffusion Processes*, ICML 2024.
>
> [8] Kapusniak et al., *Metric Flow Matching for Smooth Interpolations on the Data Manifold*, NeurIPS 2024
>
>
> ## Scalability to high-dimensional manifolds
> > Discussion on scalability to high-dimensional manifolds would be helpful.
>
> First, learning generative models when the ambient dimension $d$ approaches the sample size $n$ is statistically hard. For example, Theorem 3.1 of [9] shows that the estimation error of neural networks trained with diffusion decays no faster than $n^{-1/d}$, which is the curse of dimensionality. In our fMRI experiments we adopted the MSDL atlas with 39 regions, giving a manifold dimension of $39 \times 38 / 2 = 741$, which remains below the number of available samples (e.g. 1900 ADNI scans). Using finer atlases would push us into the $n<d$ regime where theory predicts over-fitting and degraded generative quality. Designing priors or low-rank parameterisations to tame this regime is an important avenue for future work.
>
> We have inserted the following clarification into the Section 6:
>
> >> **Added in Section 6 Conclusions, Limitations, and Future Works:**
> Learning brain connectivities at higher parcellation granularity is challenging: the manifold dimension grows quadratically with the number of regions, while the number of required samples increases exponentially with dimension [9]. Addressing this high-dimensional regime, e.g. through latent representations, remains an important direction for future work.
>
> Second, from a computational standpoint, DiffeoCFM inherits the scalability of CFM, which was originally developed for high-dimensional data and large sample sizes in vision tasks. In contrast, manifold-based methods like SPD-DDPM scale poorly in practice due to their reliance on costly geometric operations.
>
> [9] Oko, et al. *Diffusion models are minimax optimal distribution estimators*. ICML 2023

---

> > ### Comment · Reviewer_DK8R · 2025-08-06
> >
> > I appreciate the authors’ response and clarification. The added SPD-CFM comparison, the clarifications on diffeomorphisms/Jacobians and NLL, and the new robustness ablations on step size and depth in the authors' other responses resolve most of my concerns. What remains, such as higher-parcellation scalability and exploring alternative pullbacks/architectures, reads as natural future work. I’m keeping my Accept rating.

---

### Note · Authors · 2025-08-11

We thank the reviewers for the constructive discussion that helped improve the manuscript. We are encouraged that they highlighted the paper’s novelty, the clarity of the writing, the well-constructed mathematical framework, the efficiency gained by the pullback approach while preserving manifold constraints, and the strength and relevance of the empirical results.

In the manuscript, we studied generative modeling of Symmetric Positive Definite (SPD) and Correlation matrices via **DiffeoCFM**, a Riemannian Conditional Flow Matching (CFM) method under pullback geometry. We proved that DiffeoCFM is equivalent to Euclidean CFM after a known diffeomorphism $\phi$. Hence, DiffeoCFM preserves manifold constraints with the speed and stability of Euclidean CFM while achieving strong performance on five neuroimaging datasets (EEG and fMRI).

**Baselines & metrics.** We added a **true-manifold SPD-CFM** baseline with aligned training/solver settings for fairness. On the SPD manifold, **DiffeoCFM matches SPD-CFM in performance while being ~7× faster**. Diffusion baselines on manifolds remain orders of magnitude slower, reinforcing the practicality of our approach.
We clarified that negative log-likelihood (NLL) is not comparable because some baselines are non-injective and lack a valid log-density. This explains, we use sample-level $\alpha$-precision/$\beta$-recall and CAS metrics for fair, side-by-side evaluation across all methods.

**Robustness & details.** We added **ablations on vector-field depth and ODE step size**; one hidden layer suffices, and a $10^{-3}$ fixed step offers the best quality/speed trade-off. We moved essential implementation details (metrics, baselines, training setup) from the appendix into the main text for clarity.

**Scope & limitations.** We discussed in more detail three limitations as directions for future work: (i) using finer fMRI parcellations, which raises manifold dimension and may require priors or low-rank parameterizations to avoid small-$n$/large-$p$ issues; (ii) extending beyond globally diffeomorphic settings (e.g., Stiefel) via retractions or ambient-space “landing” strategies; and (iii) exploring alternative connectivity estimators beyond the OAS used here.

---

### Decision · Program_Chairs · 2025-09-17

**Decision:**

Accept (poster)

**Comment:**

This paper studies flow matching on matrix manifolds using pullback metrics induced by global diffeomorphisms from Euclidean spaces. Motivated by generating brain functional connectivity matrices, which are symmetric positive definite, to study population heterogeneity and disease, the proposed method, DiffeoCFM, enables conditional flow matching on the matrix manifold. The approach is validated on large-scale fMRI and EEG datasets.

Reviewers found that this paper is well-motivated and well-written, the idea is interesting, the applications are important from the scientific point of view, and the practical performance is strong. Concerns include the lack of literature review, baseline methods, and technical details in experiments, many of which were resolved during the rebuttal and discussion phase.

Finally, reviewers reached an agreement that this is a good paper, and I agree and recommend acceptance. However, the authors should revise the paper for the camera-ready version carefully based on reviewers' comments and promised changes during the discussion phase.